



## Biomass burning contribution to regional PM$_{2.5}$ during winter in the North China

Zheng Zong[1,6], Xiaoping Wang[2], Chongguo Tian[1,*], Yingjun Chen[3,*], Lin Qu[4], Ling Ji[4], Guorui Zhi[5], Jun Li[2], Gan Zhang[2]

[1] Key Laboratory of Coastal Environmental Processes and Ecological Remediation, Yantai Institute of Coastal Zone Research, Chinese Academy of Sciences, Yantai, 264003, China

[2] State Key Laboratory of Organic Geochemistry, Guangzhou Institute of Geochemistry, Chinese Academy of Sciences, Guangzhou, 510640, China

10 [3] Key Laboratory of Cities' Mitigation and Adaptation to Climate Change in Shanghai (CMA), College of Environmental Science and Engineering, Tongji University, Shanghai, 200092, China

[4] Yantai Oceanic Environmental Monitoring Central Station, SOA, Yantai, 264006, China

[5] Chinese Research Academy of Environmental Sciences, Beijing, 100012, China

[6] University of Chinese Academy of Sciences, Beijing, 100049, China

* Corresponding author:

Chongguo Tian, Yantai Institute of Coastal Zone Research, CAS. Phone: +86-535-2109-160; Fax: +86-535-2109-000; e-mail: cgtian@yic.ac.cn

Yingjun Chen, College of Environmental Science and Engineering, Tongji University. Phone:
20 +86-535-2109-160; Fax: +86-535-2109-000; e-mail: yjchentj@tongji.edu.cn

## Abstract

Source apportionment of fine particles (PM$_{2.5}$) at a background site in the North China in winter,

25 2014 was assessed by statistical analysis on the chemical species grouped by the trajectory clusters, radiocarbon ($^{14}$C) measurement and the Positive Matrix Factorization (PMF) modeling linked with the $^{14}$C measurement. During the sampling period, the concentration of PM$_{2.5}$ was 77.6 $\pm$ 59.3 µg m$^{-3}$, and the sulfate concentration was the highest, followed by nitrate, organic carbon (OC), elemental



carbon (EC) and ammonium, respectively. Demonstrated by the backward trajectory, more than half of $PM_{2.5}$ was found from the Beijing-Tianjin-Hebei (BTH) region, followed by the Mongolia and the Shandong Peninsula. The cluster analysis showed that $PM_{2.5}$ from the Shandong Peninsula had an obvious signal of biomass burning emission, while that from the BTH region showed vehicle

emission pattern. The finding was further confirmed by the radiocarbon measurement of OC and EC in two merged samples selected from a successive synoptic process. The $^{14}C$ measurement indicated that biogenic and biomass burning emission contributed 59% and 52% to OC and EC concentrations when air masses originated from the Shandong Peninsula, and the contributions fell to 46% and 38%, respectively, when the prevailing wind changed and came from the BTH region. In addition,

minimum deviation of the source apportionments from PMF results and $^{14}C$ measurement was used as the optimal choice of the model exercises. Here, two minor overestimations with the same range (3%) suggested that the PMF results provided a reasonable source apportionment of regional $PM_{2.5}$ in the North China during winter. Based on the results above, eight main sources were identified, of which, coal combustion, biomass burning and vehicle emissions were the largest contributors of

$PM_{2.5}$, accounting for 29.6%, 19.3% and 15.8%, respectively. Compared with the overall source apportionment, the contribution of vehicle emission increased slightly when air masses came from the BTH region, the contribution of mineral dust and coal combustion rose obviously when air masses were from the Mongolia with high speed, and biomass burning became the dominant contributor when air masses carried from the Shandong Peninsula. As the largest contributor to $PM_{2.5}$

in winter of North China, coal combustion has been identified as the most leading emission sector to be controlled for improving the air quality by the government. Vehicle emission contributed significantly to the $PM_{2.5}$ levels in the BTH region, which has also been considered to control as the



second major emission sector. Biomass burning emission was highlighted in the present study because of its dominant contribution to PM$_{2.5}$ burden in the Shandong Peninsula. Some suggests were provided to wake farmers from agricultural residue burning in household and field.

**Keywords:** Source apportionment, PMF, $^{14}$C measurement, PM$_{2.5}$

## 1 Introduction

In recent years, air pollution has become a top environmental issue in China, and the main concern is on the fine particulate matter less than 2.5 micrometers in diameter (PM$_{2.5}$) (Huang et al.,

2014;Sheehan et al., 2014). Fine particulate aerosols have a strong adverse effect on human health, visibility, and directly or indirectly affect weather and climate. The negative effects on the public health, including the damage to the respiratory and cardiovascular systems, the blood vessels of the brain, and the nervous system, have triggered both public alarm and official concern in China (Kessler, 2014). In response to this great concern, the Chinese government has introduced the Action

Plan for Air Pollution Prevention and Control (2013−17), which aims at marked improvements in the air quality up to 2017. In the plan, the severest supervision for the improvement is a reduction of 25% in the annual average concentrations of PM$_{2.5}$ by 2017. It has been applied in the North China because the region has become the most severely polluted area in China, characterized by increasingly frequent events and regional expansion of extreme air pollution in recent years (Hu et al.,

2015;Boynard et al., 2014;Quan et al., 2014).

Basically, the key point of reducing PM$_{2.5}$ concentrations is to control its sources. Reliable source identification and quantification are essential for the development of effective political abatement strategies. However, the sources of PM$_{2.5}$ typically emit a mixture of pollutants, including



gas and particle phases, which would be mixed further in the atmosphere and can undergo chemical transformations prior to impacting on a specific receptor site, making it difficult to quantify the impacts. This encourages researchers to use more techniques to quantify the contribution of individual sources to $PM_{2.5}$ concentrations, such as the Positive Matrix Factorization (PMF),

Chemical Mass Balance (CMB), organic tracers, and stable carbon isotopes, etc. Whereas, these different approaches often result in source contributions that can differ in magnitude and/or are poorly correlated, and which one is more reliable cannot be determined (Balachandran et al., 2013). Recently, radiocarbon ($^{14}C$) measurement has been used as a powerful tool to quantify unambiguously the fossil and non-fossil contribution of carbonaceous aerosols in China (Liu et al.,

2014;Zhang et al., 2015;Liu et al., 2013). The method provides a chance to make a more reliable source apportionment of $PM_{2.5}$ by linking with other methods although it focuses only on carbonaceous aerosols.

      In present study, a more reliable source apportionment of $PM_{2.5}$ on the regional scale in the North China in winter was provided using PMF simulation, in which the source contribution of

carbonaceous species was confirmed by the radiocarbon measurement. The effort is vital for the development of efficient mediation policies to achieve the improvement in air quality in the North China because the regional source apportionment cannot be replaced by that extensively focused in the metropolitan areas such as Beijing(Zhang et al., 2013), Tianjin(Gu et al., 2011), Jinan(Gu et al., 2014), and others within the North China. Thus, we collected continuously aerosol samples with high

intensity on Qimu Island in winter to assess the source apportionment of $PM_{2.5}$. The objectives of this study were (1) to determine the regional-scale concentration burden and the chemical composition of $PM_{2.5}$ in the North China, (2) to distinguish the source signals based on the chemical composition





grouped according to the trajectory clusters, and 3) to assess the source apportionment of $PM_{2.5}$ using the PMF model linked with $^{14}C$ measurement.

## 2 Materials and methods

### 2.1 Sampling site and sample collection

The sampling campaign was conducted, from 3 Jan to 11 Feb, 2014, at the Longkou Environmental Monitoring Station of the State Ocean Administration of China (37 °41′N, 120 °16′E) on the Qimu Island. The island is extended to Bohai Sea in the westward direction, and surrounded by sea on other three sides, as shown in Fig. 1. And the sampling site is located in about 15 km northwest from the Longkou urban district and 300 km southeast from the Beijing-Tianjin-Hebei

(BTH) region. $PM_{2.5}$ at the sampling site exhibited largely a regional pollution signal during the sampling period,   because the air masses were mainly carried from the BTH region and the Mongolia subjected by the East Asian winter monsoon as illustrated by the backward trajectories in Fig. 1, which will be elaborated later.

A total of 76 $PM_{2.5}$ samples were collected continuously on quartz fiber filters (Whatman,

QM-A, 20.3 $\times$ 25.4 cm$^2$, heated at 450 ℃ for 6 h before use) using a Tisch high volume sampler at a flow rate of 1.13 m$^3$ min$^{-1}$ during the sampling period. The duration for each sample was 12 h, from 06:00–18:00 and from 18:00–06:00 (local time) the next day. Before and after each sampling, quartz fiber filters went through a 24 h equilibration at 25±1 $^o$C temperature and 50±2% relative humidity, and then were analyzed gravimetrically using a Sartorius MC5 electronic microbalance. Each filter

was weighed at least three times. Acceptable difference among the repetitions was less than 10 μg for a blank filter and less than 20 μg for a sampled filter. After weighing, loaded filters were stored in a refrigerator at -20 ℃ until chemical analysis. Also, field blank filters were collected to subtract the





possible contamination that occurred during or after sampling.

## 2.2 Chemical analysis

2.2.1 OC and EC

Organic carbon (OC) and elemental carbon (EC) were analyzed by a Desert Research Institute

(DRI) Model 2001 Carbon analyzer (Atmoslytic Inc., Calabasas, CA) following the Interagency

Monitoring of Protected Visual Environment (IMPROVE) thermal/optical reflectance (TOR)

protocol (Chow et al., 1993). A punch of 0.544 cm$^2$ from each quartz filter was heated to produce

four fractions (OC1, OC2, OC3 and OC4) in four temperature steps (140, 280, 480, 580 ℃) under a

non-oxidizing helium atmosphere and then in 2% $O_2$/98% He atmosphere at 580 ℃ (EC1), 740 ℃

(EC2), 840 ℃ (EC3) for the EC fractions. At the same time, pyrolyzed organic carbon (POC) was

produced in the inert atmosphere, which decreased the reflected light to correct for charred OC. The

concentrations of OC and EC were obtained according to the IMPROVE protocol, OC = OC1 + OC2

+ OC3 + OC4 + POC and EC = EC1 + EC2 +EC3 –POC. The detection limits of the method for OC

and EC were 0.82 and 0.20 μg cm$^{-2}$, respectively. In addition, blank filters and replicate samples

were examined simultaneously after analyzing a batch of 10 samples in order to obtain their inherent

concentrations on the filter and to evaluate measurement accuracy, respectively. In this study, the

contributions of OC and EC from blank filters were < 3.5 and 0.6% of their respective average

concentrations. Furthermore, comparison with average values from replicate analyses showed a good

precision with relative deviations of 5.9%.

2.2.2 Water-soluble ions and metal elements

Two punches with 47 mm diameter were cut off from each quartz fiber filter, of which, one was

subjected to Milli-Q water extraction for ionic measurement and the other was induced acid digestion



for elemental measurement. The concentrations of water soluble ions ($Na^+$, $NH_4^+$, $K^+$, $Mg^{2+}$, $Ca^{2+}$, $Cl^-$,

$NO_3^-$ and $SO_4^{2-}$) were determined by ion chromatograph (Dionex ICS3000, Dionex Ltd., America)

based on the analysis method (Shahsavani et al., 2012). The concentrations of metal elements

(including Ti, V, Mn, Fe, Co, Ni, Cu, Zn, As, Cd and Pb) were estimated on the basis of inductively

5 coupled plasma coupled with mass spectrometer (ICP-MS of ELAN DRCII type, Perkin Elmer Ltd.,

Hong Kong) following the previous method (Wang et al., 2006). The detection limit of water-soluble

ions was 10 ng/ml with the error < 5%, and 1ml RbBr of 200 ppm was put in the solution as internal

standard before analysis. Similarly, the resolution of ICP-MS ranged from 0.3 to 3.0 amu with a

detection limit lower than 0.01 ng/ml, and the error < 5%. Element Indium (In) of 5 ppb, as the

10 internal standard, was put in the solution before analysis.

2.2.3 Radiocarbon measurement

To achieve more radiocarbon information on carbonaceous fractions in $PM_{2.5}$, OC was split into

water soluble organic carbon (WSOC) and water insoluble organic carbon (WIOC) fractions. WSOC

was extracted from a punch filter by Milli-Q water as described in previous study (Liu et al., 2014),

15 and was quantified as total dissolved organic carbon in solution by a total organic carbon (TOC)

analyzer (Shimadzu TOC-VCPH, Japan). WIOC was quantified by OC given by the TOR protocol

subtracting WSOC.

[14]C measurement of WSOC, WIOC and EC was performed using the OC/EC separation system

(Liu et al., 2014). Briefly, the extracted Milli-Q water was freeze dried, and the residue was

20 re-dissolved and transferred to a pre-combusted quartz tube. Then the quartz tube was combusted at

850 ℃ and WSOC was converted into $CO_2$. The extracted filters were isolated at 340 ℃ for 15 min

for WIOC, after a flash heating of 650 ℃ for 45 s for minimizing the charring. After the separation,





the filters were heated at 375 ℃ for 4 h to remove the charring, and then oxidized under a stream of pure oxygen at 650 ℃ for 10 min to analyze EC fraction. Finally, the corresponding evolving $CO_2$ (WSOC, WINSOC and EC) was cryo-trapped and reduced to graphite at 600 ℃ for accelerator mass spectrometry (AMS) target preparation. The preparation of graphite targets for AMS analysis was

performed using the graphitization line at the Guangzhou Institute of Geochemistry, CAS. The ratios of $^{14}C/^{12}C$ in the graphite samples were determined through a NEC compact AMS.

In order to compensate for the excess $^{14}C$ caused by the nuclear bomb testing in the 1950s and 1960s, the $f_m$ given by AMS was further converted into the fraction of contemporary carbon ($f_c$). The $f_c$ values in the samples were defined as $f_c = f_m/1.10$ for EC, $f_c = f_m/1.06$ for OC, and the fraction of

fossil ($f_f$) was defined as $f_f = 1-f_c$ (Zong et al., 2015).

**2.3 Data analysis methods**

The hybrid single-particle Lagrangian integrated trajectory (HYSPLIT) model was used to generate 48-h backward trajectories with 12 h intervals. The trajectories were calculated for air masses starting from the sampling site at 500 m above ground level. Finally, a total of 152

trajectories were generated and these trajectories were bunched into three clusters. The observed chemical components of $PM_{2.5}$ carried by the three air masses were compared with each other to assess their potential sources.

PMF v5.0 provided by the US EPA was utilized to assess the source apportionment in this study. PMF is a multivariate factor analysis tool, which assumes that concentrations at a receptor site are

supported by the linear combinations of different source emissions. Thus, measured mass concentrations of selected species can be mostly expressed as (Paatero et al., 2014):

$$x_{ij} = \sum_{k=1}^{p} g_{ik} f_{kj} + e_{ij} \tag{1}$$



where $x_{ij}$ is the measured concentration of the jth species in the ith sample, $f_{kj}$ is the profile of jth

chemical species emitted by the kth source, $g_{ik}$ is the amount of mass contributed by kth source to the

ith sample, and $e_{ij}$ is the residual for each samples/species. The matrixes of g and f are determined by

minimizing an objective function (Paatero et al., 2014).

5        After determination and interpretation of these factor profiles, source contributions identified by

the model were merged into two groups according to fossil and contemporary carbon sources. Then

the contribution fraction of fossil or contemporary carbon sources to OC and EC was compared with

the results derived from [14]C analysis for the specified sample to confirm the model results.

$$R_{ij} = \sum_{k=1}^{n} g_{ik} f_{kj} \bigg/ \sum_{k=1}^{p} g_{ik} f_{kj} \qquad (2)$$

10      Where R is the contribution fraction, and matrixes of g and f are the same as in eqn(1). The

subscripts i is a specified sample, j is species of OC or EC. n is the number of fossil or contemporary

carbon sources and p is the number of all sources.

**3 Results and discussion**

**3.1 General characteristics of PM$_{2.5}$ and chemical components**

15      Table 1 lists a statistical summary of the concentrations of PM$_{2.5}$, water-soluble ions,

carbonaceous species and metal elements during the sampling period. As shown, the mean

concentration of PM$_{2.5}$ was 77.6 ±59.3 μg m$^{-3}$, and the maximum value was 305 μg m$^{-3}$, which were

more than two and eight times higher than the grade I national standards (35 μg m$^{-3}$, Ministry of

Environmental Protection of China: GB 3095-2012, www.zhb.gov.cn, 2012-02-29). Although the

20      level of PM$_{2.5}$ concentration on Qimu Island was higher than the national standard, but it was much

lower than that observed in winter in megacities of the North China, such as in Beijing (208 μg m$^{-3}$

of PM$_{2.1}$ in 2013)(Tian et al., 2014) and Tianjin (221 μg m$^{-3}$ in 2013)(Han et al., 2014). Besides, the





concentration level was significantly higher than the result (42.4 µg m$^{-3}$ in winter, 2012) measured at

a national station for background atmospheric monitoring on Tuoji Island, located on the Bohai Strait

(Wang et al., 2014). The obviously high concentration of PM$_{2.5}$ on Qimu Island was possibly

attributed to the short distance between the sampling site and emission source on the Shandong

Peninsula (see Fig. 1) and strong deposition of particles due to high precipitation over the Bohai Sea

in the winter of 2012 (Zhang et al., 2014).

For the PM$_{2.5}$ components, water-soluble inorganic species (WSIS) were the dominant species,

accounting for 37 ± 16 % of the PM$_{2.5}$ mass concentrations. Among the ionic concentrations, sulfate

ranked the highest with a mean of 14.2 ± 18.0 µg m$^{-3}$, followed by nitrate (11.9 ± 16.4 µg m$^{-3}$) and

ammonium (3.11 ± 2.14 µg m$^{-3}$). The sum of the three secondary inorganic aerosols constituted the

majority (81 ± 12 %) of the total WSIS concentrations. In addition, the average concentrations of OC

and EC were 6.85 ± 4.81 and 4.90 ± 4.11 µg m$^{-3}$, accounting for 9.2 ± 2.1 % and 6.4 ± 1.8 % of the

PM$_{2.5}$ concentrations, respectively. Total concentrations of analyzed metal elements were 665 ± 472

ng m$^{-3}$, accounting for 0.93 ± 0.50 % of the PM$_{2.5}$ mass concentration. Among the measured metal

elements, the concentration of Fe (408 ± 285 ng m$^{-3}$) was the highest, followed by Zn (107 ± 142 ng

m$^{-3}$), and Pb (88.4 ± 85.7 ng m$^{-3}$).

The relative contribution of sulfate, nitrate and ammonium to the PM$_{2.5}$ at the sampling site

was obviously higher than that in the cities, such as Beijing and Tianjin within the North China,

while the organic matter was obviously lower. The high contribution agreed with the regional scale

emission of their precursors in the North China, as reported that the emission amounts of SO$_2$, NO$_2$

and NH$_3$ were about 10, 5, and 5 times higher compared to OC in the region (Zhao et al., 2012). The

finding was also in agreement with that measured at Changdao Island (Feng et al., 2012). The island,

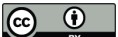



located at the demarcation line between Bohai Sea and Yellow Sea, is a resort with little industry at

about 7 km north from the Shandong Peninsula (Feng et al., 2012). Measurement at the island was

treated as providing patterns of atmospheric outflow and regional pollution in North China (Feng et

al., 2012;Feng et al., 2007). It suggested that our measurement also provided a regional signal of

$PM_{2.5}$ pollution in the North China. Besides, sulfate was the largest contributor of $PM_{2.5}$ as

aforementioned. The highest contribution is usually regarded as a regional pollution signal in winter

because of the lack of fast conversion rate of $SO_2$ to sulfate in cloud or aerosol droplet processes and

oxidation reaction via OH free radical in low temperature condition in source areas (Hu et al., 2015).

This also indicated that our measurement reflected largely a pollution pattern on a regional scale,

rather than in source areas.

**3.2 Source signals based on cluster analysis**

As shown in Fig. 1, the 48-h back trajectory clusters indicated that more than half of the air

masses (54%) during the sampling period were from the BTH region, defined as cluster 1, followed

by the air masses from the Mongolia (35%, cluster 2). Air masses of these two types traveled about

200 and 250 km, respectively, over the Bohai Sea before arriving at the sampling site. Thus, the

atmospheric pollutants carried by the two kind air masses were mixed well during the transport,

showing regional pollution signals. Only a small part of air masses (11%) were from the Shandong

Peninsula (cluster 3), reflecting potentially a mixed contribution of local and regional sources from

south area of the sampling site. To reveal the pollution patterns and source signals of $PM_{2.5}$ carried by

air masses from the three different regions, chemical species of $PM_{2.5}$ were grouped according to the

three trajectory clusters as listed in Table 2.

Generally, mean test showed that the concentration levels of $PM_{2.5}$ and its most species in



cluster 1 and cluster 3 were insignificant differences ($p > 0.05$) and statistically higher than that in

cluster 2 ($p < 0.01$) as shown in Table 2. The patterns were consistent with the spatial distributions of

their emissions and concentrations in the North China as reported the stronger emission and the more

serious pollution in the BTH region and Shandong Province compared with that in Inner-Mongolia

and Liaoning (Zhao et al., 2012; Yang et al., 2011). And the pollution in BTH region may be more

serious in that it has much longer distance from the sampling site compared with Shandong Peninsula,

while their concentration levels of $PM_{2.5}$ were insignificant different. In addition, the mean wind

speed of cluster 2 was 7.60 m s$^{-1}$, which was markedly higher than that of cluster 1 (4.79 m s$^{-1}$) and

cluster 3 (4.86 m s$^{-1}$). The wind speeds were yielded by averaged hourly moving distance of those air

masses during 48 hours. The higher wind speed of cluster 2 resulted partly in the lower $PM_{2.5}$ level at

the sampling site since that high wind speed could provide a favorable diffusion condition for

atmospheric pollutants.

       Some anomalies compared with previous discussion provided different source signals among

the clusters. For instance, the concentration of K$^+$ was significantly higher in cluster 3 than that in

cluster 1, while Ti concentration was obviously lower. This reflected the relative high emission of K$^+$

in the Shandong Peninsula and Ti in the BTH region from natural sources and anthropogenic

activities. In contrast, the concentration of Na$^+$ in cluster 2 was markedly higher than that in cluster 1

and cluster 3, showing the large contribution of sea salt particles generated by sea spray under high

wind speed in cluster 2. It suggested that sea salt sources could not be ignored due to that the

sampling site was close to the Bohai Sea.

       Sea salt emission is comprised of Cl$^-$, SO$_4^{2-}$, Na$^+$, K$^+$, Mg$^{2+}$ and Ca$^{2+}$. The amounts of the

chemical species from sea salt emission can be sustained with Na$^+$ as the tracer of sea salt, so the





amount from non sea salt (nss-) emission can be expressed as nss-X = X− $[Na^+] \times a$, where X

indicates the $Cl^-$, $SO_4^{2-}$, $K^+$, $Mg^{2+}$ and $Ca^{2+}$ concentrations, and a is the typical equivalent

concentration ratio of the corresponding species to $Na^+$, such as $Cl^-/Na^+$ (1.80), $SO_4^{2-}/Na^+$ (0.25),

$K^+/Na^+$ (0.036), $Mg^{2+}/Na^+$ (0.12) and $Ca^{2+}/Na^+$ (0.038) in average seawater (Ni et al., 2013). If the

calculated concentration of no sea salt (nss-) chemical species is negative, then there is no the species

excess exists. According to the calculation, for corresponding total chemical concentration levels

grouped in clusters from 1 to 3, nss-$Cl^-$ accounted for 55 ± 29%, 19 ± 24% and 77 ± 10%; nss-$SO_4^{2-}$

accounted for 99 ± 2%, 95 ± 4% and 99 ± 0.3%; nss-$K^+$ accounted for 98 ± 3%, 89 ± 9% and 99 ±

0.3%; nss-$Ca^{2+}$ accounted for 95 ± 4%, 91 ± 10% and 96 ± 3%, respectively. In this study, marked

contributions of nss-emission sources to the chemical concentrations were found although the values

were possibly underestimated since total $Na^+$ amount does not necessarily originate from sea salt

alone, but could partially come from dust and burning sources(Zhang et al., 2013). And the loss of

$Cl^-$ particles due to chloride depletion mechanism further supported the underestimation of $Cl^-$. The

contributions of nss-source were lower in cluster 2 than that in cluster 1 and 3, which was attributed

to the high emission of sea spray resulting from high wind speed in cluster 2. Generally, $K^+$ is often

used as a tracer for biomass burning. The high $K^+$ concentration and the largest contribution of

nss-$K^+$ in cluster 3 indicated obviously high emissions associated closely with agricultural burning in

the Shandong Peninsula. The finding agreed with the fact that Shandong is the largest producer of

crop residues in the North China (Zhao et al., 2012) and biomass burning is one of the important

sources of inorganic and organic aerosols in the Bohai sea atmosphere (Feng et al., 2012;Wang et al.,

2014). The contribution of nss-$Mg^{2+}$ to total magnesium concentration was less 4% for the all

clusters, indicating the specie came mostly from sea salt emission. The mass ratio of $Mg^{2+}$ to $Na^+$



was 0.07 $\pm$ 0.06, 0.06 $\pm$ 0.03 and 0.06 $\pm$ 0.03 for clusters from 1 to 3, respectively. The ratio was less

than 0.23, also demonstrating that $Mg^{2+}$ mostly came from sea salt source (Zhang et al., 2013).

Enrichment Factor (EF) method with an abundance element associated with crustal elements is

often used to assess enrichment characteristics of various elements in $PM_{2.5}$. In the present study, Fe

was used as the reference element for the assessment because other two major crustal elements (Si

and Al) could not be measured due to the interference from the quartz fiber substrate of the samples.

EF of each element was calculated relative to the average crustal rock composition of Fe: EF =

$(X/Fe)_{Aerosol}/(X/Fe)_{Crust}$, where $(X/Fe)_{Aerosol}$ and $(X/Fe)_{Crust}$ were the ratios of mean concentration of

target elements and Fe in $PM_{2.5}$ and continental crust, respectively. By convention, an EF value close

to unity suggests that the element is dominantly contributed by natural soil dust or the contribution of

anthropogenic sources is not significant, and the value much higher than 10 indicates that the

element is predominantly originated from human activities, such as combustion, automobile,

industrial emissions, agricultural activities and etc. (Shah et al., 2012). An EF with a range of 1 to 10

indicates a hybrid contribution from crustal and anthropogenic sources. Here, except for the EF of Ni

(9.17 $\pm$ 5.11) in cluster 3, the average EF of Ti, Ni, Cu, Zn, As, Cd and Pb in all clusters was greater

than 10, showing severe influence from anthropogenic activity. The average EF in the three clusters

also reflected that Co had a strong nature source, and the other elements (V and Mn) in all three

clusters and Ni in cluster 3 were supported from crustal dust and anthropogenic activities.

The ratios of OC/EC and $NO_3^-/nss\text{-}SO_4^{2-}$ were used to assess source signals of the three

clusters. It has been found that low temperature burning, such as agricultural residues burning, emits

more OC compared with high temperature burning, e.g. vehicle exhaust. Thus, the ratio of OC to EC

is often used to evaluate relative contribution of low and high temperature burning emission. The





OC/EC ratios were 1.41 ±0.30, 1.47 ±0.29 and 2.14 ±0.50 for the clusters from 1 to 3, respectively.

Mean test showed that the ratios were insignificant difference between cluster 1 and cluster 2 at a

95% confidence level, and the ratios of two groups were statistical low compared with that of cluster

at the same confidence level. This suggested that low temperature burning emission contributed

obviously in cluster 3, while high temperature burning emission was distinct in cluster 1 and 2.

Besides, mobile sources exhaust more $NO_x$ than $SO_2$, while stationary sources, such as coal-fired

power plants emit more $SO_2$ than $NO_x$. The two precursors convert into sulfate and nitrate in the

atmosphere, and show different ratio of $NO_3^-/SO_4^{2-}$ from the two type sources. Therefore, the ratio is

often used as an indicator of the relative importance of mobile vs. stationary sources of sulfur and

nitrogen in the atmosphere (Liu et al., 2014;Zhao et al., 2013). In this study, after deducting the

contribution of sea salt to sulfate, the mean ratios of $NO_3^-/nss-SO_4^{2-}$ were 0.96 ± 0.31, 0.47 ± 0.24

and 0.64 ±0.14 for cluster 1 to 3, respectively. Mean test showed that the ratios of the three clusters

exhibit significant difference each other at a 95% confidence level. The significant high ratio in

cluster 1 suggested the most important contribution of mobile source in the BTH region among the

three regions, followed by Shandong Peninsula (cluster 3). The ratio of $NO_3^-/nss-SO_4^{2-}$ in cluster 1

was within the range of that in large cities, such as Beijing (1.20), Tianjin (0.73), and Shijiazhuang

(0.76), the capital of Hebei province (Zhao et al., 2013), reflecting a hybrid contribution from the

BTH region. The value in cluster 2 was slightly lower than that in winter in Chengde (0.55), one city

located in the northern mountainous area of Hebei Province (Zhao et al., 2013). It indicated more

obvious contribution of stationary source emissions in the following region, such as the east area of

Inner Mongolia and the west part of Liaoning, than that in the BTH region and the Shandong

Peninsula. These stationary source emissions associated possibly with coal combustion because of



the lower ration of OC/EC in cluster 2 than that in cluster 3.

### 3.3 Source apportionment of carbonaceous aerosols in PM$_{2.5}$

The cluster analysis clearly indicated that PM$_{2.5}$ pollution on the Qimu Island was significantly influenced by the high concentrations of PM$_{2.5}$ carried by the air masses from the BTH region and

the Shandong Peninsula during the sampling period. The chemical species in PM$_{2.5}$ from the BTH region had more remarked signals of high temperature burning and mobile source, while that from the Shandong Peninsula had more obvious patterns of low temperature burning and stationary sources. To further confirm the sources of PM$_{2.5}$, source apportionment of carbonaceous aerosols, as the main components of PM$_{2.5}$, was evaluated by $^{14}$C measurement. In order to better achieve our

purpose using a few samples for radiocarbon analysis due to the extensive cost of the analysis, two combined samples were collected from a perfect synoptic process during the sampling period. The selection was reasonable as elaborated below. The synoptic process occurred during 16$^{th}$ and 18$^{th}$, January, 2014. As shown in Fig. 2, the first half air masses of the synoptic process were derived from the south and passed through the Shandong Peninsula (cluster 3) and the bottom half were from the

north and passed over the BTH region (cluster 1). Thus, two samples collected continually from 06:00 to 18:00, 16$^{th}$ January and from 18:00 to 06:00 the next day in the first half of the synoptic process were merged into one sample (M1) for the radiocarbon analysis. Similarly, other two samples collected continually from 17$^{th}$ January 18:00 to 6:00 and from 06:00 to 18:00 in the next day were combined into the other sample (M2). The M1 reflected the signal of air masses coming from the

Shandong peninsula, while M2 showed the pattern of air masses from the BTH region.

Considering that the $^{14}$C measurement will be linked with PMF result, the representative capacity of all samples in the two clusters was examined thoroughly. It is expected that PMF can



better interpret those data close to the average condition of each chemical species since the method

utilizes minimizing error estimates to decompose a matrix of sample data into two matrixes under

strict non-negativity constraints for the factors (Paatero et al., 2014). Similarly, the expectation can

be used to assess their ratio for different purposes. Therefore, OC and EC concentrations, and ratios

of $OC/PM_{2.5}$ and $EC/PM_{2.5}$ of each sample were compared with that in the corresponding cluster by

mean test. Results showed that except for a significant high ratio of $EC/PM_{2.5}$, the OC and EC

concentrations and the $OC/PM_{2.5}$ ratio of M2 were insignificant difference with that in the cluster 1

at a 95% significant level, indicating its perfect representative capability for further carbonaceous

analysis. However, the typical ability of M1 was slightly bad because only ratios of $OC/PM_{2.5}$ and

$EC/PM_{2.5}$ were no statistical difference, OC and EC concentrations were significantly higher than

that in the cluster 3 at the same significant level. Finally, the samples were still considered for

radiocarbon analysis in that they were from the most faultless synoptic process during the whole

sampling period. Such a result was more dramatic than that from two insular samples. In addition,

the insignificant difference of the ratios of $OC/PM_{2.5}$ and $EC/PM_{2.5}$ assured the validity for $PM_{2.5}$

source assessment, which was more important than carbonaceous species in this study.

Table 3 lists the concentrations and contemporary carbon fractions of OC, WSOC, WIOC and

EC of the two combined samples. The fraction of OC was yielded by the average weighted of

concentration of WSOC and WIOC fractions. It can be expressed as $f_c(OC)=[f_c(WSOC) \times C(WSOC)+$

$f_c(WIOC) \times C(WIOC)]/[C(WSOC)+ C(WIOC)]$, where $f_c(OC)$, $f_c(WSOC)$ and $f_c(WIOC)$ are the

contemporary carbon fractions of OC, WSOC and WIOC, and C(WSOC) and C(WIOC) are the

concentrations of WSOC and WIOC, respectively. Generally, WSOC is mainly associated with

biomass burning and secondary formation (Du et al., 2014), while OC directly emitted from the



combustion of fossil fuel is mostly water insoluble (Weber et al., 2007). During the earlier stage of the synoptic process, the concentrations of WSOC and WIOC were 6.38 μg m$^{-3}$ and 6.32 μg m$^{-3}$, respectively. While the concentrations of the two carbonaceous fractions fell to 3.68 μg m$^{-3}$ and 5.30 μg m$^{-3}$, respectively, after the shift of the dominant wind from southerly to northwesterly as shown in

Fig. 2. The fraction of WSOC to OC decreased from 50% to 41% and the WIOC fraction increased from 50% to 59% before and after the shift of the dominant wind. It suggested that the contribution of fossil fuel combustion was more obvious in the BTH region than that in the Shandong Peninsula, and vice versa. The implication could be further confirmed by the radiocarbon analysis of the two samples. The contemporary carbon fractions of WSOC and WIOC decreased from 0.59 to 0.49 and

from 0.60 to 0.43, respectively,which indicated a decrease of the impact of biogenic and biomass burning emission, and an increase contribution of the fossil fuel combustion to the two OC fractions after the shift of the prevailing wind. After the weighted average of the WSOC and WIOC fractions, the $f_c(OC)$ values were 0.59 and 0.46 for the M1 and M2 samples, respectively. Together with $f_c(EC)$, it could be determined that biogenic and biomass burning emission contributed 59% of OC and 52%

of EC concentrations when the air masses were from the Shandong Peninsula. After the change of wind field, the contribution of biogenic and biomass burning emission fell to 46% for OC and 38% for EC, respectively, which suggested that fossil fuel combustion contributed a dominant portion to the carbonaceous aerosols from the BTH region.

The synoptic process showed clearly a shift of the dominant wind from southerly to

northwesterly, namely from the Shandong Peninsula to the BTH region. In the meanwhile, the pattern of biogenic and biomass burning emission was more and more weak, and the signal of fossil fuel combustion was more and more obvious. It was in agreement with that discussed above. For





instance, the emission in the BTH region exhibited more signals of high temperature burning and

vehicle exhausts characterized by the lower ratio of OC/EC (1.41 $\pm$ 0.30), the higher ratio of

$NO_3^-$/nss-$SO_4^{2-}$ (0.96 $\pm$ 0.31) and the relatively lower concentration of nss-$K^+$ compared with that in

the Shandong Peninsula (2.14 $\pm$ 0.50 for OC/EC ratio, 0.64 $\pm$ 0.14 for $NO_3^-$/nss-$SO_4^{2-}$ ratio). The

contribution of the biogenic and biomass burning emission to the carbonaceous aerosol in the

Shandong Peninsula was still significant although there was combustion of fossil fuel (e.g., coal) for

not only industrial activity but also heating in winter. The emission of biogenic and biomass burning

in the Shandong Peninsula was often highlighted in previous studies (Feng et al., 2012;Zong et al.,

2015;Wang et al., 2014).

**3.3 Source apportionment of PM$_{2.5}$**

The EPA PMF 5.0 model and the data sets of 76$\times$22 (76 samples with 22 species) were

conducted to further quantitatively estimate the source contribution of PM$_{2.5}$. After iterative testing

from 5 to 15 factors for the model scenarios, we found the minimum deviation of the source

apportionments of OC and EC between the results from radiocarbon measurement and a PMF model

scenario with an $F_{peak}$ value of 0 and the lowest Q values. The comparison will be elaborated below.

Based on the PMF modeling results, eight main source factors were identified as shown in Fig. 3.

Secondary inorganic aerosols, which were extensively identified in previous studies (Tao et al.,

2014;Zhang et al., 2013), were apportioned into the primary source sectors in the present study.

Because the results are more useful for guiding PM$_{2.5}$ source controls. The contributions of the eight

sources to PM$_{2.5}$ were summarized in Table 4. The total and cluster fractional contributions (%) from

each source were calculated based on the corresponding sample values simulated by the PMF

modeling.





Traffic emission has attracted considerable concern in megacities of China (e.g., Beijing and Shanghai) due to the remarkable growth of vehicles in China. On-road vehicles were estimated as the largest local emission source and contributed 22% of $PM_{2.5}$, including primary and secondary fine particles, but excluding vehicle-induced road dust, in Beijing in 2012 (Zhang et al., 2014c). The first

source factor was characterized by high loadings of $NO_3^-$, $SO_4^{2-}$, $NH_4^+$, OC, EC, Zn and Cu, which encloses vehicle emission profile. Generally, nitrate, sulfate, OC and EC are mainly from engine exhaust emission, and ammonia is from vehicles equipped with three-way catalytic converters. Not only Zn and Cu, but also Pb and Cd are emitted directly bounded particles from exhaust (Zhang et al., 2014c). In addition, the high $NO_3^-/SO_4^{2-}$ ratios of 1.28 calculated by the PMF results showed the

pattern of high temperature burning and vehicle emission. And the source was the largest contributor for $NO_3^-$, which contributed 41% of the chemical species of $PM_{2.5}$ during the sampling period. The contribution was higher than 31% of NOx emitted by traffic sectors in the North China in 2003, which expected an increase of the contribution due to the rapid rise of vehicles in the North China in recent years (Shi et al., 2014b). This factor was the prevalent anthropogenic $PM_{2.5}$ source in the

North China, with an average contribution of 16% during the sampling period. The contribution was lower than that in Beijing (Zhang et al., 2014c), agreeing with the regional contribution characteristic in our study, rather than ones in large city, where a large number of vehicles were running. The second factor consisted of mineral dust elements, such as Mn, Fe and Co, and chemical species from human activities, such as Zn and EC, showing a mixed pattern of natural and anthropogenic

emissions. Vehicle emission is an important source of atmospheric zinc pollution because it can be emitted from direct exhaust, lubricating oil additives, tyre and brake abrasion, wearing and corrosion from anticorrosion galvanized automobile sheet, and reentrainment dust enriched with zinc (Duan





and Tan, 2013). Thus, the source factor was treated as traffic dust under the relative high contribution

of vehicle emission to PM$_{2.5}$ concentration.

The third source factor was ship emissions, typically characterized by high proportion of Ni and

V, and high ratio of V/Ni. High loading of the two metals is typically associated with the emission

from residual oil, probably derived from shipping activities and some industrial processes (Pey et al.,

2013). In addition, a V/Ni ratio of more than 0.7 is always considered as a sign of PM$_{2.5}$ influenced

by shipping emissions (Zhang et al., 2014a). The average ratios of V/Ni from the measured data were

0.93 ± 0.46, indicating an obvious contribution of shipping emission. The average ratio of V/Ni

calculated from the PMF source profile was 1.02, which was the second high value among that

derived from the eight sources. While the highest value of 1.29 was for the mineral dust source,

which agrees with a high ratio of 3.06 for soil background concentrations of the two metals in

mainland China (Pan et al., 2013). The fourth factor showed high loading of Cu, Zn, As, Cd and Pb,

which was treated as mainly contributed by industrial processes. The emission from the iron and

steel industry are possibly important among those industrial processes based on two proofs. One is

the sintering process in iron and steel industries emits lots of Pb, Hg, Zn and other heavy metal

pollutants, and other processes such as ironmaking and steelmaking also emit fugitive dust

containing high concentration of heavy metals (Duan and Tan, 2013). The other is the huge

production of steel in the North China. The national statistical data shows that China produced about

half world production of crude steel in 2014, and the productions in the BTH and Shandong were

25.3% and 7.8% of the total amount in China, respectively, which is available at the website

http://www.stats.gov.cn/tjsj/ndsj/. Thus, iron and steel industries are the main atmospheric sources of

the metal elements. In addition, the contribution of the source to sulfate was 12%, which is similar



with the value of 15% contribution of industrial processes to the amount of sulfur dioxide as reported

previously (Zhao et al., 2012).

The fifth source factor was biomass burning, characterized by high concentrations of $K^+$, OC,

EC and $NH_4^+$, which have been used extensively as tracers of biomass-burning aerosols (Zhou et al.,

2015;Tao et al., 2014). The contribution of the source was significantly higher in cluster 3 than that

in cluster 1 and cluster 2 as listed in Table 4. It agreed with more biomass burning emission in

Shandong Peninsula characterized by rich $K^+$ and the high OC/EC ratio. Besides, the average ratio of

OC to EC from the source was also the highest (1.84) among the eight identified sources (0.23-1.84)

calculated by the PMF modeling. The sixth source factor was mineral dust, characterized typically by

crustal elements, such as $Ca^{2+}$, Ti and Fe, which are always used as the markers of soil dust (Zhang et

al., 2013). The contribution of the source was obviously higher in cluster 2 than that in cluster 1 and

3, matching with the high wind speed in cluster 2. The average ratio of OC to EC (1.53) from the

source was obviously higher than that (0.23) from vehicle dust, possibly suggesting that the source

contributed more OC derived from biogenic dust, such as plant debris.

The seventh source factor was characterized by high loading of $Cl^-$, $Na^+$, OC, EC, $SO_4^{2-}$ Ni and

As. Coal combustion is often showed by elevated $Cl^-$ linked with high $Na^+$, OC and EC (Zhang et al.,

2013). The source was the largest contributor of sulfate in the present study, matching with the

inventory results in the North China (Zhao et al., 2012). Also, the source was the largest contributor

of $PM_{2.5}$ as listed in Table 4, which was coherent with the fact that coal combustion is the

predominant source of fine particle aerosols over China (Pui et al., 2014). As and Ni showing high

loading in the factor was also used as the markers for coal-fired power plant emissions (Tan et al.,

2016). The last source factor was sea salt, characterized by high loading of $Na^+$, $Mg^{2+}$ and $Cl^-$, which



are related to the primary sea-salt aerosols produced by the mechanical disruption of the ocean surface. Similar to the second source (mineral dust), high wind speed in cluster 2 made the high contribution of the source in cluster 2 than that in cluster 1 and 3. In addition, the higher contribution fractions for $Mg^{2+}$ than for $Cl^-$ of the source were in agreement with that discussed above. The

concentration ratios of $Cl^-/Na^+$ and $Mg^{2+}/Na^+$ calculated from the PMF source profile were 1.79 and 0.11, respectively, similar to the corresponding ratios of the species (1.80 and 0.12, respectively) in average seawater (Ni et al., 2013). The sea salt source contributed 2.53%, 15.2% and 1.93% of OC concentrations in cluster from 1 to 3, respectively, but nothing for EC in all the clusters. It indicated the source consisted of sea-spray organic aerosol, which was produced by marine biogenic activities

(Wilson et al., 2015).

To obtain more reliable source apportionment, the source contribution to OC and EC derived from those PMF scenarios for the two specified samples (M1 and M2) was compared with the source apportionment indicated by the [14]C assessment. The model results with the minimum deviation were treated as providing a more reliable solution for the source assessment. For the final choice, the

contributions of coal combustion, vehicle emission, industrial process and ship emissions derived from the PMF modeling to OC and EC were ranked as the fossil fuel combustion for the comparison. Sea salt as marine biogenic source of OC was merged with biomass burning as contemporary carbon fractions. While mineral dust and vehicle dust were not taken into consideration for the classification because they originated from a hybrid sources of fossil and contemporary carbon emissions. Fig. 4

shows the comparison of the PMF results and the [14]C measurement.

In M1, the biogenic and biomass burning (B&B) emission identified by PMF modeling contributed 52% and 49% of OC and EC concentrations, which were 7 and 3 percent, respectively,





below the respective fraction indicated by the radiocarbon measurement. Accordingly, the contributions of fossil fuel combustion to OC and EC from the PMF result were the same fraction of 44%, which were 3 percent over and 4 percent below the corresponding values in the $^{14}$C result. Similarly, in M2, the B&B emission contributed 41% and 33% to OC and EC in the PMF result,

showing 4 and 5 percent below that in the $^{14}$C result. The contribution of fossil fuel combustion to OC and EC in the PMF result was 52% and 65%, which were the same percent (3%) below and over the corresponding values in the $^{14}$C result. In general, the source contributions merged from the PMF results were lower than that from the radiocarbon measurements. The underestimation may be attributed to the absence of considering the contribution of mineral dust and vehicle dust because of

their hybrid sources of B&B and fossil fuel combustion. The largest difference was 7 percent, indicating a minor contribution of the two sources to carbonaceous species in $PM_{2.5}$. The substantial difference was the two overestimations with the same range (3%), one was the contribution of fossil fuel combustion to OC in the M1 and the other was the contribution of fossil fuel combustion to EC in the M2. The overestimation was attributed to classify irrelevantly B&B emission into fossil fuel

combustion. In conclusion, the minor irrelevant classification suggested that the PMF results provided a reasonable source apportionment of regional $PM_{2.5}$ in the North China in winter.

As listed in Table 4, among the eight sources identified by the PMF modeling, coal combustion, biomass burning and vehicle emissions were the largest contributors of $PM_{2.5}$, which accounted for 29.6%, 19.3% and 15.8%, respectively, during the sampling period. They were followed, in

decreasing order, by mineral dust (12.8%), ship emissions (8.95%), sea salt (6.58%), traffic dust (4.24%) and industrial process (2.64%). Generally, the source apportionment profile of $PM_{2.5}$ in cluster 1 was similar to that during the whole sampling period because the regional scale pollution



meantime exhibited a pattern of atmospheric outflow of $PM_{2.5}$ from the BTH region in winter (Feng et al., 2007;Feng et al., 2012). A slight increase of the contribution of vehicle emission in cluster 1 corresponded the great concern on the source in megacities of China (Zhang et al., 2014c). However, the source signals were obviously different from that in cluster 2 and 3. The strong northwesterly wind in the cluster 2 provided more large spatial scale signals of source apportionment of $PM_{2.5}$, which indicated that coal combustion (37.7%) and mineral dust (26.8%) were the largest contributors in north areas of China in winter. The large scale $PM_{2.5}$ pattern linked with coal combustion agreed with the dominant position of coal consumption in Chinese energy structure as reported that the coal consumption accounted for 66% of primary energy in China in 2014 by the national bureau of statistics of China (available at http://www.stats.gov.cn/tjsj/ndsj/). Except for industrial consumptions, an additional use of coal is for residential heating in north areas of China during winter, which was extensively consumed. Although the household use of coal accounts for a small portion of total coal consumption in China, the release is still a major source in winter (Cao et al., 2012) since those household stoves always run under a condition with lack or backward environmental protection equipment. The traffic emission concerned extensively in large cities contributed a minor part (3.57%) of $PM_{2.5}$ concentration on the large spatial scale because motor exhaust concentrates mainly in urban areas. In addition, Biomass burning emission dominated the $PM_{2.5}$ pollution when air masses came from the Shandong Peninsula. The abundant emission from biomass burning was mainly attributed to the residential heating in the cold season.

**4 Implications for PM alleviation**

According to the source apportionment, coal combustion was the largest contributor of $PM_{2.5}$ in the North China during winter, and the source posed a larger spatial pattern of $PM_{2.5}$ pollution in



north areas of China compared with the North China. Therefore, PM emissions generated by coal

combustion should be firstly abated. Also, the source has been identified as the most leading

emission sector with a goal of controlling the annual $PM_{2.5}$ concentration in the air pollution control

program. The contributions of traffic emission and biomass burning to $PM_{2.5}$ concentrations also had

an obviously spatial pattern in the North China during winter. Vehicle emission contributed

significantly to $PM_{2.5}$ in the BTH region, so the source should be prior considered to control as the

second major emission sector listed in the air pollution control program.

It should be paid close attention to the biomass burning emission due to its dominant

contribution to the $PM_{2.5}$ burden in the Shandong Peninsula and because the emission has been only

considered slightly in the control program. Indeed, the first national pollution source survey showed

that Shandong province is the largest producer of crop stalks with a production of 132 million tons in

China in 2007 (Compilation-Committee-of-the-first-China-pollution-source-census, 2011). Of which,

about 20 million tons were produced in the Shandong Peninsula (including the cities of Weifang,

Yantai, Weihai, Qingdao and Rizhao). Approximate 40% production was used as household fuels for

cooking and heating in countryside of the peninsula. The fraction was significantly higher than that

in western areas of the Shandong province, such as Zibo (9%) and Jinan (8%), and the fraction of

open burning of crop residues in the peninsula (3%). Besides, the fraction of biomass open burning

in the peninsula was also higher than the average fraction (1.5%) in Shandong province in 2007.

Generally, the emissions from agricultural field burning are mainly concentrated in harvest season

and contribute significantly to regional haze and smog events in the region, which have attracted

special concern(Feng et al., 2012;Zong et al., 2015;Wang et al., 2014). But the open burning

emission has been considered as a minor source sector in the control program. Similarly, as an larger





emitter, household emission of agricultural waste is continuous or semi-continuous, and also can

induce $PM_{2.5}$ pollution on a regional scale, which has been despised or ignored (Zhang and Cao,

2015). Although the government has enacted a series of regulations to prohibit field burning since the

1990s and strengthens the force of the supervision recently, the open burning is not fully controlled

in China. The most basic reason is that there is no reasonable alternative to utilize or handle the huge

production of agricultural waste per year. Under such a case, a part of agricultural wastes are

collected and stored as fuel for household cooking and heating, and others are rapidly removed by

open burning in field for the next planting during harvest season. Although farmers know that such

use and disposal of agricultural residues are harm to the environment. They still tend to use

agricultural waste as household fuel and remove them via burning in field mainly due to the low cost

of the methods. A more permanent solution is to find the high economic value of agricultural wastes

via development of renewable techniques. Indeed, agricultural waste can be utilized to produce

manifold renewable energy, such as biogas, feedstuffs, biochar, bioethanol, bio-succinic acid, and so

on. China has provided relevant energy regulations, legislations, and policy initiatives for rural

renewable energy (Li et al., 2015). The government also encourages and sustains the development of

the renewable energy industry to increase the demand of raw feedstocks. Through these efforts,

China has achieved some success in renewable development in rural areas, but the efforts are not

effective solution to the problem of surplus crop waste because the cost and the benefit cannot be

offset. For instance, Zhangziying town located in the east area of Daxing district of Beijing has

developed household biogas and straw gas since 1980s, but the renewable energy only occupied

about 10% of household energy consumption, which was much lower than the fraction of coal (30%)

in 2011(Li et al., 2015). Before the achievement of high economic value, except for the ban on crop





straw burning, the government should compensate farmers to collect crop residues as feedstocks of

renewable energy, rather than burning in field or household (Shi et al., 2014a). The revenue from the

subsidy and the sale of crop residues can support famer's economic burdens for the use of clean

energy, such as electricity, liquefied petroleum gas, biogas and so on, for household consumption

(Kung and Zhang, 2015). The efforts will not only significantly improve air quality, but also make

famers to learn the convenient of clean energy and wake from agricultural residue burning.

**5 Conclusions**

Source apportionment of wintertime $PM_{2.5}$ at a background site in the North China in 2014 was

conducted by statistical analysis of the chemical species grouped according to the trajectory clusters,

radiocarbon measurement of the carbonaceous species and the PMF modeling confirmed by the $^{14}C$

analysis. During the sampling period, the mean concentration of $PM_{2.5}$ was $77.6 \pm 59.3$ μg m$^{-3}$, and

sulfate concentrations were the highest with a mean of $14.2 \pm 18.0$ μg m$^{-3}$, followed by nitrate (11.9

$\pm 16.4$ μg m$^{-3}$), OC ($6.85 \pm 4.81$ μg m$^{-3}$), EC ($4.90 \pm 4.11$ μg m$^{-3}$) and ammonium ($3.11 \pm 2.14$ μg

m$^{-3}$). The fractions of sulfate, nitrate and ammonium to $PM_{2.5}$ were obviously higher than that in

metropolis (e.g. Beijing and Tianjin) within the North China, while fractions of carbonaceous species

was markedly lower, which showed regional pollution signals. More than half of air masses were

from the BTH region, followed by the air masses from the Mongolia (35%) and the Shandong

Peninsula (11%) during the sampling period.

The concentrations of $PM_{2.5}$ and its most species carried by the air masses from the BTH region

and the Shandong Peninsula were comparable ($p > 0.05$), and they were statistically higher than that

carried by the air masses from the Mongolia ($p < 0.01$). The patterns were attributed to the spatial

distributions of their emissions and the high wind speed when air masses were from the Mongolia.



The PM$_{2.5}$ had an obvious signal of biomass burning emission, characterized by high OC/EC ratio, low NO$_3^-$/nss-SO$_4^{2-}$ ratio and high nss-K$^+$ concentration, when air masses came from the Shandong Peninsula. While the PM$_{2.5}$ carried from the BTH region showed vehicle emission pattern, characterized by low OC/EC ratio, high NO$_3^-$/nss-SO$_4^{2-}$ ratio and low nss-K$^+$ concentration. The

finding was confirmed by the radiocarbon measurement of OC and EC in two merged samples (One was collected when air masses were from the Shandong Peninsula and the other was air masses from the BTH region) selected from a successive synoptic process. The $^{14}$C measurement indicated that biogenic and biomass burning emission contributed 59% and 52% of OC and EC concentrations when air masses were from the Shandong Peninsula, and the contributions fell to 46% and 38% for

OC and EC, respectively, when the prevailing wind changed and came from the BTH region.

Based on the PMF modeling results, eight main source factors were identified. The source contribution to OC and EC derived from PMF for the two specified samples was compared with that indicated by the $^{14}$C assessment. Two minor overestimations with the same range (3%) showed the excellent model capacities, suggesting that the PMF results provided a reasonable source

apportionment of regional PM$_{2.5}$ in the North China in winter. The PMF results indicated that coal combustion, biomass burning and vehicle emissions were the largest contributors of PM$_{2.5}$, which account for 29.6%, 19.3% and 15.8%, respectively, during the sampling period. Compared with the overall source apportionment, the contribution of vehicle emission increased slightly when air masses came from the BTH region, the fraction of mineral dust and coal combustion rose obviously

when air masses with high speed were from the Mongolia, and biomass burning became the dominant contributor when air masses were from the Shandong Peninsula.

As the largest contributor of PM$_{2.5}$ in the North China in winter, coal combustion has been

considered as the most leading emission sector to be controlled for improving the air quality. Vehicle

emission contributed significantly the $PM_{2.5}$ levels in the BTH region, which has also been

considered to control as the second major emission sector listed in the air pollution control program.

Biomass burning emission was highlighted in the present study because of its dominant contribution

to the $PM_{2.5}$ burden in Shandong Peninsula, which has been despised or ignored in the control

program. To improve air quality, some suggests were provided to wake famers from agricultural

residue burning in field and household.

**Acknowledgment**

This work was supported by the Promotive Research Foundation for Excellent Young and

Middle-aged Scientists of Shandong Province (No. BS2012HZ028), the CAS Strategic Priority

Research Program (Nos. XDA11020402, XDB05030303 and XDB05040503), and the Natural

Scientific Foundation of China (Nos. 41471413 and 41373131). The authors gratefully acknowledge

the National Oceanic and Atmospheric Administration's Air Resources Laboratory for providing the

HYSPLIT transport model and the READY website (http://www.arl.noaa.gov/ready.html).

**Notes**

The authors declare no competing financial interest.

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

Table 1. Statistics of PM$_{2.5}$ chemical components on the Qimu Island during the sampling period

| Species | Mean ± std. (µg m$^{-3}$) | Range (µg m$^{-3}$) | Species | Mean ± std (ng m$^{-3}$) | Range (ng m$^{-3}$) |
|---|---|---|---|---|---|
| PM$_{2.5}$ | 77.6 ±59.3 | 12.7 – 305 | Ti | 7.72 ±7.34 | 0.01 – 30.7 |
| EC | 4.90 ±4.11 | 0.80 – 19.6 | V | 3.90 ±2.47 | 0.45 – 12.5 |
| OC | 6.85 ±4.81 | 0.81 – 21.3 | Mn | 29.3 ±28.0 | 1.38 – 108 |
| Cl$^-$ | 2.06 ±1.78 | 0.10 – 8.90 | Fe | 408 ±285 | 7.12 – 1588 |
| NO$_3^-$ | 11.9 ±16.4 | 0.27 – 87.1 | Co | 0.24 ±0.18 | 0.01 – 0.73 |
| SO$_4^{2-}$ | 14.2 ±18.0 | 1.37 – 96.2 | Ni | 4.28 ±2.30 | 1.68 – 13.8 |
| Na$^+$ | 0.43 ±0.25 | 0.05 – 1.58 | Cu | 9.08 ±11.4 | 0.03 – 77.7 |
| NH$_4^+$ | 3.11 ±2.14 | 0.61 – 10.1 | Zn | 107 ±142 | 5.56 – 987 |
| K$^+$ | 0.96 ±0.84 | 0.07 – 3.95 | As | 6.61 ±7.86 | 0.67 – 43.4 |
| Mg$^{2+}$ | 0.03 ±0.03 | 0.01 – 0.17 | Cd | 1.82 ±4.06 | 0.04 – 25.9 |
| Ca$^{2+}$ | 0.38 ±0.22 | 0.07 – 1.32 | Pb | 88.4 ±85.7 | 3.02 – 412 |

Table 2. Statistics of PM$_{2.5}$ chemical species in different clusters and significant level by mean test

| Species (unit) | Mean ± standard deviation (range) | | | Significant level | | |
|---|---|---|---|---|---|---|
| | Cluster1(n=42) | Cluster2(n=25) | Cluster3(n=9) | 1&2 | 1&3 | 2&3 |
| PM$_{2.5}$ (µg m$^{-3}$) | 93.0 ±66.1 (24.5–305) | 41.6 ±26.7 (12.7–143) | 106 ±42.3 (50.3–193) | 0.00 | 0.59 | 0.00 |
| EC (µg m$^{-3}$) | 6.53 ±4.66 (1.39–19.6) | 2.50 ±1.84 (0.80–8.85) | 3.94 ±1.49 (2.53–7.66) | 0.00 | 0.11 | 0.05 |
| OC (µg m$^{-3}$) | 8.58 ±5.23 (1.45–21.3) | 3.51 ±2.35 (0.81–11.4) | 8.04 ±2.32 (5.25–13.5) | 0.00 | 0.76 | 0.00 |
| Cl$^-$ (µg m$^{-3}$) | 2.37 ±2.11 (0.10–8.90) | 1.22 ±0.65 (0.20–2.85) | 2.94 ±1.35 (1.42–5.53) | 0.01 | 0.45 | 0.00 |
| NO$_3^-$ (µg m$^{-3}$) | 17.6 ±19.6 (1.75–87.0) | 2.75 ±4.25 (0.27–20.1) | 10.6 ±6.09 (4.41–20.3) | 0.00 | 0.30 | 0.00 |





| | | | | | | |
|---|---|---|---|---|---|---|
| $SO_4^{2-}$ | 19.4 ±21.8 | 4.55 ±4.06 | 16.4 ±8.74 | 0.00 | 0.69 | 0.00 |
| (µg m$^{-3}$) | (2.09–96.2) | (1.37–19.5) | (5.34–35.6) | | | |
| $Na^+$ | 0.38 ±0.24 | 0.55 ±0.26 | 0.31 ±0.06 | 0.01 | 0.41 | 0.01 |
| (µg m$^{-3}$) | (0.05–1.58) | (0.18–1.08) | (0.22–0.40) | | | |
| $NH_4^+$ | 3.97 ±2.29 | 1.53 ±0.98 | 3.52 ±0.96 | 0.00 | 0.57 | 0.00 |
| (µg m$^{-3}$) | (1.28–10.1) | (0.61–4.70) | (1.93–4.90) | | | |
| $K^+$ | 1.11 ±0.74 | 0.35 ±0.36 | 2.01 ±0.93 | 0.00 | 0.00 | 0.00 |
| (µg m$^{-3}$) | (0.28–3.10) | (0.07–1.69) | (0.78–3.95) | | | |
| $Mg^{2+}$ | 0.03 ±0.03 | 0.03 ±0.02 | 0.02 ±0.01 | 0.66 | 0.41 | 0.13 |
| (µg m$^{-3}$) | (0.01–0.17) | (0.01–0.11) | (0.01–0.04) | | | |
| $Ca^{2+}$ | 0.37 ±0.22 | 0.37 ±0.18 | 0.44 ±0.29 | 1.00 | 0.46 | 0.46 |
| (µg m$^{-3}$) | (0.11–1.32) | (0.07–0.74) | (0.09–0.97) | | | |
| Ti | 6.96 ±5.98 | 10.9 ±9.10 | 2.51 ±0.85 | 0.04 | 0.03 | 0.01 |
| (ng m$^{-3}$) | (0.35–25.9) | (0.01–30.7) | (1.16–3.58) | | | |
| V | 4.68 ±2.29 | 2.83 ±2.55 | 3.24 ±1.50 | 0.00 | 0.08 | 0.66 |
| (ng m$^{-3}$) | (0.76–11.3) | (0.45–12.4) | (2.05–7.12) | | | |
| Mn | 33.8 ±31.3 | 17.6 ±19.3 | 40.9 ±20.3 | 0.02 | 0.53 | 0.01 |
| (ng m$^{-3}$) | (1.97–108) | (1.38–95.4) | (9.14–69.8) | | | |
| Fe | 404 ±308 | 375 ±263 | 521 ±188 | 0.70 | 0.29 | 0.15 |
| (ng m$^{-3}$) | (7.12–1588) | (9.13–826) | (244–960) | | | |
| Co | 0.26 ±0.20 | 0.17 ±0.14 | 0.36 ±0.13 | 0.08 | 0.14 | 0.00 |
| (ng m$^{-3}$) | (0.01–0.73) | (0.01–0.48) | (0.10–0.59) | | | |
| Ni | 4.85 ±2.56 | 3.51 ±1.85 | 3.80 ±1.02 | 0.03 | 0.24 | 0.67 |
| (ng m$^{-3}$) | (1.68–13.8) | (1.68–6.79) | (2.45–5.84) | | | |
| Cu | 11.6 ±13.6 | 3.06 ±2.93 | 13.9 ±7.05 | 0.00 | 0.64 | 0.00 |
| (ng m$^{-3}$) | (0.72–77.7) | (0.03–8.99) | (3.90–26.4) | | | |
| Zn | 146 ±176 | 46.4 ±50.1 | 90.4 ±47.4 | 0.01 | 0.36 | 0.03 |
| (ng m$^{-3}$) | (9.92–987) | (5.56–208) | (24.2–201) | | | |
| As | 9.03 ±9.52 | 3.00 ±2.82 | 5.35 ±3.35 | 0.00 | 0.27 | 0.06 |
| (ng m$^{-3}$) | (1.11–43.4) | (0.67–14.0) | (2.25–13.6) | | | |
| Cd | 2.70 ±5.26 | 0.45 ±0.41 | 1.54 ±0.65 | 0.04 | 0.52 | 0.00 |
| (ng m$^{-3}$) | (0.11–25.9) | (0.04–1.29) | (0.49–2.66) | | | |
| Pb | 110 ±95.3 | 36.9 ±44.8 | 128 ±53.2 | 0.00 | 0.59 | 0.00 |
| (ng m$^{-3}$) | (5.30–412) | (3.02–176) | (45.4–215) | | | |

Table 3. Concentration and contemporary carbon fraction of carbonaceous species in M1 and M2

| | M1 | M2 | | M1 | M2 |
|---|---|---|---|---|---|
| PM$_{2.5}$(µg m$^{-3}$) | 159 | 91.8 | | | |
| OC(µg m$^{-3}$) | 12.7 | 8.98 | $f_c$(OC) | 0.59 | 0.46 |
| WSOC(µg m$^{-3}$) | 6.38 | 3.68 | $f_c$(WSOC) | 0.59 | 0.49 |
| WIOC(µg m$^{-3}$) | 6.32 | 5.30 | $f_c$(WIOC) | 0.60 | 0.43 |
| EC(µg m$^{-3}$) | 8.58 | 5.81 | $f_c$(EC) | 0.52 | 0.38 |



Table 4. Averages of fractional contributions (%) from eight sources identified by PMF model

|          | Vehicle emission | Traffic dust | Ship emission | Industrial process | Biomass burning | Mineral dust | Coal combustion | Sea salt |
|----------|------------------|--------------|---------------|--------------------|-----------------|--------------|-----------------|----------|
| All      | 15.8             | 4.24         | 8.95          | 2.64               | 19.3            | 12.8         | 29.6            | 6.58     |
| Cluster1 | 23.6             | 4.89         | 8.79          | 3.64               | 19.6            | 6.32         | 29.2            | 3.96     |
| Cluster2 | 3.57             | 3.60         | 9.34          | 1.19               | 4.88            | 26.8         | 37.7            | 13.0     |
| Cluster3 | 12.5             | 3.08         | 8.67          | 1.96               | 52.7            | 6.46         | 12.4            | 2.32     |

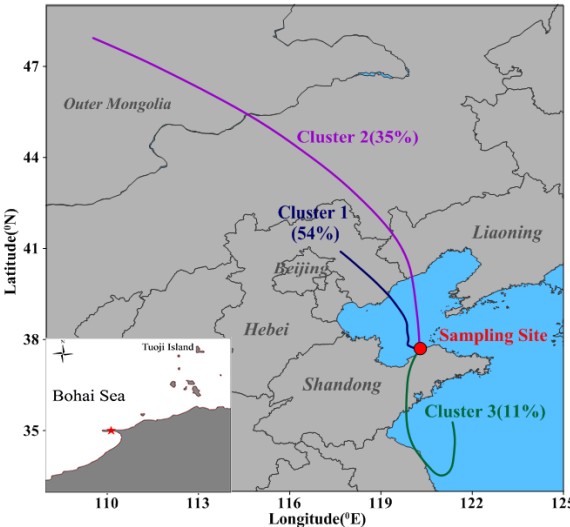

5                Figure 1. The sampling site and 48-h back trajectory clusters during the sampling period



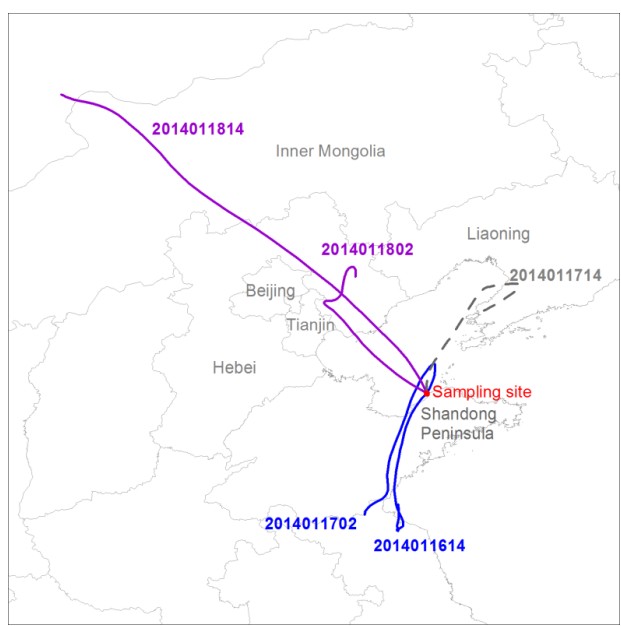

Figure 2. 48-h back trajectories with 12 h intervals during 16-18 January, 2014. (The digit in the figure is date

and time with the format of YYYYMMDDHH, the time is local time)

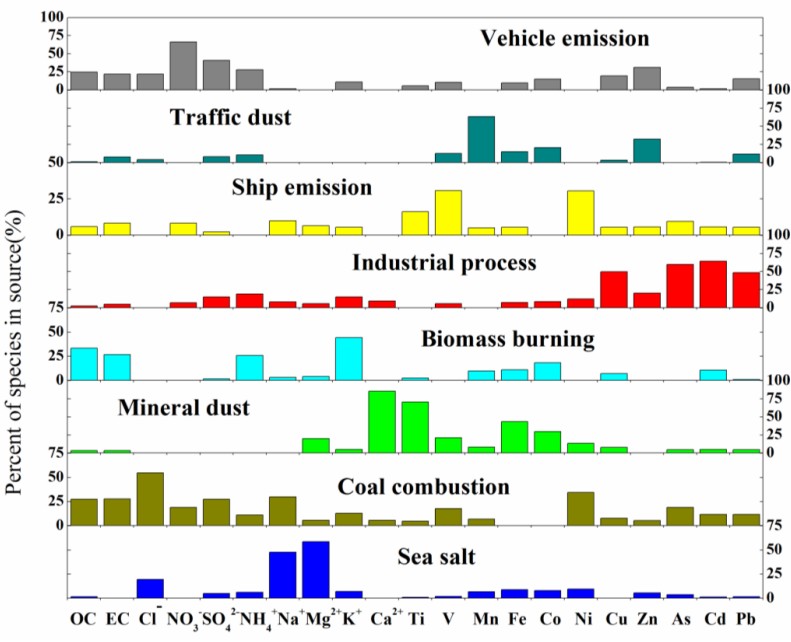

Figure 3. The contribution profiles of eight sources identified by PMF model




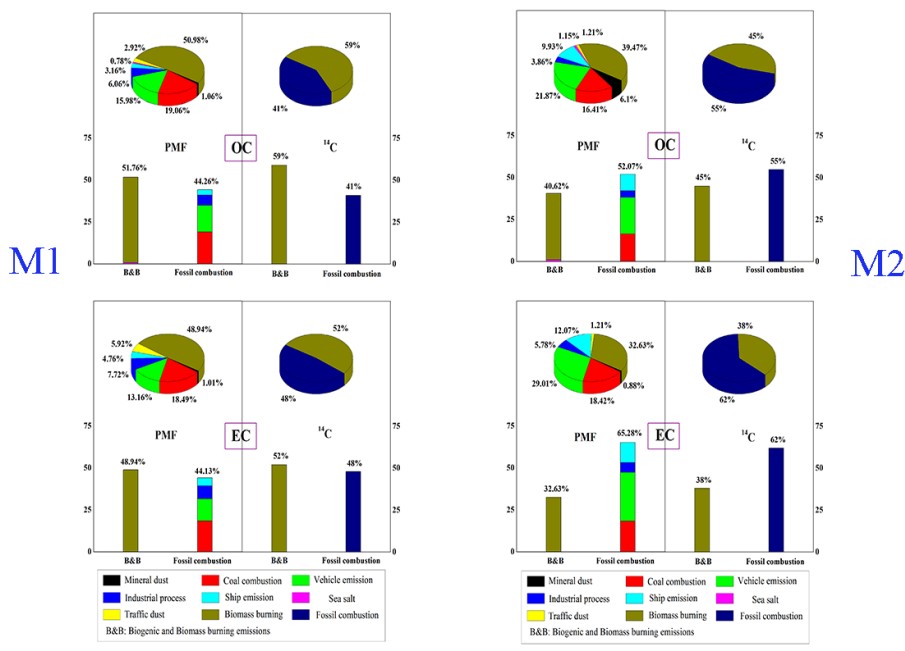

Figure 4. Comparison of source apportionment of OC and EC in two specified samples (M1 and M2) from PMF

and [14]C measurement. B&B is the source of biogenic and biomass burning.

