# Peer review of "Source apportionment of $PM_{2.5}$ at a regional background site in North China using PMF linked with radiocarbon analysis: Insight into the contribution of biomass burning"

_Atmospheric Chemistry and Physics, 2016_

## Referee Comment (RC1) · Anonymous Referee #1 · 1 Apr 2016

One concern is the use of only one station at a background location, to interprets sources of PM2.5 over a very large area without any discussion of the possibility of local sources affecting the measurements. In one part of the manuscript the authors hypothesize that the elevated PM2.5 at the sampling site are due to the proximity of sources (p.10, line 3-5). But later based on their cluster analysis they say only 11% of the air masses were from the Shandog Peninsula (p. 11 line 18).

Further, measuring PM2.5 concentrations based on filter mass is not accurate due to variability within the filter area. Each punch might weigh differently due to the inhomogeneity of the filter itself. There are no references about this technique being used anywhere else.

Radiocarbon was only measured on two samples, each representing an air mass coming from a different region. One sample cannot capture any variability in sources and is not a true representative of regional sources. It is also not clear what the uncertainties in the results are and how they could affect the source apportionment. When using radiocarbon, it is particularly important to accurately represent the uncertainty in aerosol samples, as they are very small and usually have high degree of uncertainty, which needs to be considered when interpreting the results.

With respect to the back trajectory analysis and clustering, the manuscript did not provide enough detail to explain how this was performed and the model for the clustering. Also the authors did not include the reference, giving credit to the original HYSPLIT model development.

In general, the manuscript was very hard to read and understand due to the language (and multiple typos and errors), which needs significant improvement. The title of the manuscript is also misleading, since the biomass-burning signal is not the main focus of the manuscript. The lack of appropriate citations makes it hard to track the accuracy and reliability of the statements and hypothesis throughout the manuscript and makes the interpretation of the results difficult. The details provided in the methods section are not enough to fully understand the analysis or to allow traceability or reproducibility of the results. The results and discussion section was hard to follow too due to the lack of structure within the text and paragraphs summarizing the result from the different experiments and drawing conclusions together.

A list of specific comments and questions are provided below:
- Missing citations throughout the introduction
- Spaces missing p. 3 line 20; p. 4 line 10, p. 13 line 20, p. 27, line 22
- The introduction (and the rest of the manuscript) does not explain how 14C provides information about fossil and modern sources.
- Page 5 line 9: Longkou district and Tianjin are not shown on map. Is Longkou the closest urban region and can it be considered a local source of urban emissions?
- Page 5, line 18: Why do the filters undergo a 24h equilibrium period at 25C? Wouldn't that potentially introduce contamination with the filters absorbing volatile organic compounds during that period?
- OC and EC page 6 lines 6-7: The DRI currently uses an updated protocol – IMPROVE_A (*Chow et al., 2007*). Which one was used for this manuscript? http://www.dri.edu/eaf-projects
- Page 7, line 14: Radiocarbon methods are not cited properly (p. 7, line 14 Liu et al 2014 is not explaining the process) – correct citation is Zhang et al. 2015 (Env. Sci. & Tech) Radiocarbon-Based Source Apportionment of Carbonaceous Aerosols at a Regional Background Site on Hainan Island, South China
- Page 7 Line 16: which TOR protocol? (also acronym was not defined anywhere)
- Graphitization at CAS and AMS analysis – can you provide citation, sizes, black corrections?
- Page 8: HYSPLIT and EPA PMF v5.0 citations
- Page 9, line 15 The text does not follow the order reported in Table 1 (notes on tables and figures will be provided at the end)
- Page 9, line 17 What is the uncertainty of the max PM2.5 value? Also, there is no need to report significant digits, as the range is so large
- Page 10, line 2: It is not clear where Tuiji Island is and how it is relevant in the comparison.
- Page 10, line 8-9. Write chemical formula for sulfate, nitrate and ammonium and use them throughout the text. The lack of consistency with the names of the chemical elements and radiocarbon vs 14C makes the text harder to read.
- Page 10, line 20-1. There are major issues with the chemical elements here – The ones written out are not the correct elements!
- Page 11, line 22: Mean test?
- Break next section in paragraphs. It is very hard to read.

- Page 12, second paragraph – write out chemical elements
- Page 12, line 21. Cite the sources for sea salt composition
- Page 13: It is really hard to follow and it might be helpful to break it up in a couple paragraphs
- Page 14: It is not clear how the Enrichment Factor (EF) method works, if/how it has been applied in other studies and how reliable it is. Original citations are missing. More description is necessary. It will also be helpful to plot the EF calculated for all elements.
- Page 14: Using OC/EC ratio and NO3/nss-SO4 a traces :
    - Lack of citations OC/EC p. 14 line 21
    - Mean test (p. 15 line 2)?
    - When calculating OC/EC ratios what is the uncertainty in the OC and EC and the ratio.
    - Lack of citations SO2 and NOx p. 15 line 7
- Page 15, line18: Where is Chengde and how is it relevant?
- Source app of carbonaceous PM2.5 (pages 16-19)
    - The back trajectory analysis is not clear in the description of how the samples were combined. The figure does not help either.
    - Page 16 line 11 – 19 unclear and belong in the methods
    - Page 18 line 2: What is the WSOC and WIOC concentration uncertainty? The reported numbers have two significant digints, but are the methods accurate enough to report concentrations of this precision?
    - M1 and M2 are significantly different, but could this be due to high level of uncertainty in the measurement? Can it possibly be affected by local sources?
- Page 19, line 15: Fpeak and Q are not defined
- Page 20, line 6: Citations missing
- Page 24, line 3: PMF and 14C results are compared within 3-4% precision, but it is likely the uncertainty is higher.
- Implications for PM alleviation – since it was mostly focused on biomass burning it seemed to not fully capture the scope of the work.
- It would have been very helpful if there were a section in the manuscript where all the results were drawn together so the reader can logically conclude how the different techniques complemented each other and what the final result of the different analytical techniques was.
- There are a lot of typos in the references.

Tables and Figures:

Table 1: It is not clear why the table is split in two and why the units are different. It would be much easier to interpret the content if everything is aligned into 3 columns and the units are consistent.
Table 2: Make into one page
Table 3: Uncertainty needs to be reported for each measurement

Table 4: Rows 1, 3, and 4 do not add up to a 100% when horizontally summed. The difference is minimal, but what is the reason for it? Also the significant digits are probably not necessary, based on the accuracy of the analysis.

Figure 1. Some locations that were mentioned in the text are not included in the figure.
Figure 2. This is very difficult to follow, a more detailed description of what the figure is showing will be helpful
Figure 4. The figure caption should explain why the B&B and Fossil emissions from the PMF do not add to a hundred in the bars. It is described in the text, but a note in the figure will make it much easier to understand.

---

## Short Comment (SC1) · 1 Apr 2016

Comments to a paper Biomass burning contribution to regional PM2.5 during winter in the North China In present study, a more reliable source apportionment of PM2.5 on the regional scale in the North China in winter is provided using PMF simulation and the radiocarbon measurements. It is consistent work performed on a good level. It looks like the title of the paper does not relate the content of the paper well. It announces Biomass burning contribution to regional PM2.5 during winter in general in the North China while the work was done only on one site. If one site represents the whole North China?

---

## Author Comment (AC1) · 25 Apr 2016

1. It looks like the title of the paper does not relate the content of the paper well.

Thanks for the comment. As mentioned by the reviewer, the sources apportioned by the paper are not only biomass burning but also coal combustion, vehicle emission and so on. We use the title "Biomass burning contribution to regional $PM_{2.5}$ during winter in the North China" in order to highlight the important of biomass burning, which is often underestimated or ignored in some areas in China (Gao et al., 2016). However, we agree with the opinion of the reviewer, and change the title as "Radiocarbon and PMF based the source apportionment of regional $PM_{2.5}$ in North China: insight into the contribution of biomass burning ".

2. If one site represents the whole North China?

Thanks for the comment. Background site is usually adopted to determine the regional-scale concentration burden and the source apportionment of aerosols (Moon et al., 2008; Sheesley et al., 2012; Waked et al., 2014; Yao et al., 2016; Yin et al., 2010; Yttri et al., 2011). The sampling point in our study is a regional background site in North China, where no obvious emission source nearby. Besides, the back trajectory clusters indicated that more than half of the air masses (54%) during the sampling period were from the Beijing-Tianjin-Hebei region, followed by the air masses from the Mongolia (35%), and Shandong Peninsula (11%), suggesting the area air mass passed through in this paper included the whole North China. This proved the significance of our sampling site for the source apportionment of aerosol in North China in winter. Similar methods were applied frequently, for example, Liu conducted the source apportionment of carbonaceous aerosol in East China based on Ningbo, a background site in Zhejiang Province, China (Liu et al., 2013); Also, Zhang analyzed the carbon species in Mt. Jianfeng, a regional background site on Hainan Island, for the carbonaceous source in South China (Zhang et al., 2014).

Thanks again.

Reference

Gao J, Peng X, Chen G, Xu J, Shi GL, Zhang YC, et al. Insights into the chemical characterization and sources of PM(2.5) in Beijing at a 1-h time resolution. Sci Total Environ 2016; 542: 162-71.

Liu D, Li J, Zhang Y, Xu Y, Liu X, Ding P, et al. The use of levoglucosan and radiocarbon for source apportionment of PM(2.5) carbonaceous aerosols at a background site in East China. Environ Sci Technol 2013; 47: 10454-61.

Moon KJ, Han JS, Ghim YS, Kim YJ. Source apportionment of fine carbonaceous particles by positive matrix factorization at Gosan background site in East Asia. Environment International 2008; 34: 654-664.

Sheesley RJ, Kirillova E, Andersson A, Krusa M, Praveen PS, Budhavant K, et al. Year-round radiocarbon-based source apportionment of carbonaceous aerosols at two background sites in South Asia. Journal of Geophysical Research-Atmospheres 2012; 117.

Waked A, Favez O, Alleman LY, Piot C, Petit JE, Delaunay T, et al. Source apportionment of PM10 in a north-western Europe regional urban background site (Lens, France) using positive matrix factorization and including primary biogenic emissions. Atmospheric Chemistry and Physics 2014; 14: 3325-3346.

Yao L, Yang L, Yuan Q, Yan C, Dong C, Meng C, et al. Sources apportionment of PM2.5 in a background site in the North China Plain. Science of the Total Environment 2016; 541: 590-598.

Yin J, Harrison RM, Chen Q, Rutter A, Schauer JJ. Source apportionment of fine particles at urban background and rural sites in the UK atmosphere. Atmospheric Environment 2010; 44: 841-851.

Yttri KE, Simpson D, Nojgaard JK, Kristensen K, Genberg J, Stenstrom K, et al. Source apportionment of the summer time carbonaceous aerosol at Nordic rural background sites. Atmospheric Chemistry and Physics 2011; 11: 13339-13357.

Zhang YL, Li J, Zhang G, Zotter P, Huang RJ, Tang JH, et al. Radiocarbon-based source apportionment of carbonaceous aerosols at a regional background site on Hainan Island, South China. Environ Sci Technol 2014; 48: 2651-9.

---

## Author Response (AR1)

**Response to Reviewers' Comments and Suggestions - doi: 10.5194/acp-2016-97**

**Dear Editor,**

5 We are thankful very much to you and the anonymous reviewers for the thoughtful comments and suggestions. We have revised this manuscript accordingly. Listed below are our point-by-point responses (blue) to each reviewer's comments (black). In addition, we would like to ask you if we can change the title of our paper as "Radiocarbon and PMF based source apportionment of PM2.5 at a regional background site in North China: insight into the contribution of biomass burning" following the reviewers' comments?

Best regards,

Dr. Chongguo Tian

**15 **Reviewer 1**

Air pollution from PM2.5 is a major health and environmental concern in China. Accurate PM2.5 source attribution is therefore critical for the development of successful air pollution control measures in the region. The study by Zong et al. investigates the sources of PM2.5 at a background site in Northern China during the winter of 2014, utilizing different source apportionment techniques.
By reporting detailed measurements of the elemental composition of PM2.5 collected on 76 quartz filter samples, the manuscript provides interesting new data, which is further grouped into three source regions based on back trajectories analysis. In addition, radiocarbon measurements are used to apportion the carbonaceous fractions of the aerosols into fossil and biomass combustion contributions. These results are combined with a PMF model to derive the major sources of PM2.5 in the region. While the combination of these different techniques is a very good approach for achieving accurate PM2.5 source apportionment and can provide important insight into what is driving the elevated concentration in the region, there are several major problems in the reasoning, explanation and interpretation of the methods and the data reported in the manuscript.

Response and Revisions: Thanks very much for these comments. We have revised this manuscript

30 accordingly. Please find our detailed responses below.

One concern is the use of only one station at a background location, to interpret sources of PM2.5 over a very large area without any discussion of the possibility of local sources affecting the measurements. In one part of the manuscript the authors hypothesize that the elevated PM2.5 at the sampling site are due to the proximity of sources (p.10, line 3-5). But later based on their cluster analysis they say only 11% of the air masses were from the Shandong Peninsula (p. 11 line 18).

- Response and Revisions: Thanks for the comment. Background site is usually adopted to determine the regional-scale concentration burden and the source apportionment of aerosols (Moon et al., 2008; Sheesley et al., 2012; Waked et al., 2014; Yao et al., 2016; Yin et al., 2010). The sampling point in our study is a regional background site in North China, where no obvious emission source nearby. 10 Besides, the back trajectory clusters indicated that more than half of the air masses (54%) during the sampling period were from the Beijing-Tianjin-Hebei region, followed by the air masses from the Mongolia (35%), and Shandong Peninsula (11%), suggesting the area air mass passed through in this paper included the whole North China. This proved the significance of our sampling site for the source apportionment of aerosol in North China in winter. Longkou city is on south side and is 15 closest to the sampling site, so the emission from the city can be considered as the primary local sources. However, we found that only one trajectory in cluster 3 passed the urban area of Longkou, when measured  $PM_{2.5}$  concentration was 95.3  $\mu$ g/m3. The level was lower than the average concentration in cluster 3 listed in Table 2, indicating minor contribution of local source emissions. So following the comment, we have deleted the description of "the elevated PM2.5 at the sampling 20 site was due to the proximity of sources" and added the corresponding content in the revised manuscript (page 10 line 27).

25

5

Further, measuring PM2.5 concentrations based on filter mass is not accurate due to variability within the filter area. Each punch might weigh differently due to the inhomogeneity of the filter itself. There are no references about this technique being used anywhere else.

Response and Revisions: Thanks for the comment. The method that PM2.5 mass collected on filters was very common and usually adopted for the PM2.5 analysis in various studies (Huang et al., 2014; Kong et al., 2015; Liu et al., 2013; Zhang et al., 2015). In this study, before and after every sampling, the filters went through a 24h equilibration at  $25\pm1^{\circ}$ C temperature and  $50\pm2$  % relative humidity and were gravimetrically measured using a temperature and relative humidity-controlled microbalance, such as reported in previous studies (Zhang et al., 2015; Kong et al., 2015; Huang et al., 2014; Liu et al., 2013). Besieds, each filter was weighed at least three times. Acceptable difference among the repetition was less than 10 µg for blank filters and 20 µg for sampled filters, which greatly ensured the precision of our results. According to the reviewer's suggestion, we have added aforementioned references in the revised manuscript (page 4 line 30).

Radiocarbon was only measured on two samples, each representing an air mass coming from a different region. One sample cannot capture any variability in sources and is not a true representative of regional sources. It is also not clear what the uncertainties in the results are and how they could affect the source apportionment. When using radiocarbon, it is particularly important to accurately represent the uncertainty in aerosol samples, as they are very small and usually have high degree of uncertainty, which needs to be considered when interpreting the results.

Response and Revisions: We appreciate the review's comment. It is difficult to capture variability of 15 regional sources by a few samples. However, in order to better achieve our purpose using a few samples for 14C analysis due to the extensive cost of the analysis, the representative capacity of all samples in the two clusters (cluster 1 and 3) was examined thoroughly. OC and EC concentration, and ratios of OC/PM2.5 and EC/PM2.5 of each sample were compared with that in the corresponding cluster by mean test. Finally, two combined samples (including four samples) were selected for 14C 20 measurement from a perfect synoptic process during the sampling period. We added these descriptions in the revised manuscript. In addition, the sizes of samples (M1 and M2) selected for 14C analysis were described as follows. The carbon content of the WSOC, WIOC, EC in the combined sample M1 were 211.03, 209.12 and 283.86 µg, respectively, and these of M2 were 126.76, 182.59, and 200.10 µg, respectively. While the WSOC, WIOC in the blank samples only accounted for 1.94% 25 and 1.15%, respectively, of the average value of M1 and M2. Furthermore, EC was not found in the blank samples. Thus, the blank interference for the fm value of M1 and M2 in the 14C measurement was very small (Liu et al., 2014), implying the uncentainties of 14C measurement was minor in our study (page 6 line 26).

30

5

With respect to the back trajectory analysis and clustering, the manuscript did not provide enough detail to explain how this was performed and the model for the clustering. Also the authors did not include the reference, giving credit to the original HYSPLIT model development.

Response and Revisions: We appreciate the review's comment. Following the reviewer's suggestion,
we have added a sentence "The HYSPLIT model is available on the National Oceanic and Atmospheric Administration Air Resource Laboratory website (www. arl.noaa.gov/ready/hysplit4.html)." for the clarification of the model in the revised manuscript. We also rephrased a sentence to clarify the cluster analysis as: "Finally, a total of 152 trajectories were generated and these trajectories were bunched into three clusters by the toolkit of trajectory cluster analysis embedded in the model." in the revised manuscript (page 7 line 12-14).

In general, the manuscript was very hard to read and understand due to the language (and multiple typos and errors), which needs significant improvement. The title of the manuscript is also misleading, since the biomass-burning signal is not the main focus of the manuscript. The lack of appropriate citations makes it hard to track the accuracy and reliability of the statements and hypothesis throughout the manuscript and makes the interpretation of the results difficult. The details provided in the methods section are not enough to fully understand the analysis or to allow traceability or reproducibility of the results. The results and discussion section was hard to follow too due to the lack of structure within the text and paragraphs summarizing the result from the different experiments and drawing conclusions together.

Response and Revisions: Thanks for the comment. We have made every effort to polish our English and asked a native English speaker to take a proof reading of the final version of the revised manuscript. Inconsistent sentences have been corrected; Following the opinion of the reviewers, we have changed the title as "Radiocarbon and PMF based the source apportionment of regional PM2.5
in North China: insight into the contribution of biomass burning "; Besides, many appropriate citations has been inserted into the revised manuscript, and some details of methods section were also added into manuscript; Some structure changes, such as segmenting some complicated section and adding detailed explain for the figures and tables, were conducted to make readers follow our results and discussion easier.

30

15

4

A list of specific comments and questions are provided below:

- Missing citations throughout the introduction

Response and Revisions: Thanks for the comment. According to the reviewer's suggestion, we have added some references(Cao et al., 2011; Chow and Watson, 2002; Ding et al., 2013; Pui et al., 2014; Sheesley et al., 2012; Szidat, 2009; Szidat et al., 2004) in the revised manuscript.

5

- Spaces missing p. 3 line 20; p. 4 line 10, p. 13 line 20, p. 27, line 22

Response and Revisions: Thanks for the comment. The missing spaces have been added in the revised manuscript.

10

- The introduction (and the rest of the manuscript) does not explain how 14C provides information about fossil and modern sources.

15

Response and Revisions: Thanks for the comment. Following the reviewer's suggestion, we have added a sentence "The underlying principle of 14C measurements is that the radioisotope has become extinct in fossil fuel carbon due to its age (half-life 5730 years), while its contemporary level in nonfossil carbon sources is relatively constant (Szidat, 2009; Szidat et al., 2004)." in the revised manuscript (page 3 line 28-31).

Page 5 line 9: Longkou district and Tianjin are not shown on map. Is Longkou the closest urban region and can it be considered a local source of urban emissions?
Response and Revisions: Thanks for the comment. Following the reviewer's suggestion, we have redrawn the Figure 1, and the locations of Longkou, Tianjin, Changdao and Chengde mentioned in the paper were labeled in the figure. As showed, Longkou city is closest to the sampling site, the
emission from the city can be considered as primary local sources. The sentence has beed added in the revised manuscript (page 4 line 22-24). However, it should be noted that we found that only one trajectory during the sampling period passed the urban area of Longkou, when measured PM2.5 concentration was 95.3 µg/m3. The level was lower than the average concentration in cluster 3 listed in Table 2, indicating minor contribution of local source emissions (page 10 line 27).

Page 5, line 18: Why do the filters undergo a 24h equilibrium period at 25°C? Wouldn't that potentially introduce contamination with the filters absorbing volatile organic compounds during that period?

Response and Revisions: Thanks for the comment. The equilibrium condition and equilibrium period were selected according to the analytical standard of determination of atmospheric particles PM10 5 and PM2.5 in ambient air by gravimetric method (HJ 618-2011, in Chinese) published by the Ministry of Environmental Protection of Republic of the People's China (http://kjs.mep.gov.cn/hjbhbz/bzwb/dqhjbh/jcgfffbz/201109/t20110914\_217272.htm). The method was extensively adopted in previous studies (Liu et al., 2013; Tao et al., 2014). Besides, blank samples were also conducted through the equilibrium process. Concentrations of OC in blank 10 samples were < 3.5% of the average concentration for the total samples, indicating our samples were not contaminated throughout the whole analysis process (including equilibrium).

OC and EC page 6 lines 6-7: The DRI currently uses an updated protocol –IMPROVE\_A (Chow et al., 2007). Which one was used for this manuscript? http://www.dri.edu/eaf-projects Response and Revisions: We appreciate the review's comment. The citation has been corrected (Chow et al., 2007) in the revised manuscript (page 5 line 10).

Page 7, line 14: Radiocarbon methods are not cited properly (p. 7, line 14 Liu et al 2014 is not

20 explaining the process) – correct citation is Zhang et al. 2015 (Env. Sci. & Tech) Radiocarbon-Based Source Apportionment of Carbonaceous Aerosols at a Regional Background Site on Hainan Island, South China

Response and Revisions: We appreciate the review's comment. The citation (Zhang et al., 2014) has been corrected in the revised manuscript (page 6 line 9).

25

Page 7 Line 16: which TOR protocol? (also acronym was not defined anywhere)Response and Revisions: We appreciate the review's comment. It is thermal/optical reflectance (TOR), which has been defined in the revised manuscript (page 5 line 9).

30 Graphitization at CAS and AMS analysis – can you provide citation, sizes, black corrections?

Response and Revisions: Thanks for the comment. The citations (Wacker et al., 2013; Xu et al., 2007; Zhang et al., 2010) have been inserted into the text. On the other hand, the sizes of samples (M1 and M2) selected for 14C analysis were described as follows. The carbon content of the WSOC, WIOC, EC in the combined sample M1 were 211.03, 209.12 and 283.86  $\mu$ g, respectively, and these of M2 were 126.76 182.59 and 200.10  $\mu$ g respectively. While the WSOC, WIOC in the blank samples

5 N

were 126.76, 182.59, and 200.10  $\mu$ g, respectively. While the WSOC, WIOC in the blank samples only accounted for 1.94% and 1.15%, respectively, of the average value of M1 and M2. Furthermore, EC was not found in the blank samples. Thus, the blank interference for the fm value of M1 and M2 in the 14C measurement was very small and was ignored in this study.

**10 Page 8: HYSPLIT and EPA PMF v5.0 citations**

Response and Revisions: Thanks for the comment. Following the reviewer's suggestion, we have added website, from which the software can be obtained, in the revised manuscript (page 7 line 12 and page 7 line 19).

- Page 9, line 15 the text does not follow the order reported in Table 1 (notes on tables and figures will be provided at the end)
   Response and Revisions: Thanks for the comment. The sequence of chemical species in Table 1 has been rearranged according to the description of the text.
- Page 9, line 17 what is the uncertainty of the max PM2.5 value? Also, there is no need to report significant digits, as the range is so large
   Response and Revisions: We appreciate the review's comment. Following the reviewer's comments, the description about the maximum value has been deleted in the revised manuscript.
- 25 Page 10, line 2: It is not clear where Tuoji Island is and how it is relevant in the comparison. Response and Revisions: Thanks for the comment. After careful consideration, the description about Tuoji Island in original draft may not be appropriate for our purpose. So the section about Tuoji has been deleted in the revised manuscript.
- 30 Page 10, line 8-9. Write chemical formula for sulfate, nitrate and ammonium and use them

throughout the text. The lack of consistency with the names of the chemical elements and radiocarbon vs  $^{14}$ C makes the text harder to read.

Response and Revisions: We appreciate the review's comment. Following the reviewer's suggestion, we replaced sulfate, nitrate, ammonium by their chemical formulas and radiocarbon by  ${}^{14}C$  throughout the revised menuscript.

5 throughout the revised manuscript.

Page 10, line 20-1. There are major issues with the chemical elements here – The ones written out are not the correct elements!

Response and Revisions: Thanks for the comment. According the review's suggestion, the sentence

10 has been rewritten as "The high contribution agreed with the regional scale emission of their precursors in the North China, as reported that the emission amounts of  $SO_2$ ,  $NO_x$  and  $NH_3$  were about 10, 5, and 5 times higher compared to OC in the region." in the revised manuscript (page 10 line 6).

15 Page 11, line 22: Mean test?

Response and Revisions: Thanks for the comment. Mean test is a statistical method, which can be used to determine if two sets of data are significantly different from each other. The analysis can be performed by SPSS software.

20 Break next section in paragraphs. It is very hard to read. Response and Revisions: Thanks for the comment. This suggestion has been employed in the revised manuscript.

Page 12, second paragraph – write out chemical elements

25 Response and Revisions: Thanks for the comment. This suggestion has been employed in the revised manuscript.

Page 12, line 21. Cite the sources for sea salt composition

30 Response and Revisions: Thanks for the comment. This suggestion has been employed in the revised

manuscript (page 11 line 24).

Page 13: It is really hard to follow and it might be helpful to break it up in a couple paragraphs Response and Revisions: Thanks for the comment. This suggestion has been employed in the revised manuscript.

5

Page 14: It is not clear how the Enrichment Factor (EF) method works, if/how it has been applied in other studies and how reliable it is. Original citations are missing. More description is necessary. It will also be helpful to plot the EF calculated for all elements.

Response and Revisions: Thanks for the comment. The assistance of EF section for the source 10 signals of aerosols in original draft may not be appropriate for our purpose. So after careful consideration, the section about EF has been deleted in the revised manuscript.

Page 14: Using OC/EC ratio and NO3/nss-SO4 a traces :

Response and Revisions: We appreciate the review's comment. The sentence has been changed as 15 "The ratios of OC/EC and NO3"/nss-SO42 were used as tracers to assess source signals of the three clusters." in the revised manuscript (page 12 line 25).

Lack of citations OC/EC p. 14 line 21

Response and Revisions: Thanks for the comment. Following the review's suggestion, a citation has 20 been listed in the revised manuscript (page 12 line 28).

**Mean test (p. 15 line 2)?**

Response and Revisions: Thanks for the comment. As stated above, mean test is a statistical method,

25

which can be used to determine if two sets of data are significantly different from each other. The analysis can be performed by SPSS software.

When calculating OC/EC ratios what is the uncertainty in the OC and EC and the ratio.

Response and Revisions: We appreciate the review's comment. Following the reviewer's suggestion,

we assessed the uncertainty in OC, EC and their ratio based on results of eight pair replicate samples 30

using the method of extreme difference. The results showed that the uncertainties of OC, EC and the ratio were 5.61%, 5.51% and 4.72%.

Lack of citations SO2 and NOx p. 15 line 7

5 Response and Revisions: Thanks for the comment. Following the review's suggestion, a citation has been listed in the revised manuscript (page 13 line 5).

Page 15, line18: Where is Chengde and how is it relevant?

Response and Revisions: We appreciate the review's comment. Chengde city has been displayed in
 the revised Figure 1. We mentioned the ratio of NO3-/nss-SO42- in Chengde here in order to infer more obvious emission of stationary sources in the region where air masses of cluster 2 were passed.

Source app of carbonaceous PM2.5 (pages 16-19)

15

Response and Revisions: Thanks for the comment. The description has been changed as "Source apportionment of carbonaceous  $PM_{2.5}$ ".

The back trajectory analysis is not clear in the description of how the samples were combined. The figure does not help either.

Response and Revisions: Thanks for the comment. Following the reviewer's suggestion, we added
one section about how the samples were combined in the method section of the revised manuscript.
Besides, selecting reason and back trajectory analysis were also mentioned in this section (page 8 line 12- page 9 line 12).

Page 16 line 11 - 19 unclear and belong in the methods

25 Response and Revisions: We appreciate the review's comment. Following the suggestion, we have rewritten the description and changed it in the method section (page 8 line 22- page 9 line 12).

Page 18 line 2: What is the WSOC and WIOC concentration uncertainty? The reported numbers have two significant digits, but are the methods accurate enough to report concentrations of this precision?

30 Response and Revisions: Thanks for the suggestion. In this study, WSOC were directly measured by

a total organic carbon (TOC) analyzer (Shimadzu TOC-VCPH, Japan). WIOC was quantified by OC given by the TOR protocol subtracting WSOC. The uncertainties of WSOC were calculated by the method of extreme difference. The uncertainties of for M1 and M2 were 6.7% and 5.3%, respectively. The uncertainties of WIOC were calculated in this study according to the following equations: WIOC = sqrt ( $OC^{^2} + WSOC^{^2}$ ). The uncertainties of for M1 and M2 were 8.8% and 7.8%, respectively. According to the uncertainty analysis, we gave 2 digits for the data as shown in revised Table 3. We have mentioned this calculation method in our revised paper.

M1 and M2 are significantly different, but could this be due to high level of uncertainty in the measurement? Can it possibly be affected by local sources?

Response and Revisions: Thanks for the comment. According to the previous uncertainty analysis, the contemporary carbon fractions of OC were in the range from 0.5934 to 0.5936 for M1 and from 0.454 to 0.457 for M2, respectively. It suggests that the significant differences were attributed the source contribution, rather than the uncertainty of the measurement. The uncertainty of EC was less
15 than that of OC, so the differences for EC were also attributed the source contribution. The local sources affected weakly the difference because the air masses didn't pass the local source region as shown in Figure 2.

Page 19, line 15: Fpeak and Q are not defined

5

10

- 20 Response and Revisions: Thanks for the comment. The sentence has been rewritten as "After iterative testing from 5 to 15 factors for the model scenarios, we found the minimum deviation of the source apportionments of OC and EC between the results from 14C measurement and a PMF model scenario with an  $F_{peak}$  value of 0 and the lowest Q values (6245)." (page 15 line 12)
- Page 20, line 6: Citations missing
   Response and Revisions: Thanks for the comment. The missing citations have been added in revised
   paper (page 15 line 22).

Page 24, line 3: PMF and  $^{14}$ C results are compared within 3-4% precision, but it is likely the uncertainty is higher.

Response and Revisions: Thanks for the comment. PMF results were selected according to the minimum deviation between the results from PMF models and 14C measurements. Besides, the uncertainties of 14C measurement for OC and EC of the M1 were 6.8% and 3.8% (see Table 3), which were comparable with the difference between PMF and 14C results. The minor uncertainty of 14C measurements ensured its accuracy, thus the compare results of PMF and 14C results could be trusted in this study.

5

20

Implications for PM alleviation – since it was mostly focused on biomass burning it seemed to not fully capture the scope of the work.

- 10 Response and Revisions: We appreciate the review's comment. According to the source apportionment, coal combustion and vehicle emission were the primary contributors of  $PM_{2.5}$  in the North China during winter. The two sources have been considered as the most leading emission sectors in the air pollution control program. While biomass burning, also an important emission, has been only considered slightly in the control program. Thus, biomass burning emission was
- 15 highlighted in the manuscript and a section was conducted for the implications for PM alleviation from the aspect of biomass burning.

It would have been very helpful if there were a section in the manuscript where all the results were drawn together so the reader can logically conclude how the different techniques complemented each other and what the final result of the different analytical techniques was.

Response and Revisions: Thanks for the comment. Following the reviewer's suggestion, we have added more description about the result analysis explanation in the revised manuscript. This would be convenient readers to grasp our key point in this study.

There are a lot of typos in the references.
 Response and Revisions: Thanks for the comment. The typos in study have been corrected in revised manuscript.

**Tables and Figures:**

Table 1: It is not clear why the table is split in two and why the units are different. It would be much

easier to interpret the content if everything is aligned into 3 columns and the units are consistent.

Response and Revisions: Thanks for the comment. Following the reviewer's suggestion, we rearranged the contents as mentioned above. The table is split in two columns in order to show the data by different units. We retained the structure because it is able to show more information (page 31).

**Table 2: Make into one page**

Response and Revisions: Thanks for the comment. Table 2 has been made in one page (page 32).

10 Table 3: Uncertainty needs to be reported for each measurement Response and Revisions: Thanks for the comment. The uncertainty has been reported in revised manuscript (page 33).

Table 4: Rows 1, 3, and 4 do not add up to a 100% when horizontally summed. The difference is minimal, but what is the reason for it? Also the significant digits are probably not necessary, based on

the accuracy of the analysis. Response and Revisions: Thanks for the comment. The difference was attributed to maintain the

uniformity significance digit of the data. Following the suggestion, we gave two decimal places for all these data in the revised table (page 33).

20

15

5

Figure 1. Some locations that were mentioned in the text are not included in the figure. Response and Revisions: Thanks for the comment. Figure 1 has been redrawn, and the locations mentioned in the manuscript have been added in the revised figure (page 34).

- 25 Figure 2. This is very difficult to follow, a more detailed description of what the figure is showing will be helpful Response and Revisions: Thanks for the comment. More description has been added in the revised figure (page 35).
- 30 Figure 4. The figure caption should explain why the B&B and Fossil emissions from the PMF do not

add to a hundred in the bars. It is described in the text, but a note in the figure will make it much easier to understand.

Response and Revisions: Thanks for the suggestion. The explanation has been added in the revised figure (page 37).

**5**

**Reviewer 2**

In present study, a more reliable source apportionment of  $PM_{2.5}$  on the regional scale in the North China in winter is provided using PMF simulation and the radiocarbon measurements. It is consistent work performed on a good level. It looks like the title of the paper does not relate the content of the paper well. It announces Biomass burning contribution to regional  $PM_{2.5}$  during winter in general in the North China while the work was done only on one site. If one site represents the whole North

10

**China?**

Response and Revisions: Thanks very much for these comments. Please find our detailed responses below.

**15**

20

1. It looks like the title of the paper does not relate the content of the paper well.

We appreciate the review's comment. As mentioned by the reviewer, the sources apportioned by the paper are not only biomass burning but also coal combustion, vehicle emission and so on. We use the title "Biomass burning contribution to regional  $PM_{2.5}$  during winter in the North China" in order to highlight the important of biomass burning, which is often underestimated or ignored in some areas in China (Gao et al., 2016). However, we agree with the opinion of the reviewer, and change the title as "Radiocarbon and PMF based the source apportionment of regional  $PM_{2.5}$  at a regional background site in North China: insight into the contribution of biomass burning.".

**25 2. If one site represents the whole North China?**

Thanks for the comment. Background site is usually adopted to determine the regional-scale concentration burden and the source apportionment of aerosols (Moon et al., 2008; Sheesley et al., 2012; Waked et al., 2014; Yao et al., 2016; Yin et al., 2010; Yttri et al., 2011). The sampling point in our study is a regional background site in North China, where no obvious emission source nearby.

30 Besides, the back trajectory clusters indicated that more than half of the air masses (54%) during the

sampling period were from the Beijing-Tianjin-Hebei region, followed by the air masses from the Mongolia (35%), and Shandong Peninsula (11%), suggesting the area air mass passed through in this paper included the whole North China. This proved the significance of our sampling site for the source apportionment of aerosol in North China in winter. Similar methods were applied frequently, for example, Liu conducted the source apportionment of carbonaceous aerosol in East China based on Ningbo, a background site in Zhejiang Province, China (Liu et al., 2013); Also, Zhang analyzed the carbon species in Mt. Jianfeng, a regional background site on Hainan Island, for the carbonaceous source in South China (Zhang et al., 2014).

1

| 1  | Formatted: Font: Bold                                           |
|----|-----------------------------------------------------------------|
| 1  | Formatted: Normal                                               |
| -{ | Formatted: English (U.S.)                                       |
| ١  | Formatted: Don't add space between paragraphs of the same style |

25

**Abstract**

Source apportionment of fine particles ( $PM_{2,5}$ ) at a background site in the North China in the winter, of 2014 was assessed by via statistical analysis on the chemical species grouped by the trajectory elusters, radiocarbon (14C) measurement, and the Positive Matrix Factorization (PMF) modeling 5 linked with the 14C measurement. During the sampling period, . Results showed that the concentration of PM2.5 was 77.6 ± 59.3  $\mu$ g m-3, and the sulfate of which SO42- concentration was the highest, followed by nitrate, NO3, organic carbon (OC), elemental carbon (EC) and ammonium, NH4+, respectively. Demonstrated by the backward trajectory, more than half of PM2.5-was found the air 10 mass during the sampling period was from the Beijing-Tianjin-Hebei (BTH) region, followed by the Mongolia and the Shandong Peninsula. The cluster Cluster analysis of chemical species showed that  $PM_{2.5}$  from the Shandong Peninsula had an obvious signal of biomass burning emission in the  $PM_{2.5}$ from the Shandong Peninsula, while that the PM2.5 from the BTH region showed a vehicle emission pattern. The This finding was further confirmed by the radiocarbon14C measurement of OC and EC in two merged samples selected from a successive synoptic process. The 14C measurement results 15 indicated that biogenic and biomass burning emission contributed 59% and 52% to OC and EC concentrations, respectively, when air masses originated from the Shandong Peninsula, and the contributions fell to 46% and 38%, respectively, when the prevailing wind changed and came from the BTH region. In addition, The minimum deviation of the source apportionments from PMF results 20 and 14C measurement was usedadopted as the optimal choice of the model exercises. Here, two minor overestimations with the same range (3%) suggested that the PMF results provided a reasonable source apportionment of regional PM2.5 in the North China during winter, this study. Based on the **PMF** results above, eight main sources were identified; of which these, coal combustion, biomass burning, and vehicle emissions emission were the largest contributors of  $PM_{2.5}$ , accounting 25 for 29.6%, 19.3% and 15.8%, respectively. Compared with the overall source apportionment, the contribution contributions of vehicle emission-, mineral dust and coal combustion, biomass burning increased slightly when air masses came from the BTH region, the contribution of mineral dust and coal combustion rose obviously when air masses were from the Mongolia-with high speed, and biomass burning became the dominant contributor when air masses carried from the Shandong Peninsula. As the largest contributor to PM2.5 in winter of North China, respectively. Since coal 30

combustion hasand vehicle emission have been identified asconsidered the most-leading emission sectorsectors to be controlled for improving the air quality by the government. Vehicle emission contributed significantly to the  $PM_{2.5}$  levels in the BTH region, which has also been considered to control as the second major emission sector. Biomass, biomass burning emission was highlighted in the present study because of its dominant contribution to  $PM_{2.5}$  burden in the Shandong Peninsula. Some suggests were provided to wake farmers from agricultural residue burning in household and field.

Keywords: Source apportionment, PMF, 14C measurement, PM2.5

10

5

**1** Introduction**

15

20

25

In recent years, air pollution has become a top environmental issue in China, and the main4 concern is on the fine particulate matter less than 2.5 micrometers in diameter ( $PM_{2.5}$ ) (Huang et al., 2014;Sheehan et al., 2014)(Huang et al., 2014; Sheehan et al., 2014). Fine particulate aerosols have a strong adverse effect on human health, visibility, and directly or indirectly affect weather and climate-(Pui et al., 2014). The negative effects on the public health, including the damage to the respiratory and cardiovascular systems, the blood vessels of the brain, and the nervous system, have triggered both public alarm and official concern in China (Kessler, 2014). (Kessler, 2014). In response to this great concern, the Chinese government has introduced the Action Plan for Air Pollution Prevention and Control (2013–17), which aims at marked improvements in the air quality up tountil 2017. In the plan, the severest supervisionmost severe regulation for the improvement is a reduction of 25% in the annual average concentrations of  $PM_{2.5}$  by 2017-(Chinese-State-Council, 2013). It has been applied in the North China because the region has become the most severely polluted area in China, characterized by increasingly frequent haze events and regional expansionexpansions of extreme air pollution in recent years (Hu et al., 2015;Boynard et al., 2014;Quan et al., 2014Hu et al., 2015;

**Boynard et al., 2014).**

Basically, the key point of reducing PM2.5 concentrations is to control its sources. Reliable

**Formatted: Don't add space between paragraphs of the same style**

Formatted

paragraphs of the same style

[revised manuscript text omitted]

**2 Materials and methods**

**2.1 Sampling site and sample collection**

10

The sampling campaign was conducted, from January 3 Jan-to February 11-Feb, 2014, at the Longkou Environmental Monitoring Station of the State Ocean Administration of China (37°41'N, 120°16'E)), on the Qimu Island. The island is extended extends to the Bohai Sea in the westward direction westwards, and is surrounded by sea on its other three sides, as shown in Fig. 1. And the The sampling site is located in about approximately 15 km northwest from of the Longkou urban district and 300 km southeast from of the Beijing-Tianjin-Hebei (BTH) region. PM2.5-atLongkou city is closest to the sampling site exhibited largely a regional pollution signal during the sampling period, because the air masses were mainly carried from the BTH region, and the Mongolia subjected by the East Asian winter monsoon as illustrated by the backward trajectories in Fig. 1, which will emissions from the city can be elaborated later considered the primary local sources.

20

25

15

A total of 76 PM2.5 samples were collected continuously on quartz fiber filters (Whatman, QM-A,  $20.3 \times 25.4$  cm2, heated at 450 °C for 6 h before use) using a Tisch high volume sampler at a flow rate of 1.13 m3 min-1 during the sampling period. The duration for each sample was 12 h, from 06:00–18:00 and from 18:00–06:00 (local time) the next day. Before and after each samplingsample, quartz fiber filters went through awere subjected to 24 h equilibration at  $25\pm\pm1$  °C temperature and  $50\pm\pm2$ % relative humidity, and were then-were analyzed gravimetrically using a Sartorius MC5 electronic microbalance7 (Zhang et al., 2015; Liu et al., 2013; Huang et al., 2014). Each filter was weighed at least three times. Acceptable difference among the repetitions was less than 10 µg for a blank filter and less than 20 µg for a sampled filter. After weighing, loaded filters were stored in a refrigerator at -20 °C until chemical analysis. AlsoIn addition, field blank filters were collected to subtract the possible contamination that occurredoccurring during or after sampling.

30 2.2 Chemical analysis

**2.2.1 OC and EC**

Organic carbon (OC) and elemental carbon (EC) were analyzed by a Desert Research Institute (DRI) Model 2001 Carbon analyzer (Atmoslytic Inc., Calabasas, CA) following the Interagency Monitoring of Protected Visual Environment (IMPROVE A) thermal/optical reflectance (TOR) protocol (Chow et al., 1993).(Chow et al., 2007). A punch of 0.544 cm2 from each quartz filter was 5 heated to produce four fractions (OC1, OC2, OC3 and OC4) in four temperature steps (140, 280, 480, 580 °C) under a non-oxidizing helium atmosphere and then in 2%  $O_2/98\%$  He atmosphere at 580 °C (EC1), 740 °C (EC2), and 840 °C (EC3) for the EC fractions. At the same time, pyrolyzed organic carbon (POC) was produced in the inert atmosphere, which decreased the reflected light to correct 10 for charred OC. The concentrations of OC and EC were obtained according to the IMPROVE protocol, OC = OC1 + OC2 + OC3 + OC4 + POC and EC = EC1 + EC2 + EC3 - POC. The detection limits of the method for OC and EC were 0.82 and 0.20 µg cm-2, respectively. In addition, blank filters and replicate samples were examined simultaneously after analyzing a batch of 10 samples in order to obtain their inherent OC and EC concentrations on the filterfilters and to evaluate 15 measurement accuracy, respectively. In this study, the contributions of OC and EC from blank filters were < 3.5 and 0.6% of their respective average concentrations. Furthermore, comparison with average values from replicate analyses showed a good precision with relative deviations of 5.9%. The uncertainties of OC (5.6%) and EC (5.5%) were calculated from the replicate measurements.

2.2.2 Water-soluble ions and metal elements

Two punches with 47 mm diameter punches were cut off from each quartz fiber filter, one of which, one was subjected to Milli-Q water extraction for ionic measurement, and the other wasunderwent induced acid digestion for elemental measurement. The concentrations of water soluble ions (Na+, NH4+, K+, Mg2+, Ca2+, Cl-, NO3- and SO42-) were determined by ion chromatograph (Dionex ICS3000, Dionex Ltd., America) based on the analysis method (Shahsavani et al., 2012). (Shahsavani et al., 2012). The concentrations of metal elements (including Ti, V, Mn, Fe, Co, Ni, Cu, Zn, As, Cd and Pb) were estimated on the basis of via inductively coupled plasma coupled with mass spectrometerspectrometry (ICP-MS of ELAN DRCII type, Perkin Elmer Ltd., Hong Kong) following the previous method (Wang et al., 2006). (Wang et al., 2006). The detection limit of water-soluble ions was 10 ng4 ml-1 with the error < 5%, and 1ml1 ml RbBr of 200 ppm was put in the solution as an internal standard before analysis. Similarly, the The resolution of ICP-MS

25

20

ranged from 0.3 to 3.0 amu with a detection limit lower than  $\leq 0.01$  ng/ml, and the error < 5%. ElementFive ppb elemental Indium (In) of 5 ppb, as the internal standard, was put in the solution before analysis- as an internal standard.

2.2.3 Radiocarbon14C measurement

[revised manuscript text omitted]

$$x_{ij} = \sum_{k=1}^{p} g_{ik} f_{kj} + e_{ij}$$
(1)

Formatted

where  $x_{ij} - x_{ij}$  is the measured concentration of the jth - jth species in the ith - jth sample,  $f_{kj} - f_{kj}$  is the profile of jth - jth chemical species emitted by the kth - kth source,  $g_{ik} - g_{ik}$  is the amount of mass contributed by kth - kth source to the ith - ith sample, and  $e_{ij} - e_{ij}$  is the residual for each samples/species. The matrixes matrices of g - g and f - f are determined by minimizing an objective function (Paatero et al., 2014) (Paatero et al., 2014).

After determination and interpretation of these factor profiles, source contributions identified by the modelTo further confirm PM2.5 sources apportioned by the PMF model, the source contributions of OC and EC were examined by 14C measurement. The modeled source contributions were merged into two groups according to fossil and contemporary carbon sources. Then the contribution fractionfractions of fossil or contemporary carbon sources to OC and EC wascould be compared with the results derived from 14C analysismeasurement for the specified sample to confirm the model results samples as:

$$R_{ij} = \sum_{k=1}^{n} g_{ik} f_{kj} / \sum_{k=1}^{p} g_{ik} f_{kj}$$
(2)

Where R where R is the contribution fraction, and matrixesmatrices of  $\underline{g}$  and  $\underline{f}$  are the same as in eqn(1). The subscripts i subscript *i* is a specified sample,  $\underline{j}$  is species of OC or EC. In species, *n* is the number of fossil or contemporary carbon sources, and  $\underline{p}$  is the number of all sources. The minimum deviation of PM2.5 source contributions apportioned by the PMF exercises and 14C measurements was used to determine the final model scenario. The model results were treated as providing a more reliable solution for the source apportionment.

**20 **2.4 Principle of samples selected for 14C analysis**

5

10

15

25

The comparison of OC and EC focused on cluster 1 and cluster 3 because most species of  $PM_{2.5}$ in these two clusters were statistically greater than in cluster 2, as elaborated later. To better achieve the comparison using a few samples for 14C analysis due to its extensive cost, the representative capacity of all samples in the two clusters was examined thoroughly. It is expected that PMF can better interpret those data close to the average condition of each chemical species, since the method utilizes error-minimizing estimates to decompose a matrix of sample data into two matrices under

| Field Code Changed |  |
|--------------------|--|
| Field Code Changed |  |
|                    |  |

| 1 | Field Code Changed |
|---|--------------------|
| - | Field Code Changed |
| 1 | Field Code Changed |
|   | Field Code Changed |
| 1 | Field Code Changed |
|   | Field Code Changed |
| 1 | Field Code Changed |
|   |                    |

strict non-negativity constraints for the factors (Paatero et al., 2014). Therefore, OC and EC concentrations, and ratios of OC/PM2.5 and EC/PM2.5 of each sample, were compared with those in the corresponding cluster by mean test.

Finally, two combined samples were selected from a perfect synoptic process during the sampling period. The synoptic process occurred during January 16th and 18th, 2014. As shown in Fig. 5 2, the first half of air masses in the synoptic process were derived from the south and passed through the Shandong Peninsula (cluster 3) and the bottom half were from the north and passed over the BTH region (cluster 1). Thus, two samples collected continually from 06:00 to 18:00, January 16th and from 18:00 to 06:00 the next day in the first half of the synoptic process were merged into one sample (M1) for the 14C analysis. Similarly, other two samples collected continually from 18:00 to 10 6:00, January 17th and from 06:00 to 18:00 in the next day were combined into the other sample (M2). M1 reflected the signal of air masses coming from the Shandong peninsula, while M2 showed the pattern of air masses from the BTH region. Mean test showed that except for a significant high ratio of EC/PM2.5, the OC and EC concentrations and the OC/PM2.5 ratio of M2 were negligibly different from cluster 1, at a 95% significance level, indicating its perfect representative capability 15 for further carbonaceous analysis. However, M1 was not ideal because only ratios of OC/PM2.5 and EC/PM2.5 had no statistical difference, OC and EC concentrations were significantly higher than that in the cluster 3 at the same significance level. Even so, the samples were still considered for  $^{14}C$ analysis because they were from a faultless synoptic process during the sampling period. Continuous 20 samples were more dramatic than insular samples. In addition, the insignificant difference of the ratios of OC/PM2.5 and EC/PM2.5 assured the validity for PM2.5 source assessment, which was more

important than concentration in this study.

**3 Results and discussion**

25

3.1 General characteristics of PM2.5 and chemical components

Table 1 lists a statistical summary of the concentrations of  $PM_{2.5}$ , water-soluble ions, carbonaceousspecies and metal elements during the sampling period.**3 Results and discussion**

**3.1 General characteristics of PM2.5 and chemical components**

Table 1 lists a statistical summary of the concentrations of PM2.5, water soluble ions,

earbonaceous species and metal elements during the sampling period. As shown, the mean

Formatted: Don't add space between paragraphs of the same style, Line spacing: 1.5 lines concentration of PM2.5 was 77.6 ± 59.3 μg m-3, and the maximum value was 305 μg m-3, which werewas more than two-and eight times higher than the grade I national standards (35 μg m-3, Ministry of Environmental Protection of China: GB 3095-2012, www.zhb.gov.cn<del>, 2012 02 29).</del> Although the level of PM2.5 concentration on Qimu Island was higher than the national standard, but it was much lower than that observed in winter in megacities of the North China, such as in Beijing (208 μg m-3 of PM2.1 in 2013)(Tian et al., 2014) and Tianjin (221 μg m-3 in 2013)(Han et al., 2014). Besides, the concentration level was significantly higher than the result (42.4 μg m-3 in winter, 2012) measured at a national station for background atmospheric monitoring on Tuoji Island, located on the Bohai Strait (Wang et al., 2014). The obviously high concentration of PM2.5 on Qimu Island was possibly attributed to the short distance between the sampling site and emission source on the Shandong Peninsula (see Fig. 1) and strong deposition of particles due to high precipitation over the Bohai Sea in the winter of 2012 (Zhang et al., 2014).

5

10

15

, 2012-02-29). Although the level of  $PM_{2.5}$  concentration on Qimu Island was higher than the national standard, it was much lower than that observed in winter in the megacities of North China, such as in Beijing (208 µg m-3 of  $PM_{2.1}$  in 2013) (Tian et al., 2014) and Tianjin (221 µg m-3 in 2013) (Han et al., 2014).

For the PM2.5 components, water-soluble inorganic species (WSIS) were the dominant species, accounting for 37 ± 16 % of the PM2.5 mass concentrations. Among the ionic concentrations, sulfateions,  $SO_4^{2^2}$  ranked the highest with a mean concentration of  $14.2 \pm 18.0 \ \mu g \ m^{-3}$ , followed by nitrateNO3± (11.9 ± 16.4  $\mu g \ m^{-3}$ ) and ammoniumNH4± (3.11 ± 2.14  $\mu g \ m^{-3}$ ). The sum of the three secondary inorganic aerosols constituted the majority (81 ± 12 %) of the total WSIS concentrations. In addition, the average concentrations of OC and EC were 6.85 ± 4.81 and 4.90 ± 4.11  $\mu g \ m^{-3}$ , accounting for 9.2 ± 2.1 % and 6.4 ± 1.8 % of the PM2.5 concentrations, respectively. Total concentrations of analyzed metal elements were 665 ± 472 ng m-3, accounting for 0.93 ± 0.50 % of the PM2.5 mass concentration. Among the measured metal elements, the concentration of Fe (408 ± 285 ng m-3) was the highest, followed by Zn (107 ± 142 ng m-3), and Pb (88.4 ± 85.7 ng m-3). Formatted: Don't add space between paragraphs of the same style, Line spacing: 1.5 lines, Widow/Orphan control Formatted: Font: 12 pt Formatted

The relative contribution of sulfate, nitrate  $SO_4^{2-}$ ,  $NO_3^{-}$ , and ammonium  $NH_4^{\pm}$  to the PM2.5 at the sampling site was obviouslyclearly higher than that in the cities, such as Beijing and Tianjin, within the North China, while the organic matter was obviouslyclearly lower. The high contribution <del>agreed</del>contributions of  $SO_4^{2-}$ ,  $NO_3^{-}$ , and  $NH_4^+$  agree with the regional scale emissionemissions of their precursors in the North China, as it has been reported that the emission amounts of SO2, 5  $NO_2NO_{xx}$  and  $NH_3$  emissions were about approximately 10, 5, and 5 times higher compared to OC in the region, respectively (Zhao et al., 2012)(Zhao et al., 2012). The This finding was also in agreement with that results measured at Changdao Island (Feng et al., 2012)(Feng et al., 2012). The island, located at the demarcation line between the Bohai Sea and the Yellow Sea, is a resort with little industry at about approximately 7 km north from of the Shandong Peninsula (Feng et al., 2012)(Feng al., 2012). MeasurementMeasurements at the island was treated were interpreted as et providingshowing patterns of atmospheric outflow and regional pollution in North China (Feng et al., al., 2007)(Feng et al., 2012; Feng et al., 2007). It suggested that our 2012:Feng measurementmeasurements also provided provide a regional signal of  $PM_{2.5}$  pollution in the North China. Besides, sulfateFurthermore,  $SO_4^{2-}$  was the largest contributor of  $PM_{2.5}$ -as aforementioned. The, and the highest contribution contributor is usually regarded as a regional pollution signal in winter. This is because of the during low temperature conditions in PM2.5 source areas there is a lack of a fast conversion rate of SO2 to sulfate SO42- in eloud clouds or aerosol droplet processes droplets and oxidation reactionreactions via OH free radical in low temperature condition in source areas (Hu 20 et al., 2015). This also indicated that our measurement radicals (Hu et al., 2015). Thus, our measurement largely reflected largely a pollution pattern on a regional scale, rather than just in source areas.

**3.2 Source signals based on cluster analysis\_**

25

10

15

As shown in Fig. 1, the 48-h back trajectory clusters indicate dindicate that more than half of the air masses (54%) during the sampling period were from the BTH region, defined as (cluster  $1_{7}$ ), followed by the air masses from the Mongolia (35%, cluster 2). Air masses of these two types traveled about 200 and 250 km, respectively, over the Bohai Sea before arriving at the sampling site. Thus, the atmospheric pollutants carried by the two kindkinds of air masses were mixed well during the transport, showingcreating regional pollution signals. Only a small part of the air masses (11%) were from the Shandong Peninsula (cluster 3), reflecting potentially reflecting a mixed contribution

30

of local and regional sources from south area of the sampling site. In addition, only one trajectory in cluster 3 passed the urban area of Longkou, when measured  $PM_{2.5}$  concentration was 95.3 µg m-3. This level was lower than the average of  $PM_{2.5}$  concentrations in cluster 3, listed in Table 2, indicating minor contribution of local source emissions. To reveal the pollution patterns and source signals of  $PM_{2.5}$  carried by air masses from the three different regions, chemical species of  $PM_{2.5}$

5

10

15

20

were grouped according to the three trajectory clusters, as listed in Table 2. Generally, mean test showed that the concentration levels of PM2.5-and its most abundant

species types of PM2.5 in elusterclusters 1 and eluster 3 were insignificant differencesare both insignificantly different (p > 0.05) and statistically higher than that in cluster 2 (p < 0.01), as shown in Table 2. The patterns wereobserved are consistent with the spatial distributions of their emissions and concentrations in the-North China; as reported the, there are stronger emissionemissions and the more serious pollution in the BTH region and Shandong Province compared with thatthan in Inner-Mongolia and Liaoning (Zhao et al., 2012;Yang et al., 2011)(Zhao et al., 2012; Yang et al., 2011). And-Compared with Shandong Peninsula, the pollution in BTH region may be more serious in thatbecause it hastravels much longer distance fromdistances to the sampling site-compared with Shandong Peninsula, while their concentration levels of, yet PM2.5 were insignificantconcentrations attributed to the two areas are insignificantly different. In addition, the mean wind speed of cluster 2 was 7.60 m s-1, which was markedly higher than that of cluster 1 (4.79 m s-1) and cluster 3 (4.86 m s-1). The windWind speeds were yieldeddetermined by averagedaveraging hourly moving distancedistances of those-air masses during a 48 hoursh period. The higher wind speed of cluster 2 resultedlikely partly incontributes to the lower PM2.5 level at the-sampling site, since that-high wind speed could provide a favorable diffusion conditions for atmospheric pollutants.

25

Some anomalies compared with previous discussion provided different source signals amongamongst the clusters. For instance, theK+ concentration of K+-was significantly higher in cluster 3 than that in cluster 1, while the titanium (Ti) concentration was obviously lower. This reflected the relativereflects relatively high emissionemissions of K+ in the Shandong Peninsula and Ti in the BTH region from both natural sources and anthropogenic activities. In contrastLikewise, the concentration of Na+ in cluster 2 was markedly higher than that in clusterclusters 1 and cluster 3, showing the large contribution of sea salt particles generated by sea spray under high wind speed into cluster 2.-It PM2.5 concentrations. This suggested that sea salt sources couldshould not be ignored in

14

this study, due to that the proximity of the sampling site was close to the Bohai Sea.\_
Sea salt emission is emissions are comprised of Cl-, SO42-, Na+, K+, Mg2+ and Ca2+, (Ni et al., 2013). The amounts of the different chemical species from in sea salt emission emissions can be sustained with determined from using Na+ as the tracer of sea salt, so; the amount amounts of these species from non--sea salt (nss-) emission emissions can be expressed as nss X = X - [Na+] × a, :

$$nss - x = x - [Na^+] \times a$$

where X indicates the Cl-, SO42-, K+, Mg2+ and Ca2+ concentrations, and a is the typical equivalent concentration ratio of the corresponding species to Na+, such as+ in average seawater:  $Cl^{-}/Na^{+}$  (1.80),  $SO_{4}^{2-}/Na^{+}$  (0.25),  $K^{+}/Na^{+}$  (0.036),  $Mg^{2+}/Na^{+}$  (0.12) and  $Ca^{2+}/Na^{+}$  (0.038) in average seawater (Ni et al., 2013).(Ni et al., 2013). If the calculated concentration of no-non-sea salt (nss-) chemical species is negative, then there is no the species excess exists.species exist. According to the calculation, for corresponding total chemical concentration levels grouped in clusters from 1 to 3, nss-Cl- accounted for 55  $\pm$  29%, 19  $\pm$  24% and 77  $\pm$  10%;% of total Cl-; nss-SO42- accounted for 99  $\pm$ 2%, 95 ± 4% and 99 ±  $0.3\frac{6}{3}$ ; of total SO42-; nss-K+ accounted for 98 ± 3%, 89 ± 9% and 99 ±  $0.3\frac{1}{2}$ , of total K+; nss-Ca2+ accounted for 95 ± 4%, 91 ± 10% and 96 ± 3%, respectively. In this study, % of total Ca2+. Thus, marked contributions of nss-emission sources to the chemical concentrations at all three clusters were found although the. However, these values were possibly may be underestimated, since total Na+ amount does concentrations do not necessarily originate from sea salt alone, but could partially come from dust and burning sources(Zhang et al., 2013). And (Zhang et al., 2013). In addition, the loss of Cl particles due to a chloride depletion mechanism further supported supports the underestimation of Cl-. The contributions of nss-sources were lower in cluster 2 than that in elusterclusters 1 and 3, which was attributed to the high

Field Code Changed

(3)

Field Code Changed

10

15

20

emissionemissions of sea spray resulting from coupled with high wind speed in cluster 2. Generally,

 $K^{\scriptscriptstyle +}$  is often used as a tracer for biomass burning. The high  $K^{\scriptscriptstyle +}$  concentration and the largest contribution of  $nss-K^+$  in cluster 3 indicated obviouslyclearly high emissions associated closely with agricultural burning in the Shandong Peninsula. The This finding agreed with the fact that Shandong is the largest producer of crop residues in the North China (Zhao et al., 2012) (Zhao et al., 2012), and biomass burning is one of thean important sourcesource of inorganic and organic aerosols in the Bohai sea atmosphere (Feng et al., 2012; Wang et al., 2014) (Feng et al., 2012; Wang et al., 2014). The contribution of nss-Mg2+ to total magnesium concentration was less than 4% for the all clusters, indicating the species came mostly from sea salt emission. The mass ratio of Mg2+ to Na+ was  $0.07 \pm 0.06$ ,  $0.06 \pm 0.03$  and  $0.06 \pm 0.03$  for clusters from 1 to 3, respectively. The ratio was ratios were less than 0.23, also demonstrating that Mg2+ mostly came from sea salt source (Zhang et al., 2013).

Enrichment Factor (EF) method with an abundance element associated with crustal elements is often used to assess enrichment characteristics of various elements in PM2.5. In the present study, Fe was used as the reference element for the assessment because other two major crustal elements (Si and Al) could not be measured due to the interference from the quartz fiber substrate of the samples. EF of 15 each element was calculated relative to the average crustal rock composition of Fe: EF = (X/Fe)Aerosol/(X/Fe)Crust-where (X/Fe)Aerosol- and (X/Fe)Crust- were the ratios of mean concentration of target elements and Fe in PM2.5 and continental crust, respectively. By convention, an EF value close to unity suggests that the element is dominantly contributed by natural soil dust or the contribution of anthropogenic sources is not significant, and the value much higher than 10 indicates that the 20 element is predominantly originated from human activities, such as combustion, automobile, industrial emissions, agricultural activities and etc. (Shah et al., 2012), An EF with a range of 1 to 10 indicates a hybrid contribution from crustal and anthropogenic sources. Here, except for the EF of Ni (9.17 ± 5.11) in cluster 3, the average EF of Ti, Ni, Cu, Zn, As, Cd and Pb in all clusters was greater than 10, showing severe influence from anthropogenic activity. The average EF in the three clusters also reflected that Co had a strong nature source, and the other elements (V and Mn) in all three

5

elusters and Ni in cluster 3 were supported from crustal dust and anthropogenic activities.(Zhang et al., 2013).

The ratios of OC/EC and NO3-/nss-SO42- were used as tracers to assess source signals of the

three clusters. It has been found that lowLow temperature burning, such as agricultural 5 residuesresidue burning, emits more OC compared with high temperature burning, e.g. vehicle exhaust. Thus, the ratio of OC to EC is often used to evaluate relative contribution of low and high temperature burning emission- (Zhao et al., 2012). The OC/EC ratios were  $1.41 \pm 0.30$ ,  $1.47 \pm 0.29$  and  $2.14 \pm 0.50$  for the-clusters from-1 to 3, respectively. Mean test showed that the ratios were insignificant difference between cluster 1 and cluster 2 ratios were insignificant at a 95% 10 confidence level, and theboth clusters 1 and 2 ratios of two groups were statistical lowstatistically lower compared with that of cluster 3, at the same confidence level. This suggested suggests that low temperature burning emission clearly contributed obviously to emission in cluster 3, while high temperature burning emission was more distinct in clusterclusters 1 and 2. BesidesFurthermore, mobile sources, such as vehicles, exhaust more  $NO_x$  than  $SO_2$ , while stationary sources, such as coal-fired power plants, emit more SO2 than NOx. The- (Wang et al., 2005). These two precursors 15 convert into  $\frac{\text{sulfate} SO_4^{2-}}{\text{and } \frac{\text{nitrate} NO_3^{-}}{\text{in the atmosphere, and the two type sources show different}}$ ratioratios of NO3-/SO42- from the two type sources. Therefore, the ... Hence, this ratio is often used as an indicator of the relative importance of mobile vs. stationary sources of sulfur and nitrogen in the atmosphere (Liu et al., 2014;Zhao et al., 2013)(Zhao et al., 2013; Liu et al., 2014). In this study, after deducting the contribution of sea salt to sulfate,  $SO_4^{2-}$ , the mean ratios of  $NO_3^{-7}$ /nss- $SO_4^{2-}$  were 0.96 ± 20  $0.31, 0.47 \pm 0.24$  and  $0.64 \pm 0.14$  for elusterclusters 1 to 3, respectively. Mean test showed that the three cluster ratios of the three clusters exhibit significant difference differences from each other at a 95% confidence level. The significant highhighest ratio in cluster 1 suggested suggests that amongst the three regions, mobile sources are the most important contribution of mobile source contributors of in the BTH region-among the three regions, followed by Shandong Peninsula (cluster 3). The ratio 25 of NO3-/nss-SO42- in cluster 1 was within the range of that those found in large cities, such as Beijing (1.20), Tianjin (0.73), and Shijiazhuang (0.76), the capital of Hebei province (Zhao et al., 2013)(Zhao et al., 2013), reflecting a hybrid contribution from the BTH region. The value in cluster 2 was slightly lower than that in winter in Chengde (0.55), one city located in the northern mountainous area of Hebei Province (Zhao et al., 2013). (Zhao et al., 2013). It indicated more Formatted: Don't add space between paragraphs of the same style, Line spacing: 1.5 lines

obvious contribution of stationary source emissions in the following region, areas such as the east area of eastern Inner Mongolia and the west part of Liaoning, than that infrom the BTH region and the Shandong Peninsula. These stationary source emissions are possibly associated possibly with coal combustion because of the lower ration of OC/EC ratio in cluster 2 than that incompared to cluster 3.

**5**

10

**3.3 Source apportionment of carbonaceous aerosols in PM2.5**

The cluster analysis clearly indicated that PM2.5 pollution on the Qimu Island wasconcentrations increased significantly influenced by the high concentrations of PM2.5-carried by the when air masses came from the BTH region and the Shandong Peninsula during the sampling period. The chemical species in PM2.5 from the BTH region hadpossessed more remarkedmarked signals of high temperature burning and mobile sources, while thatthose from the Shandong Peninsula had more obvious patterns of low temperature burning and stationary sources. To further confirm the sources of PM2.5, source apportionment of carbonaceous aerosols, as the main components of PM2.5, valuated by 14C measurement. In order to better achieve our purpose using a few samples for radiocarbon analysis due to the extensive cost of the analysis, two combined samples were collected nerfect synoptic process during the sampling period. The selection was reasonable as elaborated below. The synoptic process occurred during 16th and 18th, January, 2014. As shown in Fig. 2, the first half air masses of the synoptic process were derived from the south and passed through the Shandong Peninsula (cluster 3) and the bottom half were from the north and passed over the BTH region (cluster 1). Thus, two samples collected continually from 06:00 to 18:00, 16th January and from 18:00 to 06:00 the next day in the first half of the synoptic process were merged into one (M1) for the radiocarbon analysis. Similarly, other two samples collected continually from 17th January 18:00 to 6:00 and from 06:00 to 18:00 in the next day were combined into the other sample (M2). The M1 reflected the signal of air masses coming from the Shandong peninsula, while

18

M2 showed the pattern of air masses from the BTH region.

measurement will be linked with PMF result, the Considering that the representative capacity of all samples in the two clusters was examined thoroughly. It is expected that PMF can interpret those data close to the average condition of each chemical species since the method 5 minimizing error estimates to decompose a matrix of sample data into two matrixes under constraints for the factors (Paatero et 2014) to assess their ratio for different purposes. Therefore, OC and EC concentrations, and ratios of OC/PM2 5 and EC/PM2 5 of each sample were compared with that in the corresponding cluster by mean test. Results showed that except for a significant high ratio of EC/PM25, the OC and EC ntrations and the OC/PM25 ratio of M2 were insignificant difference with that in the cluster 1 10 at a 95% significant level, indicating its perfect representative capability for further carbonaceous analysis. However, the typical ability of M1 was slightly bad because only ratios of OC/PM2 5- and EC/PM2-5-were no statistical difference, OC and EC concentrations were significantly higher than significant level. Finally, the samples still 15 radiocarbon analysis in that they were from the most faultless synoptic process during the whole dramatic than that from two was more insular the insignificant difference of the ratios of OC/PM25 assured the validity for PM25

source assessment, which was more important than carbonaceous species in this study.-

Table 3 lists the concentrations and contemporary carbon fractions of OC, WSOC, WIOC and20EC of the two combined samples-, which were selected via a perfect synoptic process during the
sampling period. The fraction of OC was yielded by the average weightedweights of
concentration
concentrations of WSOC and WIOC fractions. It can be expressed as  $f_{e:}$

 $\frac{f_{c}(OC) = [f_{e}) = [f_{c}(WSOC) \times C_{\underline{C}}(WSOC) + f_{e}f_{c}(WIOC) \times C_{\underline{C}}(WIOC)] / [C_{\underline{C}}(WSOC) + C_{\underline{C}}(WSOC) + C_{\underline{C}}(WSOC)] / [C_{\underline{C}}(WSOC) + C_{\underline{C}}$

where  $f_e f_c(OC)$ ,  $f_e f_c(WSOC)$  and  $f_e f_c(WIOC)$  are the contemporary carbon fractions of OC, WSOC and WIOC, and  $\underline{C}_c(WSOC)$ -and  $\underline{C}_c(WIOC)$  are the concentrations of WSOC and WIOC, respectively. Generally, WSOC is mainly associated with biomass burning and secondary formation (Du et al., 2014)(Du et al., 2014), while OC directly emitted from the combustion of fossil fuel is mostly water insoluble (Weber et al., 2007).(Weber et al., 2007). During the earlier stage of the Formatted: Don't add space between paragraphs of the same style, Line spacing: 1.5 lines

| ormatted: Font: SimSun |
|------------------------|
|                        |

F

| Formatted: Indent: First line: 0 ch, Don't add |  |  |  |  |  |  |  |
|------------------------------------------------|--|--|--|--|--|--|--|
| space between paragraphs of the same style,    |  |  |  |  |  |  |  |
| Line spacing: 1.5 lines                        |  |  |  |  |  |  |  |
| Formatted: Font: SimSun                        |  |  |  |  |  |  |  |
| Formatted: Font: SimSun                        |  |  |  |  |  |  |  |

synoptic process, the concentrations of WSOC and WIOC were  $6.384 \ \mu g \ m^{-3}$  and  $6.323 \ \mu g \ m^{-3}$ , respectively. While Later on, the concentrations of the two carbonaceous fractions fell to  $3.687 \,\mu g \,m^{-3}$ and 5.303  $\mu$ g m-3, respectively, after the shift of the dominant wind direction from southerly to northwesterly, as shown in Fig. 2. The fraction of WSOC to OC decreased from 50% to 41% and the WIOC fraction increased from 50% to 59% before and after the shift of the dominant wind. It direction. This suggested that the contribution of fossil fuel combustion was more obvious in the BTH region than that-in the Shandong Peninsula, and vice versa. The implication could be further confirmed by the radiocarbon analysis of the two samples. The contemporary carbon fractions of WSOC and WIOC decreased from 0.59 to 0.49 and from 0.60 to 0.43, respectively, which indicated a decrease of the impact of biogenic and biomass burning emission, and an increase in contribution of the fossil fuel combustion to the two OC fractions after the shift of the prevailing wind. After the weighted average of the WSOC and WIOC fractions, the  $f_c(OC)$  values were 0.59 and 0.46 for the M1 and M2 samples, respectively. Together with  $f_{e} f_{c}$  (EC), it could be we determined that biogenic and biomass burning emission contributed 59% of OC and 52% of EC concentrations, respectively, when the air masses were from the Shandong Peninsula. After the change of wind fielddirection, the contribution of biogenic and biomass burning emission fell to 46% for OC and 38% for EC, respectively, which suggested that fossil fuel combustion contributed a dominant portion toof the carbonaceous aerosols from the BTH region.

20

25

15

5

10

The synoptic process showed clearly showed a shift of the dominant wind from southerly to4 northwesterly, namely from the Shandong Peninsula to the BTH region. In the meanwhileMeanwhile, the pattern of biogenic and biomass burning emission wasbecame more and more weak, and the signal of fossil fuel combustion wasbecame more and more obvious. If This was in agreement with that discussed above.our previous discussion. For instance, the emissionemissions in the BTH region exhibited more signals of high temperature burning and vehicle exhaustsexhaust. It was characterized by the lower ratio of OC/EC (1.41 ± 0.30), the higher ratio of NO3-//nss-SO42- (0.96 ± 0.31)), and the relatively lower concentration of nss-K+ compared with thatthose in the Shandong Peninsula (2.14 ± 0.50 for OC/EC ratio, 0.64 ± 0.14 for NO3-//nss-SO42- ratio). The contribution of the biogenic and biomass burning emission to the carbonaceous aerosolaerosols in the Shandong Peninsula was still significant, which has often been mentioned in previous studies (Feng et al., 2012; Zong et al., 2015; Wang et al., 2014), although there was great\_combustion of fossil fuel (e.g., coal) for not only

30

Formatted: Don't add space between paragraphs of the same style, Line spacing: 1.5 lines industrial activity but also heating in winter. The emission of biogenic and biomass burning in the Shandong Peninsula was often highlighted in previous studies (Feng et al., 2012;Zong et al., 2015;Wang et al., 2014).

3.34 Source apportionment of PM2.5

The EPA PMF 5.0 model and the data setswas used together with a date set of  $76 \times 22$  (76 samples with 22 species) were conducted to further quantitatively estimate the source contributioncontributions of PM2.5. After iterative testing from 5 to 15 factors for thein model scenariosexercises, we found the minimum deviation of the source apportionments of OC and EC between the results from radiocarbon14C measurement and a PMF model scenario with an Fpeak value of 0 and the lowest Q values. The comparison will be elaborated below. (6245).

10

15

20

25

5

Based on the PMF modeling results, eight main-source factors were identified, as shown in Fig. 3. Secondary inorganic aerosols, which were extensively identified in previous studies (Tao et al., 2014;Zhang et al., 2013), were apportioned into the primary source sectors in the present study. Because the results are more useful for guiding PM2.5-source controls. The contributions of the eight sources to PM2.5 were summarized in Table 4. The total and cluster fractional contributions (%) from each source were calculated based on the corresponding sample values simulated by the PMF

modeling.

3. Traffic emission has attracted considerable concern in the megacities of China (e.g., Beijing4 and Shanghai) due to the remarkable growth of vehiclesvehicle numbers in China. OnIn Beijing in 2012, on-road vehicles were estimated asto be the largest local emission source and contributed 22% of PM2.5, including primary and secondary fine particles<del>, but</del> and excluding vehicle-induced road dust<del>, in Beijing in 2012</del> (Zhang et al., 2014e).(Zhang et al., 2014b). The first source factor was characterized by high loadings of NO3-, SO42-, NH4+, OC, EC, Zn and Cu, which enclosesmatched a vehicle emission profile- (Zhang et al., 2013). Generally, nitrate, sulfate, NO3-, SO42-, OC and EC are mainly from engine exhaust emissionemissions, and ammonia is from vehicles equipped with three-way catalytic converters. Not only Zn and Cu, but also Pb and Cd are emitted directly bounded particles from exhaust (Zhang et al., 2014e).(Zhang et al., 2014b). In addition, the high NO3-/SO42-

ratiosratio of 1.28 calculated by the PMF results showed the pattern of suggested high temperature burning and vehicle emission. And the emissions. This source was the largest contributor forof NO3, which contributed 41% of the chemical species of PM2.5-during the sampling period. The contribution was higher than 31% of NOx emitted by traffic sectors in the-North China in 2003, which an expected an-increase of the contribution due to the rapid rise of vehicles in the North China in recent years (Shi et al., 2014b)(Shi et al., 2014b). This factor was the prevalent anthropogenic PM2.5 source in the-North China, with an average contribution of 16% during the sampling period. The contribution was lower than that in Beijing (Zhang et al., 2014c)(Zhang et al., 2014b), agreeing with the regional contribution characteristic in our study, rather than ones in large eitycities, where a large number of vehicles were runningrun. The second factor consisted of mineral dust elements, such as Mn, Fe and Co, and chemical species from human activities, such as Zn and EC, showing a mixed pattern of natural and anthropogenic emissions. Vehicle emission is an important source of atmospheric zineZn pollution because it can be emitted from direct exhaust, lubricating oil additives, tyretire and brake abrasion, wearing and corrosion from anticorrosion galvanized automobile sheet, and reentrainment re-entrainment dust enriched with zinc (Duan and Tan, 2013). Zn (Duan and Tan, 2013). Thus, the source factor was treated identified as traffic dust under the relative high contribution of vehicle emission to PM2.5 concentration.

The third source factor was ship emissions, typically characterized by high proportionproportions of Ni and V, and a high V/Ni ratio of V/Ni. High loading of thethese two metals is typically associated with the emissionemissions from residual oil, probably derived from shipping activities and some industrial processes (Pey et al., 2013). (Pey et al., 2013). In addition, a V/Ni ratio of more than 0.7 is always considered as a sign of PM2.5 influenced by shipping emissions (Zhang et al., 2014a)(Zhang et al., 2014a). The average ratiosratio of V/Ni from the measured data werewas  $0.93 \pm 0.46$ , indicating an obvious contribution of shipping emission. The average ratio of V/Ni calculated from the PMF source profile was 1.02, which was the second highhighest value among thatamongst those derived from the eight sources. While the The highest value of 1.29 was for the mineral dust source, which agreesagreed with a high ratio of 3.06 for soil background concentrations of the two metals in mainland China (Pan et al., 2013). (Pan et al., 2013).

5

10

15

20

The fourth factor showed high loadingloadings of Cu, Zn, As, Cd and Pb, which waswere4 treated as mainly contributed by signals of industrial processes. The emission Emissions from the iron

and steel industry are possibly important amongamongst those industrial processes based onfor two proofsreasons. One is that the sintering process in the iron and steel industries emits lotslarge amounts of Pb, Hg, Zn and other heavy metal pollutants, and other processes such as ironmaking and steelmaking also emit fugitive dust containing high concentration\_concentrations of heavy metals (Duan and Tan, 2013)(Duan and Tan, 2013). The other reason is the huge scale of steel production <del>of</del> steel\_in\_the North China. The\_nationalNational statistical data shows that China produced aboutapproximately half worldthe 
[revised manuscript text omitted]
 harmharmful to the environment. They, they still tend to use agricultural wastewastes as household fuel and remove them via burningburn wastes in field fields, mainly due to the low costcosts of the

29

30

25

methods. A more permanent solution is would be to find the high higher economic value of in agricultural wastes via development of renewable techniques. Indeed, agricultural wastewastes can be utilized to produce manifoldmany renewable energyenergies, such as biogas, feedstuffs, biochar, bioethanol, and bio-succinic acid, and so on. China has provided relevant energy regulations, 5 legislations legislation, and policy initiatives for rural renewable energy (Li et al., 2015).(Li et al., 2015). The government has also encouragesencouraged and sustainssustained the development of the renewable energy industry to increase the demand offor raw feedstocks. Through these efforts, China has achieved some success in renewable development in rural areas, but thethese efforts are not an effective solution to the problem of surplus crop waste, because the costcosts and the benefits 10 cannot yet be offset. For instance, Zhangziying, a town located in the easteastern area of the Daxing district of Beijing, has developed household biogas and straw gas since the 1980s, but thein 2011 renewable energy only occupied aboutmade up approximately 10% of household energy consumption, which was much lower than the fraction of coal (30%) in 2011(Li et al., 2015).(Li et al., 2015). Before the achievement of high economic value, except for the ban on crop straw burning, 15 the government should compensate farmers to collect crop residues as feedstocks of renewable energy, rather than burning in fieldfields or householdhouseholds (Shi et al., 2014a).(Shi et al., 2014a). The revenue from the subsidy and the sale of crop residues can support famer's could help alleviate economic burdens for theon farmers, so they can use of clean energy, such as electricity, liquefied petroleum gas, biogas-and so on, etc., for household consumption (Kung and Zhang, 20 2015)(Kung and Zhang, 2015). The These efforts will not only significantly improve air quality, but also make famers to-learn the convenient convenience of clean energy and wake from agricultural residue burning.\_\_

**5** Conclusions**

Source apportionment of wintertime PM2.5 at a background site in the North China in 2014 was 25 conducted by statistical analysis of the chemical species grouped according to the trajectory clusters, radiocarbon measurement of the carbonaceous species and the PMF modeling confirmed by the 14C analysis. During the sampling period, the mean-average  $PM_{2.5}$  concentration of  $PM_{2.5}$ -was 77.6 ± 59.3  $\mu$ g m3, and sulfate concentrations were SO42- concentration was the highest of any constituent, with a mean of  $14.2 \pm 18.0 \ \mu g \ m^{-3}$ , followed by mitrateNO3± (11.9 ± 16.4 \ \mu g \ m^{-3}), OC (6.85 ± 4.81 \ \mu g m-3), EC (4.90 ± 4.11 µg m-3), and ammonium  $NH_4^{\pm}$  (3.11 ± 2.14 µg m-3). The fractions of sulfate,

nitrateSO42-, NO3- and ammoniumNH4± to PM2.5 were obviously higher than thatthose in metropolismetropolises (e.g. Beijing and Tianjin) within the-North China, while fractions of carbonaceous species waswere markedly lower, which; these showed regional pollution signals.

More than half of air masses during the sampling period were from the BTH region, followed

by the air masses from the Mongolia (35%) and the Shandong Peninsula (11%) during the sampling period.

5

10

15

20

25

%). The concentrations of PM2.5 and its-most of the species carried by the air masses from the BTH region and the Shandong Peninsula were comparable (p > 0.05), and they wereoccurred in statistically highergreater concentrations than thatthose carried by the air masses from the-Mongolia (p < 0.01). The patterns were attributed to the spatial distributions of their emissions and the high wind speed when air masses were from the Mongolia. The PM2.5 had an obvious signal of biomass burning emission, characterized by a high OC/EC ratio, low NO3-/nss-SO42- ratio and high nss-K+ concentration7 when air masses came from the Shandong Peninsula. WhileIn contrast, the PM2.5 carried from the BTH region showed vehicle emission pattern, characterized by low OC/EC ratio, high NO3-/nss-SO42- ratio and low nss-K+ concentration. TheThis finding was confirmed by the radiocarbon14C 
[revised manuscript text omitted]

10.1016/j.scitotenv.2015.01.113, 2015.

15

10

5

| Species                             | Mean $\pm$ std. Range                        |                               | Spacios                | $Mean \pm std$                 | Range                         |
|-------------------------------------|----------------------------------------------|-------------------------------|------------------------|--------------------------------|-------------------------------|
|                                     | $(\mu g m^{-3})$                             | (µg m -3 )         | species                | $(ng m^{-3})$                  | $(ng m^{-3})$                 |
|                                     |                                              |                               |                        | 408 ±                   | <del>0.01</del> —             |
| PM 2.5                   | $77.6\pm59.3$                                | 12.7 - 305                    | <del>TiFe</del> | 2857.72 ±               | <del>30.</del> 7 .12 – |
|                                     |                                              |                               |                        | 7.34                           | 1588                   |
|                                     | 4.90 ±                                       | <del>0.80</del> —             |                        | 107 ±                   | <del>0.45</del> —             |
| ECSO42-                             | 4 .11 14.2 ±                          | <del>19.61.37 –</del>  | ¥Zn             | 142 3.90 ±              | <del>12.</del> 5 .56 – |
|                                     | 18.0                                  | 96.2                   |                        | <del>2.47</del>                | 987                    |
|                                     | 6 95 + 11 0 +                                | 0. <mark>81</mark> —          |                        | 88.4 ±                  | 3.02 –                 |
| OCNO3                               | $\frac{0.03 \pm 11.9 \pm}{16.4.91}$          | <del>21.327 –</del>    | MnPb                   | 85.7 29.3 ±             | 412 1.38               |
|                                     | 10. 4 <del>.01</del>                  | 87.1                   |                        | <del>28.0</del>                | <del>108</del>                |
|                                     | 2.11 + 2.06 +                                | 0 (1 10                       |                        | 29.3 ±                  | 1.38 –                 |
| $\underline{\text{CFNH}}_{4}^{\pm}$ | $5.11 \pm 2.00 \pm 1.7814$                   | 0. 01 – 10––           | Fe Mn           | 28.0 408 ±              | 108 7.12—              |
|                                     | <del>1.7014</del>                     | <del>8.90.1</del>      |                        | <del>285</del>                 | <del>1588</del>               |
|                                     | <del>11.9 ±</del>                            | 0. <del>27</del> —            |                        | 9.08 ±                  | 0. <del>01</del> —            |
| NO 3 Cl                  | <del>16.42.06 ± 87.110 -</del> |                               | CoCu                   | 11.4 0.24 ±             | <del>0.7303 –</del>    |
|                                     | 1.78                                  | 8.90                   |                        | <del>0.18</del>                | 77.7                   |
|                                     | $14.2 \pm 18.0.06$                           | <del>1.37</del>               |                        | 7.72 ±                  | 0.01 –                 |
| $\frac{SO_4^2}{K^+}$                | +0.84                                        | <del>96.2</del> 0.07 – | Ni Ti           | 7.34 4 <del>.28 ±</del> | 30.7 1.68              |
|                                     | - 0.84                                | 3.95                   |                        | <del>2.30</del>                | <del>13.8</del>               |
|                                     | $0.43 \pm 0.25$                              |                               |                        | 6.61 ±                  | 0. <del>03</del> —            |
| $Na^+$                              |                                              | 0.05 - 1.58                   | CuAs                   | 7.86 9.08 ±             | <del>77.767 –</del>    |
|                                     |                                              |                               |                        | <del>11.4</del>                | 43.4                   |
|                                     | 0.38 ±                                | 0 61 10.07                    |                        | 4.28 ±                  | 1.68 –                 |
| $NH_4Ca^{2+}$                       | 0.22 <del>3.11 ±</del>                | 1 32                          | ZnNi            | 2.30 <del>107 ±</del>   | 13.8<mark>5.56</mark>  |
|                                     | <del>2.14</del>                              |                        |                        | <del>142</del>                 | <del>987</del>                |
|                                     | $0.9603 \pm$                                 | 0. <del>07</del> —            |                        | 3.90 ±                  | 0. <del>67</del> —            |
| $\frac{KMg^{2+}}{Mg^{2+}}$          | 0.90 03 -                      | <del>3.9501 –</del>    | AsV             | 2.47<del>6.61 ±</del>   | 43.445 –               |
|                                     | 0.04 05                               | 0.17                   |                        | <del>7.86</del>                | 12.5                   |
|                                     | <del>0.03 ±</del>                            | 0. <del>01 —</del>            |                        | 1 82 +                         |                               |
| Mg 2+ OC                 | <del>0.036.85 ±</del>                 | + 0.17 81 –     | Cd                     | 4.06                           | 0.04 - 25.9                   |
|                                     | 4.81                                  | 21.3                   |                        | 1.00                           |                               |
|                                     | 4.90 ±                                | 0. <del>07 —</del>            |                        | <del>88.4 ±</del>              | 0.01 –                 |
| <del>Ca2+EC</del>        | 4.11 0.38 ±                           | 1.3280 –               | PbCo                   | <del>85.70.24 ±</del>   | 0.73 3.02              |
|                                     | <del>0.22</del>                              | 19.6                   |                        | 0.18                    | <del>412</del>                |

Table 1. Statistics of  $PM_{2.5}$  chemical components on the Qimu Island during the sampling period

| Species                                                | Mean ±                                                       | standard deviation             | Significant level              |      |      |      |
|--------------------------------------------------------|--------------------------------------------------------------|--------------------------------|--------------------------------|------|------|------|
| (unit)                                                 | it) Cluster1(n=42) Cluster2(n=25)                            |                                | Cluster3(n=9)                  | 1&2  | 1&3  | 2&3  |
| PM 2.5
(μg m -3 )             | $93.0 \pm 66.1 \\ (24.5 - 305)$                              | 41.6 ±26.7
(12.7–143)       | 106±42.3
(50.3–193)         | 0.00 | 0.59 | 0.00 |
| EC
(μg m -3 )                            | $6.53 \pm 4.66$
(1.39–19.6)                               | 2.50 ±1.84
(0.80-8.85)      | 3.94±1.49
(2.53–7.66)       | 0.00 | 0.11 | 0.05 |
| OC
(μg m -3 )                            | $8.58 \pm 5.23$
(1.45–21.3)                               | 3.51 ±2.35
(0.81–11.4)      | 8.04±2.32
(5.25–13.5)       | 0.00 | 0.76 | 0.00 |
| Cl -
(µg m -3 )               | 2.37 ± 2.11
(0.10–8.90)                                   | 1.22 ±0.65
(0.20–2.85)      | 2.94±1.35
(1.42-5.53)       | 0.01 | 0.45 | 0.00 |
| $NO_{3}^{-1}$
(µg m -3 )                 | $17.6 \pm 19.6$
(1.75-87.0)                               | 2.75 ±4.25
(0.27-20.1)      | 10.6±6.09
(4.41–20.3)       | 0.00 | 0.30 | 0.00 |
| SO 4 2-
(µg m -3 ) | $19.4 \pm 21.8$
(2.09–96.2)                               | 4.55 ±4.06
(1.37–19.5)      | 16.4±8.74
(5.34–35.6)       | 0.00 | 0.69 | 0.00 |
| Na +
(μg m -3 )               | $\begin{array}{c} 0.38 \pm 0.24 \\ (0.05  1.58) \end{array}$ | 0.55 ±0.26
(0.18-1.08)      | 0.31±0.06
(0.22–0.40)       | 0.01 | 0.41 | 0.01 |
| $NH_4^+$
(µg m -3 )                      | $3.97 \pm 2.29$
(1.28–10.1)                               | 1.53 ±0.98
(0.61-4.70)      | 3.52±0.96
(1.93-4.90)       | 0.00 | 0.57 | 0.00 |
| $K^{+}$
(µg m -3 )                       | $1.11 \pm 0.74$
(0.28–3.10)                               | $0.35 \pm 0.36$
(0.07-1.69) | 2.01±0.93
(0.78–3.95)       | 0.00 | 0.00 | 0.00 |
| $Mg^{2+}$
(µg m -3 )                     | $0.03 \pm 0.03$
(0.01–0.17)                               | 0.03 ±0.02
(0.01-0.11)      | 0.02±0.01
(0.01-0.04)       | 0.66 | 0.41 | 0.13 |
| $Ca^{2+}$
(µg m -3 )                     | $0.37 \pm 0.22$
(0.11-1.32)                               | 0.37 ±0.18
(0.07-0.74)      | 0.44±0.29
(0.09–0.97)       | 1.00 | 0.46 | 0.46 |
| Ti                                                     | $6.96 \pm 5.98$
(0.35-25.9)                               | $10.9 \pm 9.10$
(0.01-30.7) | 2.51±0.85
(1.16-3.58)       | 0.04 | 0.03 | 0.01 |
| V (ng m -3 )                                | $4.68 \pm 2.29$
(0.76-11.3)                               | $2.83 \pm 2.55$
(0.45-12.4) | $3.24 \pm 1.50$
(2.05-7.12) | 0.00 | 0.08 | 0.66 |
| Mn                                                     | $33.8 \pm 31.3$                                              | $17.6 \pm 19.3$                | 40.9±20.3                      | 0.02 | 0.53 | 0.01 |
| (ng m - )
Fe                             | (1.97 - 108)
$404 \pm 308$                                | (1.38–95.4)
375 ±263        | (9.14-69.8)
521±188         | 0.70 | 0.29 | 0.15 |
| (ng m -3 )
Co                            | (7.12-1588)
$0.26 \pm 0.20$                               | (9.13–826)
0.17 ±0.14       | (244–960)
0.36±0.13         | 0.00 | 0.14 | 0.00 |
| (ng m -3 )
Ni                            | (0.01-0.73)
$4.85 \pm 2.56$                               | (0.01–0.48)
3.51 ±1.85      | (0.10-0.59)
3.80±1.02       | 0.00 | 0.14 | 0.00 |
| $(ng m^{-3})$                                          | (1.68-13.8)
11.6 + 13.6                                   | (1.68-6.79)
3.06 +2.93      | (2.45–5.84)
13.9+7.05       | 0.03 | 0.24 | 0.67 |
| $(ng m^{-3})$                                          | (0.72-77.7)                                                  | (0.03-8.99)                    | (3.90-26.4)                    | 0.00 | 0.64 | 0.00 |
| Zn (ng m -3 )                               | (9.92-987)                                                   | (5.56-208)                     | (24.2–201)                     | 0.01 | 0.36 | 0.03 |
| As
(ng m -3 )                            | $9.03 \pm 9.52$
(1.11-43.4)                               | $3.00 \pm 2.82$
(0.67–14.0) | 5.35±3.35
(2.25–13.6)       | 0.00 | 0.27 | 0.06 |
| Cd
(ng m -3 )                            | $2.70 \pm 5.26 \\ (0.11 - 25.9)$                             | $0.45 \pm 0.41$
(0.04-1.29) | $1.54\pm0.65$
(0.49-2.66)   | 0.04 | 0.52 | 0.00 |
| Pb
(ng m -3 )                            | $110 \pm 95.3$
(5.30-412)                                 | 36.9 ±44.8
(3.02–176)       | 128±53.2
(45.4–215)         | 0.00 | 0.59 | 0.00 |

Table 2. Statistics of  $PM_{2.5}$  chemical species in different clusters and significant level by mean test

|      | Formatted: Add space between paragraphs of the same style, Don't snap to grid |
|------|-------------------------------------------------------------------------------|
|      | Formatted: Add space between paragraphs of the same style, Don't snap to grid |
|      | Formatted: Add space between paragraphs of the same style, Don't snap to grid |
| • \) | Formatted Table                                                               |
|      | Formatted: Add space between paragraphs of the same style, Don't snap to grid |
| •    | Formatted: Add space between paragraphs of the same style, Don't snap to grid |
|      | Formatted: Add space between paragraphs of the same style, Don't snap to grid |
|      | Formatted: Add space between paragraphs of the same style, Don't snap to grid |
|      | Formatted: Add space between paragraphs of the same style, Don't snap to grid |
|      | Formatted: Add space between paragraphs of the same style, Don't snap to grid |
|      | Formatted: Add space between paragraphs of the same style, Don't snap to grid |
|      | Formatted: Add space between paragraphs of the same style, Don't snap to grid |
|      | Formatted: Add space between paragraphs of the same style, Don't snap to grid |
|      | Formatted: Add space between paragraphs of the same style, Don't snap to grid |
|      | Formatted: Add space between paragraphs of the same style, Don't snap to grid |
|      | Formatted: Add space between paragraphs of the same style, Don't snap to grid |
|      | Formatted: Add space between paragraphs of the same style                     |
|      | Formatted: Add space between paragraphs of the same style                     |
|      | Formatted: Add space between paragraphs of the same style, Don't snap to grid |
| •    | Formatted: Add space between paragraphs of the same style, Don't snap to grid |
| •    | Formatted: Add space between paragraphs of the same style, Don't snap to grid |
| •    | Formatted: Add space between paragraphs of the same style, Don't snap to grid |
|      | Formatted: Add space between paragraphs of the same style, Don't snap to grid |
| •    | Formatted: Add space between paragraphs of the same style, Don't snap to grid |
|      | Formatted: Add space between paragraphs of the same style, Don't snap to grid |

4 •

|                          |                          | 1 2                        |                               | 1                 |                   |
|--------------------------|--------------------------|----------------------------|-------------------------------|-------------------|-------------------|
|                          | M1                       | M2                         |                               | M1                | M2                |
| $PM_{2.5}(\mu g m^{-3})$ | 159 .2 ± 0.5      | 91.8 ±0.5           |                               |                   |                   |
| OC (ug m -3 ) | 12.7 ±0.7         | 8.989.0 ±           | $f_{c} f_{\underline{c}}(OC)$ | 0.59 ±0.04 | 0.46 ±0.04 |
| ΟC_(μg III )             |                          | 0.5                 |                               |                   |                   |
| WSOC ( $\mu g m^{-3}$ )  | 6. <del>38</del> 4 + 0.4 | 3. <del>687 ±</del> | $f_{e}$ $f_{e}$ (WSOC)        | $0.59 \pm 0.03$   | $0.49 \pm 0.03$   |
|                          |                          | 0.2                 | -  (                   |                   | •••• -•••  |
| WIOC ( $\mu g m^{-3}$ )  | 6. <del>32</del> 3 ± 0.6 | 5. <del>303 ±</del> | $f_{c}f_{c}$ (WIOC)           | 0.60 ±0.03 | 0.43 ±0.03 |
|                          |                          | 0.4                 |                               |                   |                   |
| $EC_{\mu g} m^{-3}$      | 8. <del>58</del> 6 ± 0.5 | 3. <del>818 ±</del> | $f_{e} f_{c}$ (EC)            | 0.52 ±0.02 | 0.38 ±0.01 |
|                          |                          | 0.3                 |                               |                   |                   |

**Table 3. Concentration and contemporary carbon fraction of carbonaceous species in M1 and M2**

Formatted Table

**Table 4. Averages of fractional contributions (%) from eight sources identified by PMF model**

|          | Vehicle                   | Traffic | Ship     | Industrial | Biomass                     | Mineral                     | Coal                        | See celt                    |
|----------|---------------------------|---------|----------|------------|-----------------------------|-----------------------------|-----------------------------|-----------------------------|
|          | emission                  | dust    | emission | process    | burning                     | dust                        | combustion                  | Sea san                     |
| All      | 15. <del>883</del> | 4.24    | 8.95     | 2.64       | 19. <mark>331</mark> | 12. <mark>881</mark> | 29. <del>664</del>   | 6.58                        |
| Cluster1 | 23. <del>664</del> | 4.89    | 8.79     | 3.64       | 19. <del>661</del>   | 6.32                        | 29. <del>216</del>   | 3.96                        |
| Cluster2 | 3.57                      | 3.60    | 9.34     | 1.19       | 4.88                        | 26. <mark>881</mark> | 37. <mark>766</mark> | <del>13.012.95</del> |
| Cluster3 | 12. 546     | 3.08    | 8.67     | 1.96       | 52. <mark>767</mark> | 6.46                        | 12.4 38              | 2.32                        |

Formatted Table

---

## Referee Report (RR1)

**Review of: "Radiocarbon and PMF based source apportionment of PM2.5 at a regional background site in North China: insight into the contribution of biomass burning"**

**General comment:**

The manuscript has undergone significant improvement in terms of written language and clarity. The manuscript does have a potential to be a very good paper as the authors have collected a large amount of data and provided extensive analysis of sources of PM2.5 using both measurements and modeling. However, there are still major problems left in the manuscript to be addressed. In their response to the comments from the first round of review, often times they have not provided any new information that was not already in the manuscript to really address the referee's comments. There are still typos throughout the manuscript and some major references missing.

In my opinion, the main problem with the paper is the accuracy and interpretation of the radiocarbon measurements. The authors make conclusions about the sources of carbonaceous aerosols from large regions of China based on only two samples from different air masses. I don't think that one sample per cluster is representative of what the sources are in a particular region. Further the authors do not report uncertainty in their source apportionment estimates, which makes it very hard to interpret the accuracy of these measurements. The title of the manuscript suggests that radiocarbon was a major part of the source apportionment study, and I don't think that is fair to say that based on what they report in the manuscript. I cannot comment on the PMF interpretation of the 14C and PMF results in general, however the 14C alone is not enough to interpret the sources.

Further the manuscript lacks a comprehensive discussion of the results. Maybe some part of the results can be moved into the discussion, but even so a more comprehensive interpretation of the results is necessary. The conclusions only provide a summary of the results, but not actual conclusions.

**Minor comments:**

**Abstract**
-Typos
-No errors on 14C source app.

**Introduction**

- Paragraph I:
        -references for climate, health etc.

-Paragraph II:
- Line 30 page 3 : "while its contemporary level in non-fossil carbon sources is relatively constant." This is not quite the case as the Bomb Spike provides a high resolution interpretation of the access 14C in the atmosphere since the 1950s, which has been steadily declining.

**Methods**

**-** Sampling site and sampling collection
- Equilibrium process – Can the sample absorbe VOCs during this time? How do you account for that?

-Chemical analysis
- Page 6 line 13 – M1 and M2 not defined.
-Page 7 line 6: why is the conversion to fc different between OC and EC and where did you get these numbers? Either a citation is missing or more detailed explanation is needed.

- Principle for selecting 14C samples – this section is hard to understand
  o Don't know what a perfect synoptic process means, don't know what the accuracy of the model is and that is why using one samples per cluster doesn't seem reasonable

**Results**

- General characteristics and chemical composition
  o Page 9 line 17 : Is this avr ± st. dev and how is the uncertainty in the pm2.5 measurement accounted for? 77-59 = 16 ug/m3  if we take the min and apply 20 ug measurement error... how does this 20ug compare to ug/m3?
  o Also, lacking consistency in significant digits
  o It would be really nice if the section describing the concentrations and percentage in PM2.5 of  the different chemical components is summed into a figure e.g. a pie chart so that the reader can better visualize the relative contributions. Very difficult to follow
  o Page 10 line 4: citations
  o In general the names of the chemical species should be written out first, so the chemical formula is defined. This is missing throughout the manuscript.
  o Last paragraph belongs to discussion
- Cluster Analysis
  o Very difficult to follow which cluster corresponds to which region.
  o "Low temperature burning, such as agricultural residue burning, emits more OC compared with high temperature burning, e.g. vehicle exhaust." Citation

- 14C source app
  - No errors reported throughout the second paragraph. It is unclear what the uncertainty in the measurement is, which makes it not clear how reliable the measurement actually is.
  - P. 14 line 27 typo
  - Last paragraph should be in discussion
- PMF analysis
  - Generally – many repeating citations from one authors. A lot of major citations are missing.

**Implications for alleviation (lack of discussion)**
While this section presents an interesting discussion on the role of biomass burning, it is only one part of a discussion section. The manuscript lacks a comprehensive discussion of the results and their importance.
- P. 19 line 25/6 – unclear
- Typos/ language
- Discussion on Shandong Peninsula p 20 line 6-14 – citations?

**Figures and Tables**
The tables of the manuscript are significantly better than the first version and are much easier to understand now. Some of the figures however need improvement. Figure 2 is cut at the bottom and the font of figure 4 is too small to be read. The manuscript can definitely benefit from the addition of figures summarizing the results.

---

## Referee Report (RR2)

**Review of:**
**Source apportionment of PM2.5 at a regional background site in North China using PMF linked with radiocarbon analysis: Insight into the contribution of biomass burning**

The authors have done a great job in adding the requested information and clarifying the manuscript. As a result, it is has once again undergone even more improvements than after the first round of revisions. However, I am still not convinced that the radiocarbon discussion is clear and reliable. The reason for that is that major conclusions are drawn for biomass burning from region represented by sample M1, which in the text itself is described as "not ideal" (p. 9 l. 21 ). Further, radiocarbon samples on the order of a few hundred micrograms are very small and the C isotopes very difficult to measure. Therefore source apportionment results using 14C usually have a much higher uncertainty than 1 – 3%, which is reported here. Instrumental uncertainties and the AMS capacity are still not clear. I think that the idea of the authors and the way they use 14C to verify the PMF results is correct, however the 14C results are suggestive rather than conclusive and I think the level of uncertainty is highly underestimated. If however this was made clear in the manuscript, it would be much easier to understand and interpret the results.

Also, despite requesting for more citation during the last two rounds of reviews, there are still citations missing for e.g. in section 3.4.  Implication for PM alleviation and also there are still a few typos throughout the manuscript. Overall the manuscript is interesting and informative, but there are still problems that need to be addressed.

---

## Author Response (AR2)

**Response to Reviewers' Comments and Suggestions - doi: 10.5194/acp-2016-97**

Comments to the Author:

The revised manuscript is an improvement over the original. However, one of the referees still finds major problems with the methods, conclusions and the manuscript presentation. Further, the referee does not think the authors have adequately addressed previous comments made by the referee.

The authors must seriously address this referee's concerns in their next revision. Also, please have the manuscript edited by a native English speaker before resubmitting.

10 Dear Editor,

15

35

We are thankful very much to you and the anonymous reviewers for the thoughtful comments and suggestions. We have revised this manuscript accordingly and asked a native English speaker to edit it. Listed below are our point-by-point responses (blue) to the reviewer's comments (black). In addition, following review's opinion, the title of this manuscript has been changed as "Source apportionment of PM2.5 at a regional background site in North China using PMF linked with radiocarbon analysis: Insight into the contribution of biomass burning".

Best regards, Dr. Chongguo Tian

- 20 The manuscript has undergone significant improvement in terms of written language and clarity. The manuscript dose have a potential to be a good paper as the authors have collected a large amount of data and provided extensive analysis of sources of  $PM_{2.5}$  using both measurement and modelling. However, there are still major problem left in the manuscript to be addressed. In their response to the comment from the first round of review, often times they have not provided any new information that was not already in the manuscript to really addressed
- 25 the referee's comments. There are still typos throughout the manuscript and some major reference missing. Response and Revisions: Thanks very much for these significant comments. We have revised this manuscript accordingly. In addition, typos throughout the manuscript have been corrected, and the missing references have also been added in the manuscript. Please find our detailed responses blow.

In my opinion, the main problem with the paper is the accuracy and interpretation of the radiocarbon measurement. 30 The authors make conclusions about the sources of carbonaceous aerosols from large regions of China based on only two samples from different air masses. I don't think that one sample per cluster is representative of what the

sources are in a particular region.

Response and Revisions: We appreciate the reviewer's thoughtful comment. We agree with the reviewer's opinion that more 14C measurement could provide more accurate information of sources. In this study, two combined background samples (M1 and M2) including four independent samples from two major air masses were selected for

14C measurement. Few samples were conducted mainly due to the busy schedule of the instruments at that time and

the expensive cost of the measurement. To be honest, new  ${}^{14}C$  measurement could not be added in the manuscript during the refereeing period. However, we will continue to conduct more  ${}^{14}C$  measurement for further confirmation of sources in future study following the reviewer' opinion.

We have done much prepare work for 14C measurement using few samples. For example, before 14C measurement,
the representative capacity of all samples in two clusters (cluster 1 and 3) was examined thoroughly. OC and EC concentration, ratios of OC/PM2.5 and EC/PM2.5 of each sample were compared with that in the corresponding cluster by mean test. Finally, independent samples with great representativeness were selected carefully for 14C measurement. One goal in this study was to explore the source signals in different region in North China based on cluster analysis. So the concentration, area passed through, etc of independent samples were considered greatly when picking them. Besides, 14C measurement was not the only indictor for source type, cluster analysis based on OC/EC, K+, and NO3-/nss-SO42-, etc., also indicated the same source information with 14C measurement, which demonstrates greatly the source identification in our study.

Further the authors do not report uncertainty in their source apportionment estimates, which makes it very hard to interpret the accuracy of those measurements. The title of the manuscript suggests that radiocarbon was a major port of the source apportionment study, and I don't think that is fair to say that based on what they report in the manuscript. I cannot comment on the PMF interpretation of the 14C and PMF results in general, however, the 14C along is not enough to interpret the sources.

Response and Revisions: Thanks for the comment. Following the reviewer's opinion, we have added uncertainty to the 14C measurement utilizing the Error Propagation Formula (Liu et al., 2016):  $\delta f_c = \operatorname{sqrt} (\delta f_m^2 + \delta a^2)$ , where a is

- 20 the conversion coefficient caused by nuclear-bomb in the 1950s and 1960s, and δa is adopted as 0.05 according to a previous study(Zhang et al., 2014). The 14C measurement uncertainty result has been inserted in the revised manuscript (page 14, line 25-30; page 15, line 1-6). On the other hand, uncertainty of PMF model was usually estimated by bootstrapping (BS), displacement (DISP), and bootstrapping with displacement (BS-DISP), which has also been reported in the revised supporting information (page 15, line 27). As shown in table 1, the percentage of BS factors assigned to each base case factor ranges from a low value of 93% for sea salt to a high of 100% for vehicle emission, traffic emission, industrial process, mineral dust and coal combustion; and there are no unmapped BS factors. About DISP, after strong-weighted species were displaced, no factors swaps were reported for any of
- the allowed  $dQ_{max}$  examined by the model (fixed 4, 8, 16. 32 in this study). Besides, only two and one factors swaps were found for sea salt and biomass burning, respectively, for each allowed  $dQ_{max}$  examined by modelling (fixed
- 30

estimates results were well within the range of stable solution of PMF model, demonstrating the effectiveness of the model results in this study.

| Boot   | Vehicle  | Traffic | Ship     | Industrial | Biomass | Mineral | Coal       | Sea  | Unmannad |
|--------|----------|---------|----------|------------|---------|---------|------------|------|----------|
| Factor | emission | dust    | emission | process    | burning | dust    | combustion | salt | Unnapped |
| 1      | 100      | 0       | 0        | 0          | 0       | 0       | 0          | 0    | 0        |
| 2      | 0        | 100     | 0        | 0          | 0       | 0       | 0          | 0    | 0        |
| 3      | 0        | 0       | 96       | 0          | 4       | 0       | 0          | 0    | 0        |
| 4      | 0        | 0       | 0        | 100        | 0       | 0       | 0          | 0    | 0        |
| 5      | 0        | 0       | 0        | 0          | 99      | 0       | 0          | 1    | 0        |
| 6      | 0        | 0       | 0        | 0          | 0       | 100     | 0          | 0    | 0        |
| 7      | 0        | 0       | 0        | 0          | 0       | 0       | 100        | 0    | 0        |
| 8      | 0        | 0       | 0        | 0          | 7       | 0       | 0          | 93   | 0        |

Table 1 Percentage of BS factors assigned to each base case factor with a correlation threshold of 0.6.

- In addition, 14C measurement along may not be enough to interpret the sources with PMF model, however, characteristics of various PM2.5 species (OC, EC, ions, mental elements, etc) were discussed in detail in this study, also providing significant source information. For example, 14C measurement indicated PM2.5 from the Beijing-Tianjin-Hebei region exhibited more signals of fossil fuel, this was further confirmed by its lower ratio of OC/EC, higher ratio of NO3-/nss-SO42-, and lower concentration of nss-K+ compared with those from Shandong peninsula; The eighth factor with high loadings of Na+, Mg2+, CI was identified as sea salt by PMF model, also confirmed by that CI/Na+, Mg2+/Na+ in this factor were 1.79 and 0.11 (similar to the corresponding ratio in sea water 1.80, 0.12). It should be noted that results given by composition analysis, 14C measurement, PMF model usually exhibited well consistency in this study, implying accuracy of source apportionment in this study. Following the review's suggestion, the title of the manuscript was changed as "Source apportionment of PM2.5 at a regional background site in North China using PMF linked with radiocarbon analysis: Insight into the contribution
  - of biomass burning".

Further the manuscript lacks a comprehensive discussion of the results. Maybe some part of the results can be moved into the discussion, but even so a more comprehensive interpretation of the results is necessary. The conclusions only provide a summary of the results, but not actual conclusion.

20 Response and Revisions: Thanks for the comment. The present manuscript was prepared originally for a merged section of result and discussion, which is allowed by ACP as some example articles below. We still keep the structure of the manuscript because of huge workload and potential indigestibility of the separation of two parts. Following the review's suggestion, more interpretation has been added in the revised manuscript. The section of conclusion has been changed as summary and conclusion.

Khan, M. F., Latif, M. T., Saw, W. H., Amil, N., Nadzir, M. S. M., Sahani, M., Tahir, N. M., and Chung, J. X.: Fine particulate matter in the tropical environment: monsoonal effects, source apportionment, and health risk assessment, Atmos. Chem. Phys., 16, 597-617, doi: 10.5194/acp-16-597-2016, 2016.

Kong, S. F., Li, L., Li, X. X., Yin, Y., Chen, K., Liu, D. T., Yuan, L., Zhang, Y. J., Shan, Y. P., and Ji, Y. Q.: The

- 5 impacts of firework burning at the Chinese Spring Festival on air quality: insights of tracers, source evolution and aging processes, Atmos. Chem. Phys., 15, 2167-2184, doi: 10.5194/acp-15-2167-2015, 2015.
  - Zhang, Y. L., Huang, R. J., El Haddad, I., Ho, K. F., Cao, J. J., Han, Y., Zotter, P., Bozzetti, C., Daellenbach, K. R., Canonaco, F., Slowik, J. G., Salazar, G., Schwikowski, M., Schnelle-Kreis, J., Abbaszade, G., Zimmermann, R., Baltensperger, U., Prévôt, A. S. H., and Szidat, S.: Fossil vs. non-fossil sources of fine carbonaceous aerosols in
- 10 four Chinese cities during the extreme winter haze episode of 2013, Atmos. Chem. Phys., 15, 1299-1312, doi: 10.5194/acp-15-1299-2015, 2015.

Zhao, X. J., Zhao, P. S., Xu, J., Meng, W., Pu, W. W., Dong, F., He, D., and Shi, Q. F.: Analysis of a winter regional haze event and its formation mechanism in the North China Plain, Atmos. Chem. Phys., 13, 5685-5696, doi: 10.5194/acp-13-5685-2013, 2013.

15 Zhao, P. S., Dong, F., He, D., Zhao, X. J., Zhang, X. L., Zhang, W. Z., Yao, Q., and Liu, H. Y.: Characteristics of concentrations and chemical compositions for PM2.5 in the region of Beijing, Tianjin, and Hebei, China, Atmos. Chem. Phys., 13, 4631-4644, doi: 10.5194/acp-13-4631-2013, 2013.

Minor comments

20 Abstract

-Typos

Response and Revisions: Thanks for the comment. The typos have been corrected in the revised manuscript.

- No errors on 14C source app

Response and Revisions: We appreciate the reviewer's comment. The 14C errors calculated by the Error

25 Propagation Formula (Liu et al., 2016) have been added in the revised manuscript (page 2, line 12-15).

Introduction

- Paragraph I

-References for climate, health etc

Response and Revisions: Thanks for the comment. More references (Chen et al., 2013; Lu et al., 2015; Pui et al.,

30 2014; Tao et al., 2014) about fine particles affecting climate, health etc have been added in the revised manuscript

(page 3, line 5).

- Paragraph II

- Line 3 page 3: "while its contemporary level in non-fossil carbon sources is relatively constant." This is not quite the case as the Bomb Spike provided a high resolution interpretation of the access 14C in the atmosphere since

5 1950s, which has been steadily declining.

Response and Revisions: We appreciate the reviewer's comment. Following the reviewer's suggestion, we have revised this sentence "The underlying principle of 14C measurement is that radioisotope carbon has become extinct in fossil fuel due to its age (half-life 5730 years), while non-fossil carbon sources contain the contemporary or near contemporary radiocarbon level" (page 3, line 29).

10 Methods

- Sampling site and Sampling collection

- Equilibrium process - Can the samples absorb VOCs during this time? How do you account for that?

Response and Revisions: Thanks for the comment. In this study, blank samples were conducted accompanying sampling samples throughout the whole process. Concentrations of OC in blank samples were all < 3.5% of the

15 average concentration for the total samples (page 5, line 20), implying our samples were not contaminated during the processes, such as transportation, equilibrium and chemical analysis. Thus, we can see that samples didn't absorb VOCs or absorb little of VOCs during the equilibrium process, which wouldn't interfere the accuracy of results in this study.

- Chemical analysis

25

20 - Page 6 line 13 – M1 and M2 not defined

Response and Revisions: Thanks for the comment. Following the reviewer's opinion, we add sentences "Two combined samples reflecting source signals from the Shandong Peninsula (M1) and the BTH region (M2), respectively, were selected for 14C measurement." in the revised manuscript (page 6, line 15).

-page 7 line 6: why is the conversion to  $f_c$  different between OC and EC and where did you get these numbers? Either a citation is missing or more detailed explanation is needed.

Response and Revisions: Thanks for the comment. The conversion factor was used for 14C data correction for 14C decay during the period between 1950 and the year of measurement. Usually,  $f_m$  value of contemporary carbon source including biogenic and biomass burning ( $f_{m,bio}$ ,  $f_{m,bb}$ , respectively) are large than 1 due to the nuclear bomb in 1950s and 1960s, which were estimated to be 1.06  $\pm$  0.015 and 1.13  $\pm$  0.05 for  $f_{m,bio}$  and  $f_{m,bb}$ , respectively 30 (Zhang et al., 2014). Of them,  $f_{m,bio}$  value was estimated from long term series of 14CO2 measurement from

Schauinsland station (Levin et al., 2010), while  $f_{m,bb}$  was estimated by a tree-growth model (Mohn et al., 2008). In this study,  $f_{m,EC}$  equals  $f_{m,bb}$  given biomass burning is the only non-fossil source for EC, while  $f_{m,oc}$  is adopted as the average of  $f_{m,bio}$  and  $f_{m,bb}$  assuming OC originated equally from biogenic and biomass burning emission. Finally, conversion factors were adopted 1.10 and 1.06 for EC and OC, respectively, considering the steadily declining tendency of 14C after factors estimation study. These factors were same with a previous study (Liu et al., 2014). Based on discussed above, additional explanation and citations have been added in the revised manuscript (page 7, line 1-10).

- Principle for selecting 14C sample – this section is hard to understand

5

Response and Revisions: Thanks for the comment. The section has been corrected for easier understand. As stated
 in the revised manuscript, PMF can better interpret those data close to the average condition of each chemical species. Thus, OC, EC and ratios of them to PM2.5 of each sample in clusters 1 and 3 were examined by mean test. The test ensured that chemical components of selected samples can be better interpreted by the PMF modelling. These selected samples were adopted for 14C measurements. For example, a sample that 90% of its components can be explained by PMF is more representativeness than that 60% can be explained by the model. Finally, few samples
 from a successive synoptic process were selected in the present study.

- Don't know what a prefect synoptic process means, don't know what the accuracy of the model is and that is why using one sample per cluster doesn't seem reasonable.

Response and Revisions: Thanks for the comment. Cluster analysis indicated Shandong Peninsula and BTH region were the major source regions, which were the key consideration in this study. In order to make clear the source 20 information in the two regions, the samples with the air mass originated directly from them (without covering any other areas) were the most optimal choice for 14C measurement. As shown in Figure 2, combined samples (M1 and M2) met this demand. They were collected from January 16th and 18th, 2014, when the first half of air masses were only derived from the south and passed through the Shandong Peninsula and the bottom half were only from the north and passed over the BTH region. Besides, this was a successive process, which was more significant for 25 source apportionment than discontinuous ones. PMF can better explain the samples with their chemical components close to the average level of each chemical species, rather than those samples with outlier of chemical components. For example, a sample that 90% of its components can be explained by PMF is more representativeness than that 60% can be explained by the model. Therefore, mean test could ensure the accuracy of the model. As mentioned above, two combined background samples (M1 and M2) including four independent samples from two major air masses were selected for 14C measurement. Few samples were conducted mainly due to the cost for 14C 30

measurement was really expensive. Sufficient prepare works have conducted before 14C measurement. For example, the concentration, area passed through of independent samples was considered greatly when picking them. In addition, the representative capacity of all samples in two major air masses was examined thoroughly. OC and EC concentration, ratios of OC/PM2.5 and EC/PM2.5 of each sample were compared with that in the corresponding cluster by mean test. Finally, one combined sample was selected per cluster.

**Results**

5

- General characteristics and chemical composition
- Page 9 line 17: is this avr  $\pm$  st. dev and how is the uncertainty in the PM2.5 measurement accounted for? 77 59 = 16 µg/m3 if we take the min and apply 20 µg measurement error, how dose this 20 µg compare to µg/m3?
- 10 Response and Revisions: Thanks for the comment. "The mean concentration of  $PM_{2.5}$  was 77.6 ± 59.3 µg/m3" in the text, of which 77.6 ± 59.3 was avr ± st. dev. The sampling site in this study was a background site, where concentration of  $PM_{2.5}$  was mainly impacted by transportation from other source region. So the concentration value was discrete with the minimum value of 12.68 µg/m3 and the maximum value of 305 µg/m3, showing large standard deviation. As mentioned in the text, acceptable difference among the weighting repetition was less than 20 µg for a
- 15 sampled filter. This value (20  $\mu$ g) was refer to the weight error, and should be divided by the sampling air volume when translated into concentration ( $\mu$ g/m3). According to the average air volume (804 m3) of every independent sample, the concentration corresponding to weight error was 0.024 ± 0.007  $\mu$ g/m3, which was < 1% of the minimum value of 12.68  $\mu$ g/m3 during the sampling.

- Also, lacking consistency in significant digits.

20 Response and Revisions: We appreciate the reviewer's comment. In this study, significant digit for concentration value is three, while that for percentage value is two. It is mainly due to some special values hard to get unified. We have revised this matter throughout the manuscript.

- It would be really nice if the section describing the concentrations and percentage in  $PM_{2.5}$  of the different chemical components is summed into a figure e.g. a pie chart so that the reader can better visualize the relative

25 contribution. Very difficult to follow

Response and Revisions: We appreciate the reviewer's comment. Following the review's comment, a pie chart describing the relative contribution of species in  $PM_{2.5}$  has been added in into the revised manuscript (page 39).

Figure 1 Pie-charts showing the relative contribution of species for  $PM_{2.5}$  in Qimu Island. Note the sum of percentage of identified species in  $PM_{2.5}$  in (a) is 58.58%, while that of (b) is 100% because the percentage is the ratio of every mental element to the total identified mental elements.

5 - Page 10 line 4: citations.

Response and Revisions: We appreciate the reviewer's comment. The citations (Tian et al., 2016; Zhang et al., 2013; Zhao et al., 2013) have been inserted into the text (page 10, line 20).

- In general the names of chemical species should be written out first, so the chemical formula is defined. This is missing throughout the manuscript.

- 10 Response and Revisions: Thanks for the comment. The names of chemical species have been written out when it first appeared, and was then defined in chemical formula in the revised manuscript.
  - Last paragraph belongs to discussion.

Response and Revisions: Thanks for the comment. As mentioned above, results and discussion is in one section.

- Cluster Analysis
- 15 Very difficult to follow which cluster corresponds to which region

Response and Revisions: Thanks for the comment. We have added corresponding region information, e.g. cluster 1 (BTH), cluster 2 (MON), cluster 3 (SDP), behind the clusters appeared along and difficult to follow in the revised manuscript (page 7, line 25).

- "low temperature burning, such as agriculture residue burning, emits more OC compared with high temperature
20 burning, e.g. vehicle exhaust." Citation

Response and Revisions: We appreciate the reviewer's comment. The citations (Cui et al., 2016; Goncalves et al.,

2011) have been inserted into the text (page 13, line 10).

-14C source apportionment

-No errors reported throughout the second paragraph. It is unclear what the uncertainty in the measurement is, which makes it not clear how reliable the measurement actually is.

- 5 Response and Revisions: We appreciate the reviewer's comment. Following the reviewer's opinion, we have added uncertainty to the 14C measurement utilizing the Error Propagation Formula (Liu et al., 2016):  $\delta f_c = \operatorname{sqrt} (\delta f_m^{2} + \delta a^{2})$ , where a is the conversion coefficient caused by nuclear-bomb in the 1950s and 1960s, and  $\delta a$  is adopted as 0.05 according to a previous study (Zhang et al., 2014). The 14C measurement uncertainty result has been inserted in to the revised manuscript (page 14, line 25-30; page 15, line 1-6).
- 10 Page 14 line 27 typo

Response and Revisions: Thanks for the comment. The typo has been corrected.

- Last paragraph should be in discussion.

Response and Revisions: Thanks for the comment. As mentioned above, results and discussion is in one section.

- PMF analysis
- 15 Generally many repeating citations from one author. A lot of major citations are missing.

Response and Revisions: We appreciate the reviewer's comment. Additional Citations (Amil et al., 2016; Bressi et al., 2014; Cappa et al., 2014; Chang et al., 2016; Chen et al., 2015; Choi et al., 2013; Gupta et al., 2015; Hu et al., 2016; Huo et al., 2013; Jing et al., 2016; Khan et al., 2016; Tan et al., 2014; Wang et al., 2002; Zhao et al., 2010; Zheng et al., 2014) have been inserted in the revised manuscript (page 15-19).

20 - Implications for alleviation (lack of discussion)

- While this section presents an interesting discussion on the role of biomass burning, it is only one part of a discussion. The manuscript lacks a comprehensive discussion of results and their importance.

Response and Revisions: Thanks for the comment. As mentioned above, the part has been moved in the section of results and discussion.

25 - Page 19 line 25/26 – unclear

Response and Revisions: We appreciate the reviewer's comment. "this source imposed a larger spatial pattern of  $PM_{2.5}$  pollution in northern areas of China compared with North China" the sentence may be misleading, and has been deleted in the revised manuscript.

- Typos/language

30 Response and Revisions: Thanks for the comment. The typos and language errors have been corrected.

- Discussion on Shandong Peninsula p 20 line 6-14 -citation.

Response and Revisions: Thanks for the comment. Corresponding citation and description have been added in the revised manuscript (page 20, line 26).

- Figures and Tables

-The tables of the manuscript are significantly better than the first version and are much easier to understand now.
 Some of the figures however need improvement. Figure 2 is cut at the bottom and the font of figure 4 is too small to be read. The manuscript can definitely benefit from the addition of figures summarizing the results.
 Response and Revisions: Thanks for the comment. Figures need to improve have been completed in the revised manuscript (page 38, page 41).

[revised manuscript text omitted]

Biomass burning emission should be paidneeds close attention to, because the emissionit has only been only lightly considered in the control program. Indeed, the first national pollution source survey showeddemonstrated that Shandong province is the largest producer of crop stalks, with a production of 132 million tonssuch as wheat and corn, in China in 2007 (Compilation-Committee-of-the-first-China-pollution-source-census, 2011). Of these, The source survey showed a production of 132 million tons in Shandong in 2007 and about 20 million tons-were produced in the Shandong Peninsula (including the cities of Weifang, Yantai, Weihai, Qingdao and Rizhao). Approximately 40% of this production was used aswas household fuel for cooking and heating in the peninsula countryside. The fraction was significantly higher than in western areas of the Shandong province, such as Zibo (9%) and Jinan (8%), and the fraction of open burning of crop residues in the peninsula (3%). The fraction of biomass open burning in the peninsula was also higher than theits average fraction (1.5%)in Shandong province in 2007 (Compilation-Committee-of-the-first-China-pollution-source-census, 2011)-. Generally, emissions

from agricultural field burning are mainly concentrated in the harvest season and contribute significantly to regional haze and smog events in the region, which have attracted special concern (Feng et al., 2012; Zong et al., 2015; Wang et al., 2014). Despite this Even so, open burning emission has been considered only aswas regarded a minor source sectorcontribution in the control program. HouseholdIn addition, household emission of agricultural waste, another largerimportant source, are released continuously for regional PM2.5, is continuous or semi-continuously, and continuous. It can also induce PM2.5 pollution on a regional scale, which has also been despised or ignored (Zhang and Cao, 2015).-

Open burning is not fully controlled in China, although Since the 1990s, the government has enacted a series of regulations to prohibit field burning since the 1990s and strengthened the force of 10 open burning. However, it is not fully controlled in China although its supervision is strengthened recently. The most basic reason for continued burning is the lack of a reasonable alternative to utilize or dispose of huge amounts of agricultural waste each year. In the current scenario, some agricultural wastes are collected and stored as fuel for household cooking and heating, and while others are rapidly removed consumed by open burning in fields for the next planting during harvest season. 15 Although farmers know that such use andthis disposal of agricultural residues areis harmful to the environment, they still tend to use agricultural wastes as household fuel and burn wastes in fields, do mainly due to the low costs of the methodsmethod. A more permanent solution would be to find higher economic value inof agricultural wastes via development of renewable techniques. IndeedIn fact, agricultural wastes can be utilizedused to produce many kinds of renewable energies, such as 20 biogas, feedstuffs, biochar, bioethanol, and bio-succinic acid. China has provided enacted relevant energy regulations, legislation, and policy initiatives for rural renewable energy (Li et al., 2015). The government has also encouraged and sustained the development of the renewable energy industry to increase the demand for raw feedstocks. Through these efforts, China has achieved some success in 25 renewable development in rural areas, but. However, these efforts are not an effective solution to the problem of surplus crop waste, because the costs and benefits cannot yet of renewable energy could not be offset. For instance, Zhangziying, a town located in the eastern area of the Daxing district of Beijing, has developed household biogas and straw gas since the 1980s, but in 2011. But renewable energy only made up approximately 10% of household energy consumption in 2011, much lower than the fraction of coal (30%) (Li et al., 2015). Before the achievement of high economic value,

30

except for the ban on crop straw burning, the government should compensate farmers to collection collecting crop residues as feedstocks of renewable energy, rather than except for the ban on crop straw burning in fields or households (Shi et al., 2014a). The revenue from the subsidy and the sale of crop residues could help alleviate economic burdens on farmers, so they canwhich promote them use clean energy, such as electricity, liquefied petroleum gas, biogas, etc., for household consumption (Kung and Zhang, 2015). These efforts will not only significantly improve air quality, but also make famers learn the convenience of clean energy and wake from agricultural residue burning.—

**5 Conclusions 4 Summary and conclusion**

10

5

During the sampling period, the average  $PM_{2.5}$  concentration was 77.6 ± 59.3 µg m-3, and  $SO_4^{2-}$  concentration was the highest of any constituentamong all constituents, with a mean of 14.2 ± 18.0 µg m-3, followed by  $NO_3^{-1}$  (11.9 ± 16.4 µg m-3), OC (6.85 ± 4.81 µg m-3), EC (4.90 ± 4.11 µg m-3), and  $NH_4^{+1}$  (3.11 ± 2.14 µg m-3). The fractions of  $SO_4^{2-}$ ,  $NO_3^{-1}$  and  $NH_4^{+1}$  to  $PM_{2.5}$  were obviously higher than those in metropolises (e.g. Beijing and Tianjin) within North China, while fractions of carbonaceous species were markedly lower; these showed regional pollution signals.

15

20

25

More than half of air masses during the sampling period were from the BTH region, followed by air masses from Mongolia (35%) and the Shandong Peninsula (11%). The concentrations of PM2.5 and most of the species carried by the air masses from the BTH region and the Shandong Peninsula were comparable (p > 0.05), and they occurred in statistically greater concentrations than those carried by the air masses from Mongolia (p < 0.01). The PM2.5 had an obvious signal of biomass burning emission, characterized by a high OC/EC ratio, low NO3-/nss-SO42- ratio and high nss-K+ concentration whenfor the air masses camecoming from the Shandong Peninsula. In contrast, the PM2.5 carriedtested 
[revised manuscript text omitted]

| Species                       | Mean ±std.                            | Ra

---

## Author Response (AR3)

**Response to Editor and Reviewer's Comments and Suggestions - doi: 10.5194/acp-2016-97**

Comments to the Author:

The authors should address a few remaining minor comments made by Referee # 1, either by revising the manuscript accordingly or explaining why further changes are not needed.

Once the authors have responded to the Referee's concern, I will decide on publication.

Dear Editor,

We are thankful very much to you and the anonymous reviewer for these useful comments and suggestions. We have addressed the remaining comments and revised the manuscript accordingly. Listed below are our point-by-point responses (blue) to the reviewer's comments (black).

Thanks again.

Best regards,

Dr. Chongguo Tian

The authors have done a great job in adding the requested information and clarifying the manuscript. As a result, it is has once again undergone even more improvements than after the first round of revisions. However, I am still not convinced that the radiocarbon discussion is clear and reliable. The reason for that is that major conclusions are drawn for biomass burning from region represented by sample M1, which in the text itself is described as "not ideal" (P 9, L 21).

Response and Revisions: Thanks for the comments. "Not ideal" in the text may be misleading. Mean test was adopted to identify the representativeness of every sample. Finally, combined samples M1 and M2 were selected for radiocarbon analysis. Compared with M2, M1 may have slight lower representativeness since OC and EC concentrations were relative higher than cluster 3. However, OC/$PM_{2.5}$, EC/$PM_{2.5}$ ratios in M1 had no statistical difference (95%) from cluster 3, which were more important than concentration and could assure the validity for $PM_{2.5}$ source apportionment in this study. In addition, the area passed through, continuity, etc of independent samples were also considered greatly when picking them. M1 were totally derived from Shandong Peninsula with a successive process, implying more dramatic than insular samples. Overall, M1 was the most perfect sample in cluster 3 for radiocarbon measurement based on the above consideration. Thus, "M1 was not ideal" in the text may be misleading. We have deleted it and revised as "M1 may have slight lower representativeness" (page 9, line 23).

Further, radiocarbon samples on the order of a few hundred micrograms are very small and the C isotopes very difficult to measure. Therefore source apportionment results using 14C usually have a much higher uncertainty than 1-3%, which is reported here. Instrumental uncertainties and the AMS capacity are still not clear.

Response and Revisions: Thanks for the comments. The uncertainty about radiocarbon analysis reported here was 1-4%, slight higher than 1-3%, but well within the previous uncertainty range (1-10%) (Liu et al., 2016a; Liu et al., 2014; Liu et al., 2016b; Zhang et al., 2014; Zotter et al., 2014). This suggests the accuracy of our experiments. Carbon species were conducted by two-step thermal separation and trapping system, which is at Guangzhou Institute of Geochemistry, Chinese Academy of Science, and then undergone graphite targets for AMS analysis by graphitization line with the sealed tube zinc reduction method (200 mg, Alfa Asear, 1.5-3 mm, 99.99%). This method could achieve a precision of 2-3‰ (Xu et al., 2007). $^{14}$C analysis was carried out at the updated PKUAMS facility, which sensitivity was less than $6 \times 10^{-15}$ and the precision and accuracy of $^{14}$C measurement for the samples was better than 0.5% (Liu et al., 2000; Vogel, 1992). Besides, $^{14}$C results here were corrected by $^{13}$C value with the uncertainty less than 0.3%. Additional information about Instrument and the AMS capacity has been added in the revised manuscript (page 6, line 27; page 7, line 1).

I think that the idea of the authors and the way they use $^{14}$C to verify the PMF results is corrected, however the $^{14}$C results are suggestive rather than conclusive and I think the level of uncertainty is highly underestimated. If however this was made clear in the manuscript, it would be much easier to understand and interpret the results.

Response and Revisions: Thanks for the comments. The present study proposed the minimum deviation between the results from $^{14}$C and PMF model could be adopted as a criterion to select a more reliable solution for source apportion of fine particles. In addition, this method could also be applied to CMB model or other isotopes (e.g. $^{13}$C, $^{15}$N and $^{35}$S), which exhibits great scientific significance. $^{14}$C measurement was not the only method to apportion the source of $PM_{2.5}$ in this study. Chemical analysis of various species in $PM_{2.5}$ and PMF model were also discussed in detail here. The three portions all indicated the same conclusion, implying the accuracy of $^{14}$C in present manuscript.

Also, despite requesting for more citation during the last two rounds of reviews, there are still citations missing for e.g. in section 3.4.

Response and Revisions: We appreciate the reviewer's comment. More references (Dorado et al., 2003; Gelencser et al., 2007; Wang et al., 2013; Zhou et al., 2016; Zong et al., 2015) have been added in the revised

manuscript.

Implication for PM alleviation and also there are still a few typos throughout the manuscript. Overall, the manuscript is interesting and informative, but there are still problems that need to be addressed.

Response and Revisions: We appreciate the reviewer's comment. The typos throughout the manuscript have been examined and corrected in the revised manuscript.

[revised manuscript text omitted]